JCB Journal of Cell Biology

# ATG8-dependent LMX1B-autophagy crosstalk shapes human midbrain dopaminergic neuronal resilience

Natalia Jiménez-Moreno[1], Madhu Kollareddy[1], Petros Stathakos[1], Joanna J. Moss[1], Zuriñe Antón[1], Deborah K. Shoemark[2], Richard B. Sessions[2], Ralph Witzgall[3], Maeve Caldwell[4], and Jon D. Lane[1]

The LIM homeodomain transcription factors LMX1A and LMX1B are essential mediators of midbrain dopaminergic neuronal (mDAN) differentiation and survival. Here we show that LMX1A and LMX1B are autophagy transcription factors that provide cellular stress protection. Their suppression dampens the autophagy response, lowers mitochondrial respiration, and elevates mitochondrial ROS, and their inducible overexpression protects against rotenone toxicity in human iPSC-derived mDANs in vitro. Significantly, we show that LMX1A and LMX1B stability is in part regulated by autophagy, and that these transcription factors bind to multiple ATG8 proteins. Binding is dependent on subcellular localization and nutrient status, with LMX1B interacting with LC3B in the nucleus under basal conditions and associating with both cytosolic and nuclear LC3B during nutrient starvation. Crucially, ATG8 binding stimulates LMX1B-mediated transcription for efficient autophagy and cell stress protection, thereby establishing a novel LMX1B-autophagy regulatory axis that contributes to mDAN maintenance and survival in the adult brain.

## Introduction

Parkinson's disease (PD) is caused by the loss of midbrain dopaminergic neurons (mDANs) and the corresponding disruption of the nigrostriatal dopaminergic pathway (Kalia and Lang, 2015; Surmeier et al., 2017), with α-synuclein–enriched Lewy bodies, mitochondrial dysfunction with increased neuronal oxidative stress, and failing autophagy cytoplasmic quality control being recognized features (Karabiyik et al., 2017; Surmeier et al., 2017). Autophagy protects cells through the delivery of toxic, damaged, or redundant cytoplasmic cargo to the lysosome for degradation and recycling. In macroautophagy (referred to herein as "autophagy"), a specialized organelle—the autophagosome—is assembled de novo through the sequential actions of the products of conserved autophagy-related genes (ATGs), and this organelle sequesters cargo in a selective or non-selective fashion for trafficking and fusion with late endosomes/lysosomes to form a degradative compartment (Lamb et al., 2013). Accumulating evidence suggests that mDANs are particularly sensitive to autophagy deficits (Friedman et al., 2012; Karabiyik et al., 2017; Kett et al., 2015; Sato et al., 2018), and correspondingly, upregulation of the autophagy and/or autolysosomal systems protects mDANs against α-synuclein toxicity in vivo (Decressac et al., 2013). Thus, understanding how autophagy responses are coordinated in mDANs remains a key objective.

The LIM homeodomain transcription factors, LMX1A and LMX1B, are essential determinants of mDAN differentiation and maintenance (Doucet-Beaupre et al., 2015). With high sequence homology (homeodomain [100%]; LIMA [67%]; LIMB [83%]; Doucet-Beaupre et al., 2015), LMX1A and LMX1B have distinct tissue expression patterns and distinct roles (Doucet-Beaupre et al., 2015): LMX1A regulates neurogenesis and neural fate in the ventral mesencephalic floor plate, and has roles in the inner ear (Doucet-Beaupre et al., 2015); LMX1B is essential for brain development, particularly dorsoventral patterning and serotonergic axonal architecture, but also coordinates aspects of eye, limb, and kidney development (Donovan et al., 2019; Doucet-Beaupre et al., 2015; Kitt et al., 2022; Wever et al., 2019). Depleted mDAN populations are common features of *Lmx1a* null and recessive mutant (dreher)mice (Deng et al., 2011; Pollack et al., 2019), meanwhile *Lmx1b* ablation reduces mDAN density through failure to establish the isthmic organizer, a structure that controls midbrain-hindbrain regional identity through regulated secretion of Fgf8 and Wnt1 (Doucet-Beaupre et al., 2015). Despite this functional divergence, *Lmx1a* and *Lmx1b* partially compensate for one another with respect to mDAN specification in the mouse (Doucet-Beaupre et al., 2015; Pollack et al., 2019). Away from the midbrain, mutations in the LIM and/

[1]Cell Biology Laboratories, School of Biochemistry, University of Bristol, Bristol, UK;   [2]School of Biochemistry, University of Bristol, Bristol, UK;   [3]Institute for Molecular and Cellular Anatomy, University of Regensburg, Regensburg, Germany;   [4]Trinity College Institute for Neuroscience, Trinity College Dublin, Dublin, Ireland.

Correspondence to Jon D. Lane: jon.lane@bristol.ac.uk.

or homeodomain of LMX1B cause Nail-Patella syndrome, a human disease characterized by skeletal developmental abnormalities, chronic nephropathy, and open-angle glaucoma (Choquet et al., 2018; Harita et al., 2017; Shiga et al., 2018). Systemic *Lmx1b* knockdown in mouse is associated with skeletal and kidney defects consistent with Nail-Patella syndrome pathology, leading to neonatal lethality ~24 h after birth (Chen et al., 1998); meanwhile, conditional podocyte *Lmx1b* ablation triggers proteinuria linked to podocyte actin disorganization and slit diaphragm failure (Burghardt et al., 2013). Finally, *LMX1B* has been identified as an osteoarthritis susceptibility gene (Henkel et al., 2022; Tachmazidou et al., 2019), thus broadening interest in its roles and regulation in human ageing pathology.

While the influence of LMX1A/LMX1B actions during mDAN specification and maturation is well established, conditional knockout (KO) mouse models have since demonstrated a need for sustained *Lmx1a/Lmx1b* expression to support mDANs in the developed midbrain. In the mouse, targeted *Lmx1a/Lmx1b* ablation triggered mDAN decline associated with neuropathological (e.g., α-synuclein–positive, distended axonal terminals) and behavioral abnormalities consistent with PD (Doucet-Beaupre et al., 2016; Laguna et al., 2015). With this in mind, it is noteworthy that *Lmx1a* expression declines with age in the mouse brain (Doucet-Beaupre et al., 2016; Laguna et al., 2015), while patient brain LMX1B levels have been reported to inversely correlate with PD progression (Laguna et al., 2015; Xia et al., 2016). In addition, *LMX1A/LMX1B* polymorphisms have been linked (albeit weakly) to PD (Bergman et al., 2009). Thus, maintaining LMX1A/LMX1B in the adult brain is likely to be important for protection against PD-associated mDAN decline, while boosting expression and/or activities of these transcription factors may be of therapeutic benefit.

Autophagy capacity is regulated by diverse families of transcription factors acting in different tissues (Fullgrabe et al., 2016; Fullgrabe et al., 2014), but those that control autophagy gene expression in human mDANs remain elusive. Notably, in the conditional *Lmx1a/Lmx1b* KO mouse model, post-mitotic mDAN functional decline was shown to be associated with dysregulated autophagy, reduced mitochondrial function, and elevated mitochondrial oxidative stress (Doucet-Beaupre et al., 2016; Laguna et al., 2015). Indeed, a similar pattern has emerged in studies of *Lmx1b* deficiency in adult 5-HT neurons (Kitt et al., 2022). This argues that LMX1A/LMX1B might contribute to the expression of autophagy and mitochondrial quality control genes in specific classes of neurons, including mDANs. Using human cell lines and induced pluripotent stem cell (iPSC)–derived mDANs, we demonstrate here that LMX1A and LMX1B are indeed autophagy transcription factors that provide protection against PD-associated cellular stress in vitro. Importantly, both LMX1A and LMX1B bind to autophagy ATG8 family members, with LMX1B interactions dependent on a conserved region C-terminal to the homeodomain. Crucially, we show that binding to ATG8 proteins stimulates LMX1B-regulated transcription of important target genes and protects against neuronal stress. These data highlight ATG8s as novel LMX1B cofactors, thereby revealing an intriguing layer of mechanistic interplay between

cell stress response pathways with implications for mDAN function and survival in the PD brain.

## Results

### LMX1A and LMX1B are autophagy transcription factors

*LMX1A* and *LMX1B* emerged as distinct paralogs alongside the development of more complex chordate brain architecture (Holland, 2015; Fig. S1, A and B). With data linking LMX1A/LMX1B with the autophagy system and the enhancement of mDAN maintenance/protection (Laguna et al., 2015), we used bioinformatics to search for potential LMX1A/LMX1B-targeting A/T-rich FLAT elements in autophagy gene promoter regions (see Materials and methods), to provide evidence for LMX1A/LMX1B-mediated autophagy transcriptional control in humans. As anticipated, FLAT elements were detected in known LMX1A/LMX1B transcriptional target gene promoters including *COL4A3* and *IFNB1*—both implicated in joint and/or kidney maintenance (Morello et al., 2001; Rascle et al., 2009)—and in the mDAN control genes, *NURR1*, *PITX3*, and *TH* (Levesque and Doucet-Beaupre, 2013; Table S1). FLAT elements were also detected in the promoters of mitochondrial *NDUFA2*, *COX1*, and *NDUFV1* (LMX1B only), although the latter two were not shown to be affected by *Lmx1a/Lmx1b* ablation in the mouse brain (Doucet-Beaupre et al., 2016; Table S1). Importantly, putative FLAT elements were also identified in the promoters of genes controlling autophagy initiation (e.g., *ULK1/2*; *ATG13*; *ATG14*; *WIPI2*; *UVRAG*) and autophagosome expansion (e.g., *ATG3*; *ATG5*; *ATG7*; *ATG10*; *ATG16L1*), in the promoters of autophagy receptors (e.g., *OPTN*; *NDP52*; *TAX1BP1*; but not *SQSTM1/p62*), and in both *PINK1* and *PRKN* (Table S1). The autophagy-related transcription factors *TFEB* and *ZKSCAN3* also harbored putative promoter FLAT elements for both LMX1A and LMX1B, while the hypoxia-responsive transcription factors *NRF1* and *NRF2* were positive for LMX1B only (Table S1). This places LMX1A and LMX1B in the context of a broad transcriptional network for cytoplasmic quality control with the potential to protect against diverse physiological cellular stresses (Fullgrabe et al., 2016; Fullgrabe et al., 2014).

Guided by the bioinformatics data, we selected a panel of candidate genes for LMX1B chromatin immunoprecipitation (ChIP) in human HEK293T kidney cells that express endogenous LMX1B (Burghardt et al., 2013). Using an arbitrary twofold cutoff for targets of interest, promoter occupancy was confirmed for several core autophagy genes, including *ULK1*, *ATG3*, *ATG16L1*, *UVRAG*, as well as the autolysosomal transcription factor *TFEB* and the receptors and/or mitophagy genes, *NDP52*, *OPTN*, and *PINK1* (Fig. 1 A). The mDAN differentiation genes *ABRA*, *NURR1*, and *PITX3* were also identified in LMX1B ChIP of HEK293T cells (Fig. 1 A). To provide experimental support for its role as a human autophagy transcription factor, we measured candidate autophagy gene expression by quantitative real-time PCR (qRT-PCR) in HEK293T cells depleted for LMX1B using siRNA (Fig. 1 B). Expression of several autophagy genes including *ULK1*, *ATG7*, *ATG16L1*, *UVRAG*, and *TFEB* was significantly reduced following LMX1B suppression, as was expression of the selective autophagy/mitophagy receptors *NDP52* and *OPTN*, and the mitophagy regulator and early-onset PD gene, *PINK1* (Fig. 1 B). An identical

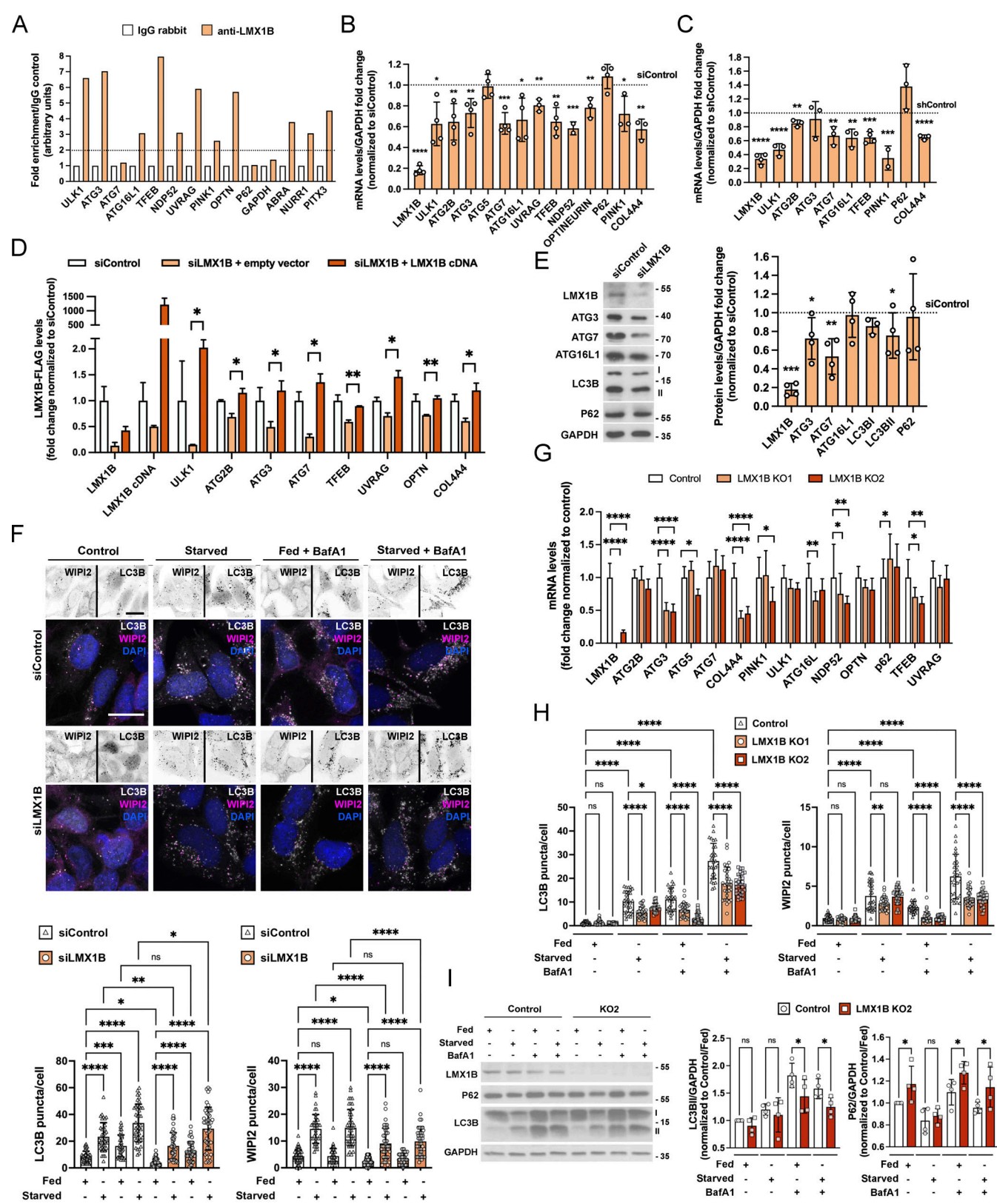

Figure 1. **LMX1B is an autophagy transcription factor in HEK293T cells. (A)** ChIP qPCR analysis of LMX1B promoter occupancy in HEK293T cells. This experiment was repeated three times with consistent results. **(B)** qRT-PCR analysis of selected genes in HEK293T cells transfected with siLMX1B smartpool or non-targeting siControl. mRNA levels were normalized to GAPDH. Mean ± SD (*n* = 3); Student's *t* test: *P < 0.05, **P < 0.01, ***P < 0.001 vs. siControl. **(C)** qRT-PCR analysis of selected genes in HEK293T cells transfected with LMX1B shRNA or non-targeting shControl. mRNA levels were normalized to GAPDH. Mean ± SD (*n* = 3); Student's *t* test: **P < 0.01, ***P < 0.001, ****P < 0.0001 vs. shControl. **(D)** qRT-PCR analysis of rescue of LMX1B suppression in HEK293T cells

double co-transfected with LMX1B smartpool siRNA and codon optimized, siRNA-resistant LMX1B plasmid (or empty vector control). Values are mean ± SD of triplicate treatments from a single knockdown/rescue experiment, representative of >3 experiments with consistent results, and were normalized to siRNA control. Student's *t* test: *P < 0.05 and **P < 0.01; siRNA + empty vector vs. siRNA + LMX1B. **(E)** LMX1B siRNA reduces expression of key autophagy proteins. Left, example immunoblots; right, quantitation relative to GAPDH levels. Molecular weight markers are shown in kD. Mean ± SD; Student's *t* test: *P < 0.05, **P < 0.01, ***P < 0.001 vs. siControl. **(F)** Imaging-based autophagic flux assay in GFP-LC3B HEK293 cells transfected with non-targeting siControl or siLMX1B, stained with anti-WIPI2 antibodies. To the top, selected example inverted single channel images with color overlays below (LC3B is depicted in grayscale; WIPI2 is depicted in magenta; DAPI is depicted in blue). Cells were counterstained with DAPI (blue). Bar = 20 μm. To the bottom, puncta counts for GFP-LC3B and WIPI2 in siControl and siLMX1B-treated HEK293 cells in full nutrients (fed) or following 2 h starvation in the presence or absence of BafA1 (20 nM, 2 h). Mean ± SD of >35 cells from three independent experiments; one-way ANOVA followed by Fisher's least significant difference (LSD) test for planned comparisons: *P < 0.05, **P < 0.01, **P < 0.01, ***P < 0.001, ****P < 0.0001. **(G)** qRT-PCR analysis of autophagy and selected gene expression in two LMX1B KO HEK293T clones (KO1 and KO2). Data show means ± SD (*n* = 4 samples from two early passage cultures); two-way ANOVA followed by Tukey's multiple comparison post-hoc test against control cells: *P < 0.05, **P < 0.01,****P < 0.0001. **(H)** Imaging-based autophagic flux assay in LMX1B CRISPR KO HEK293T cells stained with anti-LC3B and anti-WIPI2 antibodies. Puncta counts for LC3B and WIPI2 in Control and LMX1B KO HEK293T cells in full nutrients (fed) or following 2 h starvation in the presence or absence of BafA1 (20 nM). Mean ± SD of 30 fields of cells from three independent experiments, counting ~50 cells per field; one-way ANOVA followed by Fisher's LSD test for planned comparisons: *P < 0.05, **P < 0.01, ****P < 0.0001. **(I)** Immunoblot showing LMX1B levels in CRISPR KO cell line KO2 compared to controls, with analysis of p62 levels and LC3B lipidation in fed and starved conditions (2 h), in the absence and presence of BafA1 (20 nM). Molecular weight markers are shown in kD. Densitometry quantitation is shown to the right (*n* = 4); one-way ANOVA followed by Fisher's LSD test for planned comparisons: *P < 0.05.

pattern of suppression was seen using shRNA against LMX1B (Fig. 1 C), with the exception of *ATG3*, perhaps due to the relative efficiencies of LMX1B suppression. *SQSTM1/p62* was not affected by LMX1B suppression, confirming bioinformatics predictions, but neither was *ATG5* despite the presence of putative FLAT element(s) for this gene (Table S1). *ATG7* promoter occupancy was not detected by ChIP using our probes (Fig. 1 A), but ATG7 transcripts were significantly lower in LMX1B suppressed cells (Fig. 1, B and C). Importantly, the expression of several autophagy genes could be rescued in siLMX1B-treated HEK293T cells by overexpression of siRNA-resistant LMX1B (Fig. 1 D).

At the level of protein expression, the effects of LMX1B suppression were corroborated for ATG3 and ATG7 (significantly reduced), although expression of ATG16L1 was not affected (Fig. 1 E). Levels of lipidated LC3II were significantly lower in siLMX1B-treated HEK293T cells (Fig. 1 E), indicative of reduced basal autophagy. Meanwhile, analysis of LC3B and WIPI2 puncta numbers in HEK293T cells in full nutrients or in starvation conditions in the absence or presence of Bafilomycin A1 (BafA1) demonstrated reduced autophagy flux potential in siLMX1B-treated cells (Fig. 1 F). Using CRISPR editing, we next generated two LMX1B KO HEK293T clonal cell lines. Expression of FLAT element containing autophagy and associated genes was reduced in both lines (Fig. 1 G; levels tended to return to baseline levels over multiple passages [data not shown]), and autophagy flux was suppressed as measured by immunostaining for LC3B and WIPI2 puncta (Fig. 1 H). LC3B lipidation was also reduced in fed and starved cells in the presence of BafA1, while p62 levels were higher in LMX1B KO cells under basal conditions and in the presence of BafA1 (Fig. 1 I, showing data for clone KO2 only). Together, these data demonstrate that LMX1B contributes to the control of autophagy gene expression in HEK293T cells, and that LMX1B depletion dampens basal/housekeeping and nutrient stress-induced autophagic flux responses.

### LMX1A and LMX1B regulate autophagy and protect human iPSC-derived mDANs in vitro

To confirm their potential as autophagy transcription factors in the human midbrain, we prepared mDANs from the NAS2

(normal α-synuclein) iPSC line (Devine et al., 2011), using a monolayer protocol that generates high numbers of mDANs expressing midbrain dopaminergic markers, including TH, LMX1A, LMX1B, and FOXA2 (Stathakos et al., 2021). Immunofluorescence imaging indicated that our cultures comprised ~75% neurons, of which ~50% expressed TH, with ~80% of these expressing LMX1A and/or LMX1B (Fig. 2 A). LMX1A and LMX1B mRNA levels peaked and remained high from day 20 of mDAN differentiation, correlating with expression of neuronal βIII tubulin (TUJ1) and the dopaminergic transcription factor, NURR1 (Fig. 2, B–E). Based on an arbitrary twofold cut-off, ChIP analysis of iPSC mDAN cultures indicated promoter occupancy for the autophagy genes *ATG7* (LMX1A/LMX1B), *ATG16L1* (LMX1A only), and *ULK1* (LMX1B only), and the autophagy adaptors *NDP52* and *OPTN* (Fig. 2, F and G). Dopaminergic neuronal control genes *NURR1* and *PITX3* were also positive for both LMX1A and LMX1B, with *ABRA* being positive for LMX1B only (Fig. 2 G) as expected (Burghardt et al., 2013; Table S1). To test the impact of their suppression in human mDANs in vitro, we generated lentiviruses expressing GFP (synapsin promoter) with LMX1A/LMX1B shRNA (U6 promoter; Fig. 2 H). 6 d after viral transduction in day >30 cultures, LMX1A and LMX1B expression in mDANs was significantly reduced (Fig. 2 H), with suppression of each lowering expression of the other (Fig. 2 I). Importantly, shRNA suppression of either LMX1A or LMX1B significantly reduced levels of ULK1, ATG2B, ATG7, ATG16L1, NDP52, OPTN, PINK1, and TFEB transcripts in mDANs, but ATG3 and p62 were not affected (Fig. 2 I). Expression of the dopaminergic neuronal genes *NURR1* and *PITX3* was significantly reduced in both conditions as expected (Doucet-Beaupre et al., 2015; Nakatani et al., 2010), but *TH* and *TUJ1* expression levels were not altered within this timeframe (Fig. 2 I). Expression of *MSX1* was significantly reduced only in the LMX1A suppressed cells (Fig. 2 I), consistent with previous reports that *MSX1* is controlled by LMX1A in the midbrain (Andersson et al., 2006; Cai et al., 2009), but as LMX1B suppression concomitantly reduced LMX1A mRNA levels (Fig. 2 I), this implied that residual LMX1A is sufficient to maintain MSX1 levels in this setting.

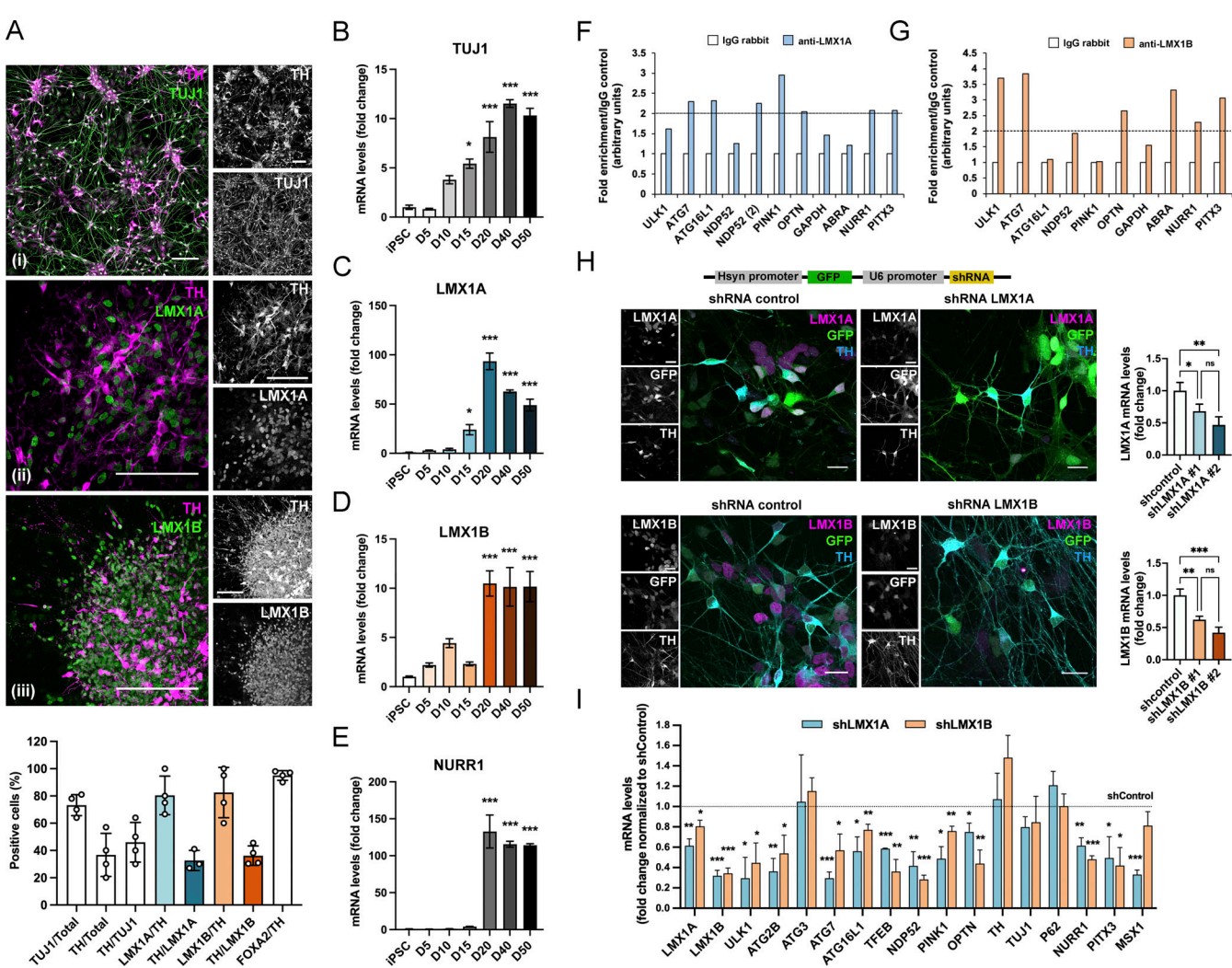

**Figure 2. LMX1B is an autophagy transcription factor in iPSC-derived human mDANs. (A)** Immunofluorescence images and quantitation of LMX1A/LMX1B and dopaminergic marker expression in human iPSC-derived mDANs (imaged at: [i] D40; [ii, iii] D20). Cells were stained for anti-TH (magenta) and either anti-TUJ1 (green, [i]), anti-LMX1A (green, [ii]), or anti-LMX1B (green, [iii]). Bars = 100 μm. Data show counts of cells positive for the marker combinations shown (mean ± SD; n = 4 experiments). **(B–E)** qRT-PCR analysis of (B) TUJ1, (C) LMX1A, (D) LMX1B, and (E) NURR1 in iPSC-derived mDNAs differentiated at days 5, 10, 15, 20, 40, and 50 (or iPSC control). Levels were normalized to GAPDH. Mean ± SEM of mDAN cultures (n = 3) plated from a single neuralization; one-way ANOVA followed by a Dunnett's multiple comparisons test. *P < 0.05, ***P < 0.001 vs. iPSC stage. **(F and G)** ChIP q-PCR analysis of (F) LMX1A and (G) LMX1B promoter occupancy in human mDANs using antibodies to isolate endogenous protein/DNA complexes. **(H)** shRNA suppression of LMX1A or LMX1B expression in iPSC-derived mDANs. To the top, a schematic representation of the pRRL plasmid construct expressing GFP under hsyn control and shRNA for LMX1A or LMX1B (or a non-target shRNA control) under U6 promoter control. Below are example fields of mDANs stained for anti-LMX1A (top, magenta) or anti-LMX1B (bottom, magenta) and anti-TH (cyan) after transduction with hsyn-GFP-U6-shControl, hsyn-GFP-U6-shLMX1A, or hsyn-GFP-U6-shLMX1B (green). To the right, LMX1A and LMX1B qRT-PCR quantitation after viral transduction at day 30–45 (normalized to GAPDH) of two different shRNAs for LMX1A and LMX1B (n = 3). Mean ± SD; one-way ANOVA followed by Tukey's multiple comparison post-hoc test: *P < 0.05; **P < 0.001; ***P < 0.001 vs. shRNA control. Bars = 20 μm. **(I)** qRT-PCR analysis of candidate gene expression in iPSC-derived mDANs (day 30–45) following hsyn-GFP-U6-shControl/shLMX1A/shLMX1B lentiviral transduction (normalized to GAPDH). Means ± SD (n = 3); Student's t test: *P < 0.05, **P < 0.01, ***P < 0.001 vs. shRNA control.

LMX1A and LMX1B suppression significantly reduced the numbers of WIPI2-positive autophagosome assembly sites in the cell bodies of mature (day >30) TH- and GFP-positive mDANs, consistent with a dampened basal autophagy response (Fig. 3 A). Suppression also elevated mitochondrial ROS levels (MitoSox; Fig. 3 B), and negatively impacted on oxidative phosphorylation, causing reduced basal oxygen consumption rates, maximal respiration, and spare capacity (Fig. 3, C and D; also reported in isolated mouse mDANs [Doucet-Beaupre et al., 2016]). Despite this, basal population cell death levels measured by caspase-3

assay were not affected (Fig. 3 E). Thus, mature mDANs tolerate the absence of either transcription factor over short timeframes in vitro. Consistent with essential roles during mDAN differentiation and maturation, the timing of LMX1A/LMX1B suppression in human iPSC-derived mDANs was critical (Fig. S1, C–E): neurite arborization was not affected in mature mDANs plated at day >35 (Fig. S1 D), but neurites collapsed following a ~12-h lag when LMX1A/LMX1B was suppressed in immature day <17 mDANs (Fig. S1 E). This was associated with marked changes in expression of dopaminergic genes (*TUJ1, TH, NURR1, MSX1*) in

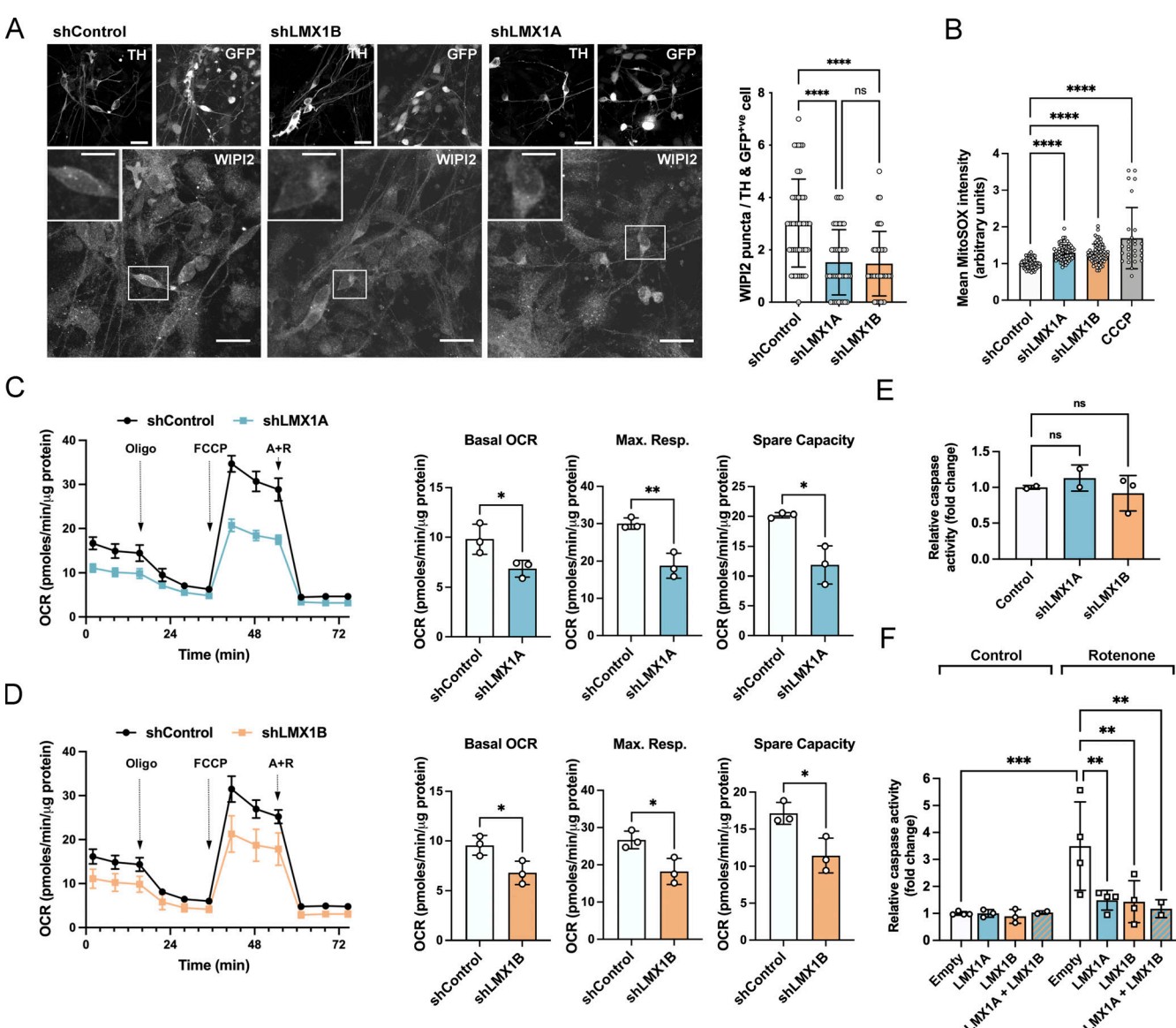

**Figure 3. LMX1A and LMX1B control basal autophagy and mitochondrial function to protect mDANs. (A)** Representative immunofluorescence images (left) and quantitation (right) of basal WIPI2 puncta in TH$^{+ve}$/GFP$^{+ve}$ iPSC-derived mDAN cell bodies (day 30–45) following hsyn-GFP-U6-shControl/shLMX1A/shLMX1B lentiviral transduction. Mean ± SD (>30 neurons/condition; three independent experiments); one-way ANOVA followed by a Tukey's multiple comparison post-hoc test: ****P < 0.0001 vs. shRNA control. Bars = 20 µm, full field; 10 µm, inset zoom. **(B)** MitoSOX intensity in TH-positive iPSC-derived mDANs transduced with hsyn-GFP-U6-shControl/shLMX1A/shLMX1B (day 30–50). Cells were treated with carbonyl cyanide chlorophenylhydrazone (10 µM, 4 h) as a positive control. Mean ± SD of individual cell bodies from four independent experiments; one-way ANOVA followed by a Tukey's multiple comparison post-hoc test: ****P < 0.0001. **(C and D)** Seahorse mitochondrial respiration profiles in iPSC-derived mDAN cultures (day 30–50) transduced with (C) hsyn-GFP-U6-shRNA LMX1A or (D) hsyn-GFP-U6-shRNA LMX1B lentiviruses. Pairwise comparisons were performed against non-targeting shRNA control. Oxygen consumption rate (OCR) was normalized to protein levels after sequential exposure to oligomycin (Oligo; 1 µM), FCCP (1 µM), and rotenone/antimycin (R+A; 0.5 µM) as indicated (n = 3). Mean ± SD; Student's t test: *P < 0.05, **P < 0.01. **(E)** Basal mDAN cell death analysis following hsyn-GFP-U6-shControl/shLMX1A/shLMX1B lentiviral transduction (day 25–50 mDANs). Fluorometric caspase assay normalized to total protein (n = 2–3). Mean ± SD; one-way ANOVA followed by Fisher's LSD test for planned comparisons. **(F)** Rotenone-induced cell death in iPSC-derived mDANs (day 25–50) transduced with TRE-empty (control), TRE-LMX1A, and/or TRE-LMX1B; expression was induced by DOX (500 ng/ml, 3 d). mDANs were treated with rotenone (15 µM) for 24 h. Caspase activity was normalized to total protein and measured relative to empty vector control (n = 2–4). Mean ± SD; one-way ANOVA followed by Fisher's LSD test for planned comparisons: **P < 0.01, ***P < 0.001.

day <17 cultures, with LMX1B suppression associated with compensatory LMX1A elevation and enhanced NURR1 mRNA expression (Fig. S1 F).

Using a doxycycline (DOX) inducible LMX1A/LMX1B lentiviral system (Fig. S2), we found that acute overexpression of LMX1B alone or in combination with LMX1A increased the expression of several key autophagy-associated genes in human mDANs (notably, *ULK1, ATG3, TFEB*; Fig. S2 E). LMX1B overexpression concomitantly increased LMX1A levels, but the converse was not observed (Fig. S2 E). The significance of these

observations became clear when we tested the effect of LMX1A/LMX1B overexpression during stress-induced cell death in mDANs treated with the mitochondrial poison, rotenone (a complex I inhibitor widely used as a PD-inducing model, e.g., Betarbet et al., 2000). In our mDAN cultures, induced LMX1A and/or LMX1B overexpression significantly reduced rotenone-induced cell death compared to the empty vector control (Fig. 3 F). As LMX1A overexpression by itself did not significantly induce expression of the autophagy genes we tested (Fig. S2 E; also reported in the mouse [Laguna et al., 2015]), its cytoprotective actions in this setting may be via activation of a different stress response pathway(s) (Fig. 3 F).

### Autophagy-dependent LMX1A and LMX1B protein turnover

The capacity of LMX1A and LMX1B to influence autophagy gene expression and to protect against rotenone poisoning in mDANs prompted us to determine how the stability of these proteins is regulated. We measured LMX1A-FLAG and LMX1B-FLAG levels in stable HEK293T cells treated with cycloheximide (CHX; 16 h) and found that degradation was blocked by BafA1 and by MG132 (proteasome inhibitor; Fig. 4, A and B), suggesting involvement of both autolysosomal and proteasomal degradation pathways. Focusing on LMX1B, which has broader tissue expression and function, we determined that turnover was accelerated during nutrient starvation and was again dependent on the proteasomal and autolysosomal pathways (Fig. 4 C). As the steady-state distribution of key transcription factors such as TFEB, ZSCAN3, and FOXO changes in response to stress (Fullgrabe et al., 2016), we used cell fractionation and imaging to monitor the localization of LMX1B during nutrient withdrawal (Fig. 4, D–F). Immunoblotting confirmed the expected enrichment of LMX1B in the nucleus in fed conditions and revealed a further shift from cytosol to nucleus during acute (2 h) starvation (Fig. 4 D). Imaging of LMX1B-FLAG in HEK293T cells confirmed its strong nuclear enrichment in full nutrients and during early starvation (2 h; Fig. 4 E) and revealed the presence of discrete cytosolic LMX1B-FLAG foci in some cells (Fig. 4 E). Strikingly, following extended starvation periods (6 h), the nuclear LMX1B-FLAG fluorescence signal appeared to diminish, concomitant with the accumulation of abundant cytosolic LMX1B puncta that strongly co-localized with GFP-LC3B positive autophagosomes (Fig. 4 E). To begin to understand the control of LMX1B distribution in living cells, we expressed DOX-inducible TRE-GFP-LMX1B in HeLa cells and used FLIP (fluorescence loss in photobleaching) to measure nuclear GFP fluorescence loss during simultaneous photobleaching of a cytoplasmic region of interest as a readout of LMX1B cycling between these compartments (Fig. 4 F). The rate of nuclear GFP-LMX1B fluorescence signal loss was significantly lower in nutrient starved cells (Fig. 4 F), consistent with the observed net nuclear LMX1B accumulation during early starvation (Fig. 4 D). Thus, LMX1B nucleus-to-cytoplasm shuttling is influenced by nutrient availability in a biphasic manner: initially, during early starvation, LMX1B further accumulates in the nucleus where it can enhance autophagy-related gene transcription; later, during extended starvation, nuclear LMX1B levels reduce, while cytosolic LC3B-positive LMX1B foci become more abundant and more prominent, and LMX1B is

progressively degraded by autophagy and the proteasome. A detailed mechanistic analysis of the relationships between LMX1B and LC3B (and other ATG8 proteins) follows below. As an initial test of whether ATG8 proteins influence LMX1B nucleocytoplasmic shuttling, we compared rates of nuclear GFP-LMX1B FLIP in wild-type and ATG8 KO HeLa cells (Nguyen et al., 2016). Although average GFP-LMX1B FLIP rate was markedly higher in starved ATG8-null HeLa cells, this was not significantly different from control cells (Fig. 4 F).

### LMX1B binds multiple ATG8 proteins via a conserved region containing a LIR-like motif

The co-localization observed between cytoplasmic LMX1B and LC3B in starved HEK293T cells (Fig. 4 E) suggested that these proteins might interact. Correspondingly, we identified a possible LIR (LC3-interacting region)-like motif C-terminal to the homeodomain of LMX1B ($^{309}$YTPL$^{312}$; https://ilir.warwick.ac.uk; Fig. 5 A). Alignment against other published LIR motifs suggested that this might conform to the tyrosine-type group identified in LIR domain containing proteins including NBR1, ATG4B, and FUNDC1 (Fig. 5 B). LIR motifs are present in proteins that interact with ATG8 family members to facilitate their autophagy-mediated turnover, in receptors for selective autophagy, and in a subset of key components of the autophagosome assembly machinery (Johansen and Lamark, 2020; Lamark et al., 2017). We therefore tested whether LMX1B interacted with ATG8 family members. GFP-TRAP immunoprecipitation (IP) of lysates of HEK293T cells co-transfected with human GFP-ATG8 and LMX1B-FLAG revealed that LMX1B-FLAG was enriched in all GFP-ATG8 fractions (Fig. 5 C). Furthermore, GFP-LC3B was detected in anti-FLAG immune complexes from lysates of HEK293T cells stably expressing LMX1B-FLAG (Fig. 5 D). Importantly, we observed coIP of endogenous LC3B using antibodies against native LMX1B in HEK293T cells (Fig. 5 E). Consistent with the involvement of LIR-type interactions, LIR docking site mutant (LDS) GFP-LC3B (F52A/L53A; Ichimura et al., 2008) co-precipitated very weakly with LMX1B-FLAG in HEK293T cells (Fig. 5 F). Finally, LMX1B-FLAG also bound strongly to immobilized recombinant ATG8 proteins in vitro, with a possible preference for GABARAPs in this context (Fig. 5 G).

Using Chimera (Pettersen et al., 2004), we visualized the LC3B crystal structure bound to the FYCO1 LIR peptide (5d94.pdb; Olsvik et al., 2015), and used this structure as a template to guide the positioning of a peptide incorporating the proposed LMX1B LIR. Throughout four repeat molecular dynamics simulations of the resulting assemblies, the LMX1B peptide remained in contact with LC3B, with Y309 and L312 anchors docked within LC3B hydrophobic pockets HP1 and HP2, respectively, in a parallel β-strand configuration (Fig. S3 A and Video 1, and Fig. S3 B). The proposed LMX1B LIR appears in an exposed region of low/very low confidence in the current available Alphafold model (DeepMind/EMBL-EMI; Fig. S3 C), consistent with it being located on a flexible/exposed protein loop. Recent data have highlighted the importance of residues outside of core LIRs in stabilizing interactions with ATG8 proteins (Johansen and Lamark, 2020; Wirth et al., 2019). In the case

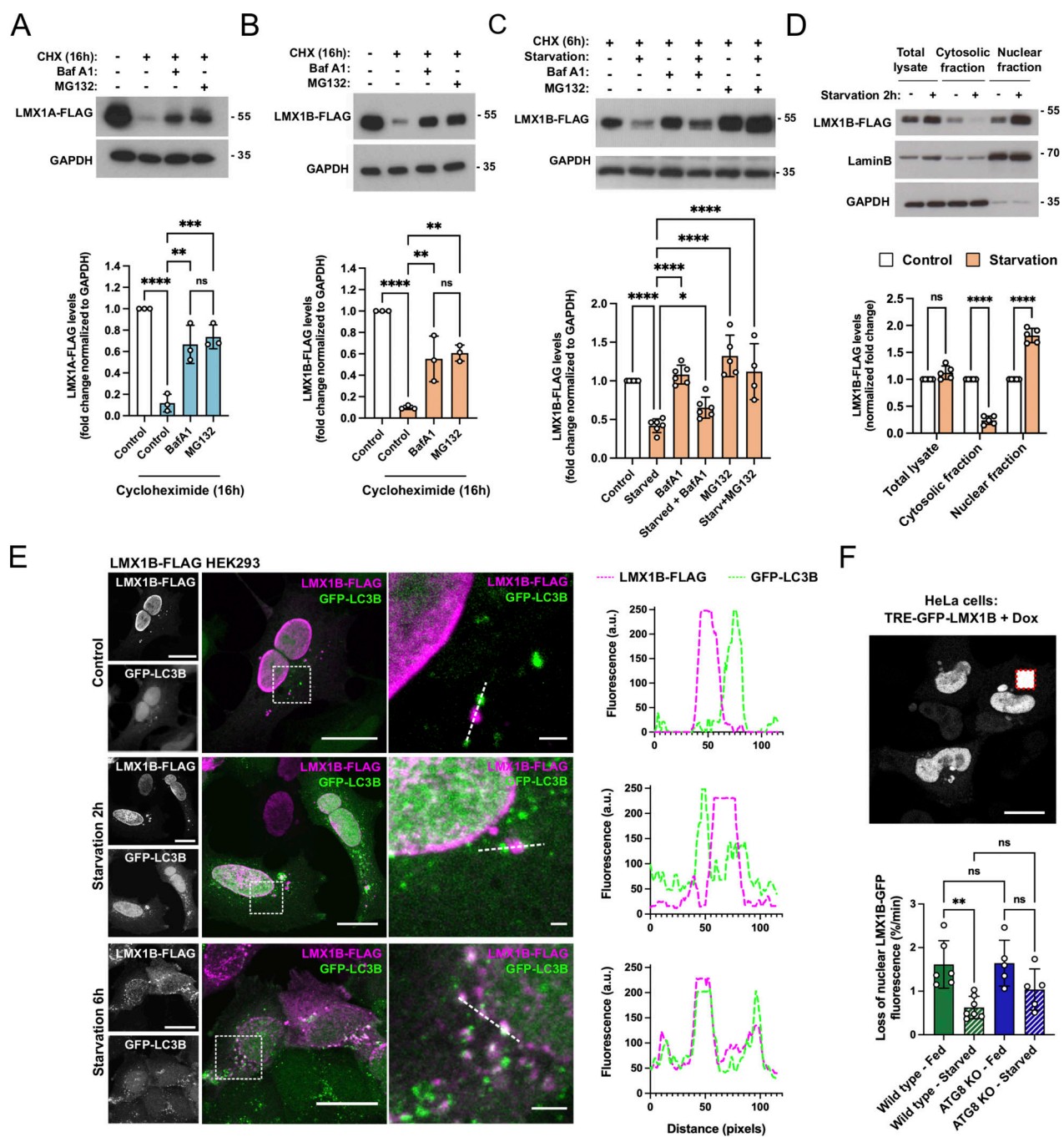

Figure 4. **LMX1B colocalizes with LC3B and is degraded during nutrient starvation. (A)** LMX1A-FLAG turnover in stable HEK293T cells treated with CHX (50 µg/ml) for 16 h in the absence or presence of BafA1 (20 nM) or MG132 (10 µM; *n* = 3). Molecular weight markers are shown in kD. Mean ± SD; one-way ANOVA followed by Tukey's multiple comparison post-hoc test: **P < 0.01, ***P < 0.001, ****P < 0.0001. **(B)** LMX1B-FLAG turnover in stable HEK293T cells treated with CHX (50 µg/ml) for 16 h in the absence or presence of BafA1 (20 nM) or MG132 (10 µM) (n = 3). Molecular weight markers are shown in kD. Mean ± SD; one-way ANOVA followed by Tukey's multiple comparison post-hoc test: **P < 0.01, ****P < 0.0001. **(C)** LMX1B-FLAG degradation in stable HEK293T cells during starvation (6 h) in the presence of CHX (50 µg/ml) ± BafA1 (20 nM) or CHX ± MG132 (10 µM; *n* = 4–6). Molecular weight markers are shown in kD. Mean ± SD; one-way ANOVA followed by Fisher's LSD test for planned comparisons: *P < 0.05, ****P < 0.0001. **(D)** Immunoblot and densitometry of cytosolic and nuclear fractions from HEK293T cells stably expressing LMX1B-FLAG with or without 2 h nutrient starvation. Molecular weight markers are shown in kD. Lamin B was used to normalize the nuclear fractions and GAPDH for the total and cytosolic fractions (*n* = 5). Mean ± SD; Student's *t* test: ****P < 0.0001. **(E)** LMX1B relocates to cytosolic puncta that co-label with LC3B during extended nutrient starvation. Representative immunofluorescence images (to the left) and linescans of the indicated areas of LMX1B-FLAG stable HEK293T cells transfected with GFP-LC3B (green), fixed, and stained with anti-FLAG antiserum (magenta) after 2 or 6 h starvation. Bar = 20 µm, full field; 2 µm, zoom inset. **(F)** FLIP analysis of nucleus to cytoplasmic GFP-LMX1B transport in fed and starved cells, in the absence or presence of human ATG8 proteins (expression induced with DOX: 500 ng/ml; 20 h). Wild-type and ATG8 KO HeLa cells (Nguyen et al., 2016) were maintained in normal growth medium or starved for 2 h before live-cell FLIP analysis for a period of 15 min constant photobleaching of a cytoplasmic region of interest (example image shown top, right). Data are a measure of the loss of GFP-LMX1B fluorescence intensity within the nucleus of the photobleached cell normalized against non-photobleached cells in the same field of view. Bar = 10 µm. Data points represent individual treated cells in multiple fields of view (*n* = 4–6). Mean ± SD; one-way ANOVA followed by Tukey's multiple comparison post-hoc test: **P < 0.01.

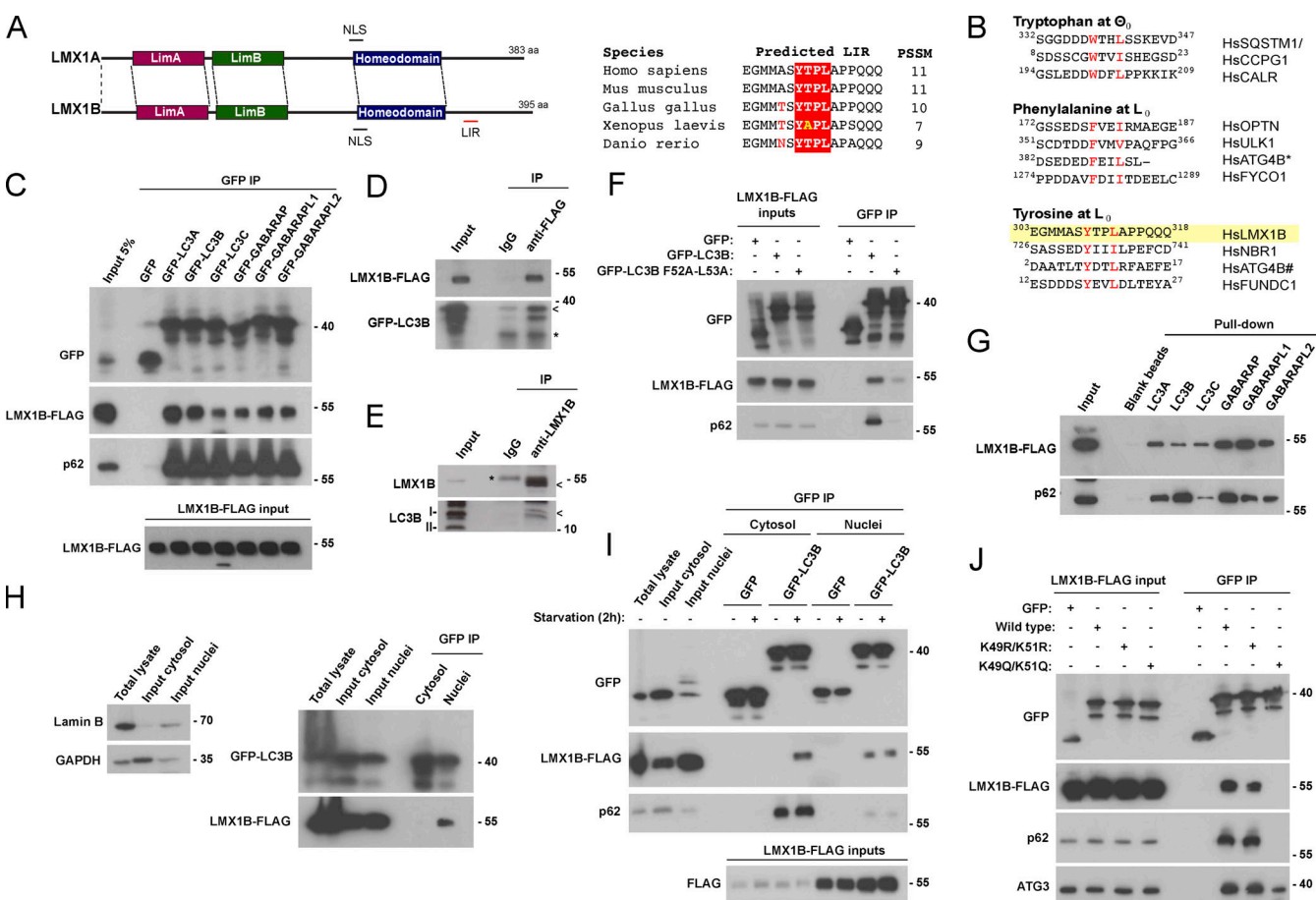

Figure 5. **LMX1B interacts with ATG8s in a compartment and context-dependent manner. (A)** Schematic of human LMX1A and LMX1B functional domains showing predicted NLS and a possible LIR motif in LMX1B (left; the same schematic is shown in Fig. S4 C depicting the location of a possible LIR in LMX1A). Alignment of the proposed LMX1B LIR in different species (right). **(B)** Sequence alignments of LIR motifs in various human (Hs) proteins designated as tryptophan-, phenylalanine- and tyrosine-type LIRs (refers to the residue at $P_0$ of the LIR). * C-terminal ATG4B LIR; # N-terminal ATG4B LIR. **(C)** GFP-TRAP co-precipitation of LMX1B-FLAG with GFP-ATG8 family members in HEK293T cells. Immunoblotting for p62 is included as a positive control for binding. 5% protein lysate from equivalent GFP-expressing cells is shown as "input." **(D)** Anti-FLAG co-precipitation of GFP-LC3B from lysates of HEK293T cells stably expressing LMX1B-FLAG. A non-specific IgG control is included. Arrow indicates position of GFP-LC3B; * indicates the position of the antibody light chain. **(E)** CoIP of endogenous LMX1B and LC3B using antibodies against native LMX1B in HEK293T cells. Arrow indicates position of LMX1B and LC3B in the anti-LMX1B lane; * indicates position of antibody heavy chain. **(F)** GFP-TRAP co-precipitation of LMX1B-FLAG with wild-type or LIR docking mutant (F52A/L53A) GFP-LC3B in HEK293T cells. Immunoblotting for p62 is included as a positive control for binding. 5% of protein lysate was used as control for protein expression (input). **(G)** In vitro pull-down of LMX1B-FLAG from lysates of HEK293T stably expressing LMX1B-FLAG using sepharose beads covalently attached to recombinant His-ATG8 family members. Immunoblotting for p62 is included as a positive control for binding. **(H)** GFP-TRAP IP of nuclear and cytosolic fractions from lysates of HEK293T cells co-expressing GFP-LC3B and LMX1B-FLAG. The LMX1B–LC3B interaction occurs exclusively in the nucleus under basal conditions. Fractionation is demonstrated to the left using Lamin B and GAPDH as markers for nuclei and cytosol, respectively. **(I)** GFP-TRAP IP of nuclear and cytosolic fractions from lysates of HEK293T cells co-expressing GFP-LC3B and LMX1B-FLAG. Comparisons of pull-downs in full nutrients or following 2 h starvation. An LMX1B–LC3B interaction emerges in the cytosol during starvation. 5% of protein lysate from equivalent GFP-expressing cells is shown as a representative of "input." Immunoblotting for p62 is included as a positive control for binding. **(J)** GFP-TRAP IP of LMX1B-FLAG in HEK293T cells co-expressing wild-type, acetylation-deficient (K49R/K51R), and acetylation mimic (K49Q/K51Q) GFP-LC3B. Immunoblotting for p62 and ATG3 is included as positive controls for binding. **(C–J)** Molecular weight markers are shown in kD. PSSM, Position Specific Scoring Matrix.

of LMX1B, differences were observed in the positioning of upstream residues relative to the FYCO1 crystallographic position, with the LMX1B peptide adopting a final pose that was displaced toward the α2 helix of the N-terminal arm of LC3B (Birgisdottir et al., 2013; Fig. S3 D). The FYCO1 LIR residues D1277 and D1281 form salt bridges between R10 and R70 of LC3B, respectively. By contrast, LMX1B lacks negatively charged residues in this region; rather A307 (LIR $X_{-2}$; Fig. 5 B) packs against L22 of LC3B. Downstream, the first glutamine in the LMX1B triple Q stretch

(LIR $X_7$; Fig. 5 B), folded back in the simulation to adopt a similar configuration to that reported for the glutamic acid at LIR $X_7$ in FYCO1 (Fig. S3 E), thus possibly stabilizing the interaction. Finally, we noted that our proposed LMX1B-LC3B model was very similar to the docking of the tyrosine-type LIR identified in the crystal lattice structure of the N-terminal region of human ATG4B-LC3B (Satoo et al., 2009), both within the core LIRs and in the near-identical manner in which the displaced upstream sequences engage with the LC3B α2 helix (Fig. S3 F).

## The LMX1B–LC3B interaction is location and context dependent

At steady state, a large and relatively immobile pool of LC3 resides in the nucleus where it engages with promyelocytic leukemia bodies and nucleolar components via LIR-type interactions, suggesting nuclear regulatory roles (He et al., 2014; Kraft et al., 2016). Notably, it has been demonstrated that cytosol-to-nuclear shuttling forms part of the normal LC3 itinerary, with only the nuclear released pool competent to undergo lipidation during autophagosome assembly (Huang et al., 2015). Strikingly, we found that the LMX1B–LC3B interaction is restricted to the nuclear compartment in LMX1B-FLAG HEK293T cells under full nutrient conditions (Fig. 5 H), and we detected endogenous LMX1B in GFP-LC3B pull-downs of nuclear extracts (Fig. S4 A). Indeed, LMX1B binding to all ATG8 family members occurred only in the nucleus in co-overexpression experiments, unlike p62 which bound ATG8 family members overwhelmingly in the cytoplasm (Fig. S4 B). To test whether nutrient availability influenced the location and apparent strength of LMX1B binding to LC3B, HEK293T cells co-expressing GFP-LC3B with LMX1B-FLAG were placed in starvation media for 2 h, then nuclear and cytoplasmic fractions were subjected to GFP-TRAP IP (Fig. 5 I). In fed and starved conditions, the nuclear LMX1B-FLAG interaction with GFP-LC3B remained consistent; however, a new, strong interaction between LMX1B-FLAG and GFP-LC3B was observed in the cytosol during starvation (Fig. 5 I). By contrast, cytosolic p62 binding to LC3B was not markedly influenced by starvation (Fig. 5 I).

The release of the nuclear LC3 pool during starvation is regulated by SIRT1-dependent deacetylation, with LC3 exiting the nucleus in complex with the diabetes- and obesity-regulated nuclear factor (DOR) in preparation for its incorporation into the nascent isolation membrane (Huang et al., 2015; Mauvezin et al., 2010). GFP-TRAP pull-downs in HEK293T cells expressing acetylation-resistant (K49R/K51R) and acetylation mimic (K49Q/K51Q) GFP-LC3B (Huang et al., 2015) revealed that LMX1B (Fig. 5 J) associates strongly with K49R/K51R LC3B, but not with K49Q/K51Q LC3B, suggesting that LMX1B binds to the non-acetylated form of LC3B. The same was true for LIR-dependent LC3B binding to p62 (Fig. 5 J). By contrast, non-LIR dependent binding to ATG3 (e.g., Nakatogawa, 2013) was retained—albeit noticeably reduced—in the K49Q/K51Q LC3B mutant (Fig. 5 J). Together, these data suggest that LMX1B binding to LC3B is regulated by acetylation/deacetylation, and that parallel LMX1B and LC3B nuclear shuttling itineraries exist. Analysis of the conditions for interaction between LMX1B and LC3B reveals novel features and regulation that are indicative of independent roles with respect to intracellular localization, nutrient status, and acetylation, with strong binding in the nucleus being a constitutive feature. We also identified a possible LIR-type motif in human LMX1A ($^{290}$YTAL$^{293}$), and LMX1A co-precipitated with GFP-ATG8s in HEK293T cells, dependent on an intact LDS in LC3B (Fig. S4, C–H). Unlike LMX1B, the steady-state LMX1A interaction with GFP-LC3B in fed cells occurred primarily in the cytosol (Fig. S4 F), although this should be interpreted with some caution since LMX1A is not normally expressed in HEK293T cells; however, in common with LMX1B,

this cytosolic interaction was strengthened during nutrient starvation (Fig. S4 G). Finally, the LMX1A-LC3B interaction was also found to be sensitive to acetylation (Fig. S4 H).

To understand the significance of the LMX1B–ATG8 interaction, we first mutated the key tyrosine and leucine residues within the proposed LMX1B LIR (Y309A/L312A). In GFP-TRAP pull-downs in HEK293T cells, this reduced binding to ATG8 family members LC3B and GABARAP-L2 (Fig. S5 A); however, it had negligible impact on binding to LC3A, LC3C, GABARAP, and GABARAP-L1 (Fig. S5 A), possibly due to stabilizing interactions outside of the core LIR (Johansen and Lamark, 2020; Wirth et al., 2019), or the presence of an alternative LIR(s) (similarly, alanine mutagenesis of the corresponding, putative LMX1A LIR [Y290A/L293A] did not alter its binding to LC3B; Fig. S4 I). It also did not alter its nuclear localization (Fig. S5 B) or turnover (Fig. S5, C and D). To generate an LMX1B mutant that failed to bind to a broader range of human ATG8 proteins, we targeted the region downstream of the core LIR by generating a Δ308-317 deletion construct, so selected based on our in silico modeling (Fig. S3) and because amino acids C-terminal to the core LIR have been highlighted as playing important stabilizing roles in other ATG8 interactors (Johansen and Lamark, 2020; Wirth et al., 2019). In GFP-TRAP pull-downs, Δ308-317 LMX1B showed substantially reduced binding to all ATG8 family members (Fig. 6 A), but this construct was targeted normally to the nucleus (Fig. 6 B). Despite reduced ATG8 binding, lysosomal turnover of Δ308-317 LMX1B occurred at comparable rates to wild-type in HEK293T cells (Fig. 6, C and D). These data suggest that LMX1B binding to ATG8s involves a non-canonical LIR-type interaction involving the LDS. Further, our results indicate that this interaction may not be required for autophagy mediated LMX1B degradation.

## ATG8 binding stimulates LMX1B-based transcription

The constitutive nuclear association of wild-type LMX1B with LC3B and other ATG8 proteins in nutrient replete conditions prompted us to test whether this novel interaction influences LMX1B transcription factor activity. We first measured expression of luciferase reporter constructs containing tandem FLAT sequences from the promoters of selected LMX1B target genes; namely, NURR1, INS (pro-insulin), TFEB, ULK1, UVRAG, and PINK1 (Fig. 6 E). All constructs tested generated robust luciferase expression responses driven by wild-type LMX1B, except for the chosen TFEB FLAT where luciferase expression was not significantly increased in this assay (Fig. 6 E). Crucially, luciferase expression was significantly lower when driven by ATG8 binding-deficient Δ308-317 LMX1B (Fig. 6 E). By contrast, and in keeping with its failure to block binding to most ATG8 proteins, the Y290A/L293A LMX1B mutation did not negatively impact on luciferase expression driven by the same FLAT element constructs (Fig. S5 E). In an orthogonal approach to establish the tenet of ATG8 proteins acting as LMX1B cofactors, we measured LMX1B-dependent FLAT sequence luciferase reporter expression in HEK293T cells siRNA depleted for LC3B, GABARAP-L1, or LC3A (Fig. 6, F and G; and Fig. S6). In siLC3B-treated cells expressing wild-type LMX1B, luciferase expression via NURR1, pro-insulin, TFEB, UVRAG, and PINK1 FLAT

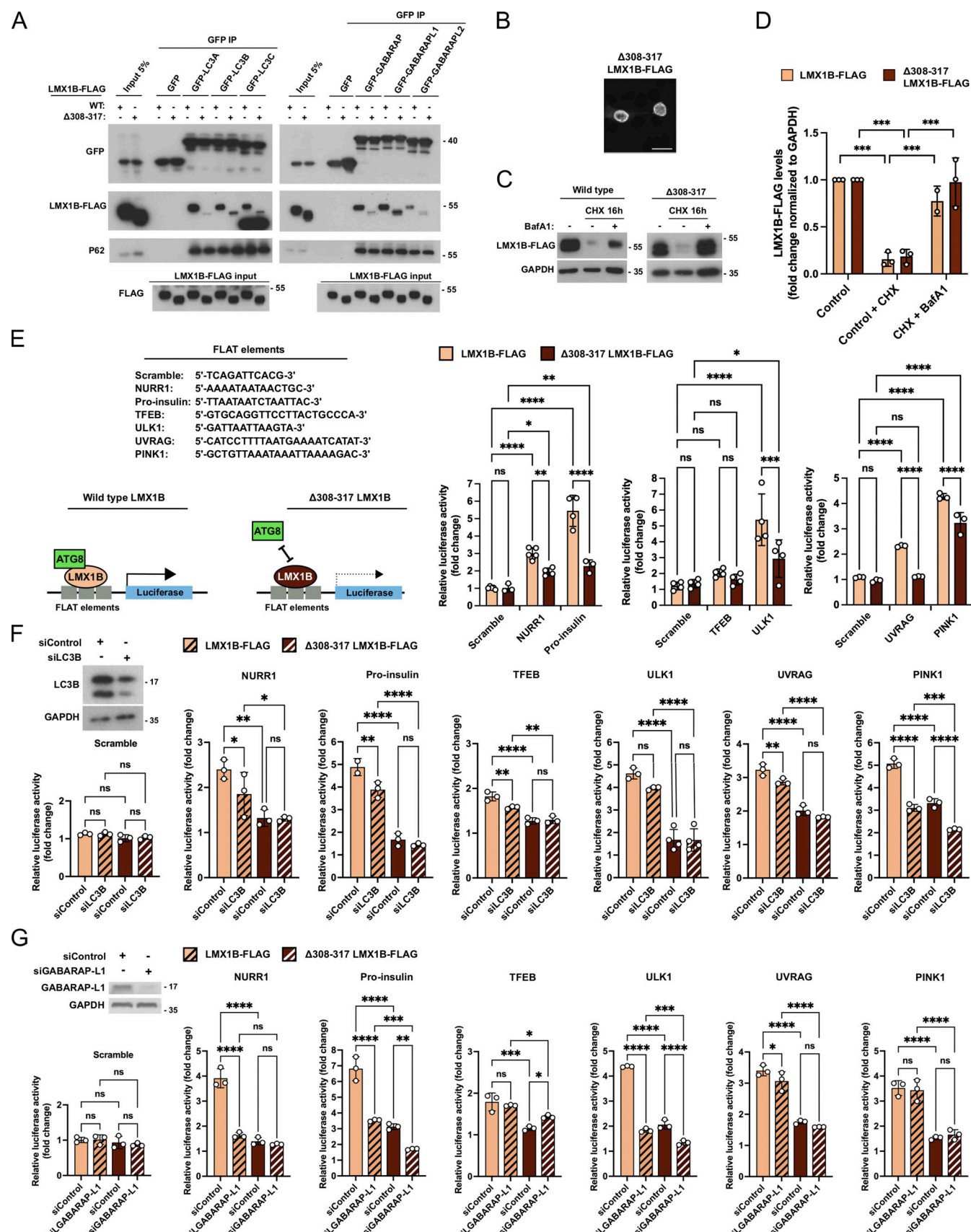

Figure 6. **LIR-dependent binding to ATG8s stimulates LMX1B-driven target gene transcription. (A)** GFP-trap IP of wild-type and LIR mutant (Δ308-317) LMX1B-FLAG in lysates of HEK293T co-expressing GFP-ATG8 family members. 5% of protein lysate from equivalent GFP-expressing cells is shown as a

representative of "input." Immunoblotting for p62 is included as a positive control for binding. Molecular weight markers are shown in kD. **(B)** Immunofluorescence staining of ΔLIR mutant (Δ308-317) LMX1B-FLAG expressed in HEK293T cells. Bar = 20 μm. **(C and D)** LMX1B turnover in HEK293T stably expressing wild-type or Δ308-317 LMX1B-FLAG. Cells were treated with CHX (50 μg/ml; 16 h) in the absence or presence BafA1 (20 nM). Representative blot (C) and quantitation relative to GAPDH (D). Molecular weight markers are shown in kD. Mean ± SD (n = 3); one-way ANOVA followed by Tukey's multiple comparison post-hoc test: ***P < 0.001. **(E)** LIR mutant Δ308-317 LMX1B fails to stimulate FLAT-luciferase activity. To the left, luciferase FLAT element sequences (top), and schematic of the proposed role of ATG8 proteins as LMX1B co-factors (bottom). To the right, quantitation of expression driven by wild-type and Δ308-317 LMX1B in HEK293T cells. Levels were normalized to empty vector control (pcDNA 3.1). Mean ± SD (n = 3–4); one-way ANOVA followed by Tukey's multiple comparison post-hoc test comparing scrambled sequence against NURR1, pro-insulin, TFEB, ULK1, UVRAG, and PINK1: *P < 0.05, **P < 0.01, ***P < 0.001, ****P < 0.0001. **(F)** LC3B siRNA silencing dampens LMX1B-mediated luciferase reporter expression. Example LC3B immunoblot (left, top) and scramble, NURR1, pro-insulin, ULK1, UVRAG, and PINK1 FLAT promoter luciferase assay (LightSwitch) driven by wild-type or Δ308-317 LMX1B-FLAG Levels are presented relative to empty vector control (pcDNA 3.1). Mean ± SD (n = 3–4); one-way ANOVA followed by Fisher's LSD test for planned comparisons: *P < 0.05, **P < 0.01, ***P < 0.001, ****P < 0.0001. **(G)** GABARAP-L1 siRNA silencing dampens LMX1B-mediated luciferase reporter expression. Example GABARAP-L1 immunoblot (left, top) and scramble, NURR1, pro-insulin, ULK1, UVRAG, and PINK1 FLAT promoter luciferase assay (LightSwitch) driven by wild-type or Δ308-317 LMX1B-FLAG. Levels are presented relative to empty vector control (pcDNA 3.1). Mean ± SD (n = 3); one-way ANOVA followed by Fisher's LSD test for planned comparisons: *P < 0.05, **P < 0.01, ***P < 0.001, ****P < 0.0001.

elements was significantly reduced (ULK1 luciferase expression was also markedly lower, but this was not statistically significant; Fig. 6 F). siRNA depletion of GABARAP-L1 significantly reduced expression of all tested FLAT elements by wild-type LMX1B, except for TFEB and PINK1 (Fig. 6 G). By contrast, siRNA suppression of LC3A did not alter luciferase expression of NURR1, pro-insulin, or ULK1 (Fig. S6 B; other genes not tested), suggesting that a restricted sub-group of ATG8s influences LMX1B transcriptional control. Notably, FLAT-luciferase readouts were significantly dampened when driven by Δ308-317 LMX1B under all conditions (Fig. 6, F and G; and Fig. S6 B), and were further reduced in LC3B (PINK1) and GABARAP-L1 (pro-insulin, TFEB, ULK1) suppressed cells (Fig. 6, F and G). Together, these data support a model in which non-canonical LIR-type LMX1B binding to selected ATG8 family members stimulates LMX1B-mediated transcription, thus revealing a novel role for ATG8s as transcription factor co-factors with broad implications for maintenance and stress resilience of important cell types.

### Non-canonical LIR-type ATG8 binding boosts LMX1B target gene expression for a robust autophagy response and cell stress resilience

To test the dependency of ATG8 binding to LMX1B in the context of autophagy and cellular stress responses, we first performed siRNA/shRNA knockdown/rescue experiments in HEK293T cells (Fig. 7 A) and human iPSC-derived mDANs (Fig. 7 B) using wild-type and Δ308-317 LMX1B rescue constructs. qRT-PCR demonstrated that wild-type LMX1B rescued expression of several autophagy and associated target genes in both settings, but the LIR deficient Δ308-317 LMX1B construct could not (Fig. 7, A and B). By contrast, and consistent with its sustained binding to most ATG8s and ability to drive expression of the FLAT luciferase constructs (Fig. S5, A and E), the Y309A/L312A LMX1B mutant efficiently rescued target gene expression (Fig. S5, F and G). These findings reinforce the proposed mechanistic relationship between ATG8 proteins and LMX1B in the context of target gene transcription.

We next rescued LMX1B KO HEK293T cells (clone KO2) with lentiviruses expressing wild-type or Δ308-317 LMX1B and measured autophagy flux by immunostaining for LC3B and WIPI2 puncta (Fig. 7 C). Expression of wild-type LMX1B significantly increased LC3B puncta in fed LMX1B KO cells treated with BafA1,

and in starved KO cells in the absence and presence of BafA1 (Fig. 7 C). Increased WIPI2 puncta were also recorded in the starved + BafA1 condition in wild-type LMX1B expressing cells (Fig. 7 C). Expression of the Δ308-317 LMX1B construct did restore LC3B puncta levels in BafA1-treated cells and in starved cells, but the rescue level in starved cells in the presence of BafA1 was significantly weaker than in cells expressing wild-type LMX1B (LC3B and WIPI2 puncta readouts; Fig. 7 C).

In addition to their muted autophagy responses (Fig. 1 I), LMX1B KO HEK293T cells were more susceptible to rotenone poisoning (Fig. 7 D). To test the influence of ATG8 binding and enhanced LMX1B-based transcriptional control on cellular stress resistance, we rescued LMX1B expression in LMX1B KO HEK293T cells using lentiviruses expressing wild-type or Δ308-317 LMX1B, then subjected them to rotenone treatment (Fig. 7 E). For both LMX1B CRISPR clones, wild-type LMX1B provided clear protection over the empty vector control, but Δ308-317 LMX1B failed to provide any level of protection (Fig. 7 E). Finally, we induced expression of wild-type and mutants LMX1B constructs (Y309A/L312A; Δ308-317) in human mDANs, and measured cell death (caspase activity) following treatment with rotenone (24 h). As seen before (Fig. 3 F), expression of wild-type LMX1B strongly protected against rotenone, as indeed did expression of the Y309A/L312A mutant; however, protection was lost in cells expressing Δ308-317 LMX1B (Fig. 7 F). These data support our model of LDS-dependent association with ATG8 family members enhancing the safeguarding role of LMX1B in human mDANs subjected to energetic stress. Taken together, our findings reveal an intriguing new regulatory interface between the core autophagy machinery and the LMX1A/LMX1B transcription factors that is important for robust expression of autophagy genes and other regulatory genes involved in cell specification, differentiation, and stress resilience. Our studies focusing mainly on LMX1B demonstrate an underlying layer of regulatory control exemplified by compartmentalized and nutrient status dependent ATG8 binding, with enhanced autolysosomal turnover during prolonged nutrient stress.

## Discussion

Autophagy is a central facet of cell and tissue homeostasis throughout the lifespan that protects against many important

Figure 7.   **LIR-dependent LMX1B binding to ATG8 proteins provides cellular stress protection. (A)** LIR mutant Δ308-317 LMX1B fails to rescue autophagy gene transcription in HEK293T cells. HEK293T cells were co-transfected twice with 50 nM smartpool LMX1B siRNA and codon optimized, siRNA-resistant wild-type, or Δ308-317 LMX1B (or empty vector control). mRNA levels were normalized to siControl. Mean ± SEM ($n$ = 3); one-way ANOVA followed by Dunnett's multiple comparison post-hoc test comparing the two LMX1B constructs to empty vector (*$P$ < 0.05) and a Student's $t$ test to compare Δ308-317 and wild-type LMX1B (#$P$ < 0.05). **(B)** LIR mutant Δ308-317 LMX1B fails to rescue autophagy gene transcription in iPSC-derived human mDANs. Mature (day ~50) mDANs were transduced with hsyn-GFP-U6-shLMX1B and LMX1B levels were rescued with codon optimized, shRNA-resistant wild-type, or Δ308-317 TRE- LMX1B (or empty vector as a control). mRNA levels were normalized to shControl. Mean ± SEM ($n$ = 3 wells from a single neuralization); one-way ANOVA followed by Dunnett's multiple comparison post-hoc test comparing the two LMX1B constructs to empty vector (*$P$ < 0.05) and a Student's $t$ test to compare Δ308-317 with wild-type LMX1B (#$P$ < 0.05). **(C)** LIR mutant Δ308-317 LMX1B fails to rescue the full autophagy response in LMX1B CRISPR KO2 HEK293T cells expressing wild-type or Δ308-317 LMX1B. To the left, example inverted single channel images and color overlays (LC3B is depicted in grayscale; WIPI2 is depicted in magenta; DAPI is depicted in blue). Cells were counterstained with DAPI (blue). Bar = 20 μm. To the right, puncta counts for GFP-LC3B and WIPI2 in nutrient replete conditions or following 2 h starvation in the presence or absence of BafA1 (20 nM, 2 h). Mean ± SD of 20–30 fields of cells from three independent experiments, counting ~50 cells per field; one-way ANOVA followed by Fisher's LSD test for planned comparisons: *$P$ < 0.05, ****$P$ < 0.0001. **(D)** Rotenone-induced cell death (10 μM, 20 μM; 48 h) in control and LMX1B CRISPR KO HEK293T cells. Data show means ± SD of three independent experiments; two-way ANOVA followed by Tukey's multiple comparison post hoc test: **$P$ < 0.01, ****$P$ < 0.0001 vs. control cells. **(E)** Wild-type LMX1B, but not LIR mutant Δ308-317 LMX1B, protects against rotenone toxicity in LMX1B CRISPR KO HEK293T cells. KO1 and KO2 cells were treated with lentiviruses expressing empty vector control, wild-type LMX1B, or LIR mutant Δ308-317 LMX1B, then exposed to rotenone (10 μM) for 48 h. Data show means ± SD of three independent experiments; one-way ANOVA followed by Bonferroni multiple comparison post-hoc test: *$P$ < 0.05, **$P$ < 0.01, ***$P$ < 0.001, ****$P$ < 0.0001. **(F)** Rotenone-induced cell death (15 μM; 24 h) assay in iPSC-derived mDAN cultures (day 30–50) transduced with TRE-empty (control), TRE-LMX1B, Y309A/L312A TRE-LMX1B, or Δ308-317 TRE-LMX1B (expression induced with DOX: 500 ng/ml; 3 d). Active caspase fluorescence relative to total protein levels was normalized to TRE-empty control (left graph) or TRE-LMX1B control (right graph) values. Mean ± SD ($n$ = 4–9); one-way ANOVA followed by Bonferroni multiple comparison post-hoc test: *$P$ < 0.05, **$P$ < 0.01, ***$P$ < 0.001, ****$P$ < 0.0001.

human diseases (see Levine and Kroemer, 2008; Thorburn, 2018). With improved understanding of autophagy control in diverse cells, tissues, and stress scenarios, the development of strategies to maintain and/or boost autophagy levels to prevent

the onset of diseases that are commonly associated with aging is anticipated (Rubinsztein et al., 2011). To reach this goal, the regulatory pathways that coordinate tissue-specific autophagy responses must first be identified and comprehensively

understood. In this context, the existence of tunable autophagy transcriptional pathways that act in distinct cell and tissue types represents an attractive potential target for autophagy control in disease.

An example of such a potential mechanism for targeted autophagy control centers on the LIM homeodomain transcription factors LMX1A and LMX1B, essential during the differentiation and maintenance of mDANs in the substantia nigra region of the ventral midbrain (Arenas et al., 2015; Doucet-Beaupre et al., 2016; Laguna et al., 2015; Nakatani et al., 2010). Here, in addition to controlling the expression of dopaminergic neuronal regulatory gene pathways, LMX1A and LMX1B help sustain efficient autophagy cytoplasmic quality control and mitochondrial function in mature mDANs (Doucet-Beaupre et al., 2016; Laguna et al., 2015). Stabilizing and/or stimulating LMX1A/LMX1B might therefore improve mDAN resilience in aging and PD (Doucet-Beaupre et al., 2016). We have demonstrated that human LMX1A and LMX1B are autophagy transcription factors that can protect against PD-associated neuronal stress in vitro. While several previous studies have highlighted the potential benefits of direct stimulation of autophagy by chemical/small molecules in cell lines and PD animal models (Harris and Rubinsztein, 2011; Malagelada et al., 2010; Renna et al., 2010; Sarkar et al., 2007; Tanji et al., 2015), our work is an important example of how autophagy upregulation through transcriptional control can have a positive impact in a disease-relevant context. This echoes the protection against α-synuclein toxicity afforded by TFEB overexpression in the rat midbrain in vivo (Decressac et al., 2013), with the added benefit of LMX1A/LMX1B acting in regulatory pathways that are strongly associated with the mDAN population (Doucet-Beaupre et al., 2015). Crucially, we provide the first evidence that LMX1A and LMX1B engage with ATG8 family members via LIR-like interactions involving the LDS and propose, for LMX1B at least, that ATG8s have cofactor functions to influence transcriptional activities in the nucleus.

To determine the significance of the LMX1A/LMX1B-ATG8 interaction, we first modeled LC3B binding to a LMX1B peptide incorporating the putative LIR in silico, templated on the LC3B-FYCO1 structure. With the clear reduction in binding to LDS mutant LC3B, the modeling data suggested a LIR-to-LDS type interaction (with differences predicted in the positioning of residues upstream of the core LMX1B LIR), so we next used mutagenesis to change the two key LIR-associated residues (the aromatic phenylalanine at $\Theta_0$ and the aliphatic leucine at $\Gamma_3$) to alanine (Y309A/L312A). This was sufficient to reduce binding to LC3B and GABARAP-L2, but it did not impact markedly on interactions with other ATG8. Functionally, the Y309A/L312A mutant LMX1B efficiently restored LMX1B target gene expression in knockdown/rescue experiments and drove strong LMX1B FLAT-promoter luciferase expression, while providing good protection in rotenone toxicity assays. Together with the observed presence of a destabilizing proline within the core LIR-type motif (see Alemu et al., 2012; Wirth et al., 2019), and the paucity of acidic residues upstream of the proposed LIR, these observations are indicative of non-canonical or LIR-like binding, such as those detailed in other example ATG8 interactions (e.g., Liu et al., 2018; Satoo et al., 2009). We have, therefore, chosen to use this terminology throughout when referring to LMX1B–ATG8 interactions.

Previous studies have highlighted the significance of residues C-terminal to the core LIR in mediating interactions with the GABARAP family (Wirth et al., 2019). We, therefore, generated an LMX1B deletion construct that disabled the putative core LIR along with several downstream residues (Δ308-317 LMX1B). Δ308-317 LMX1B bound poorly to all ATG8s, failed to rescue autophagy gene expression, was unable to stimulate FLAT-promoter luciferase reporter expression, and crucially, failed to protect against rotenone in HEK293T cells and iPSC-derived human mDANs. In parallel, using an orthogonal approach, we found that siRNA suppression of LC3B and GABARAP-LI (but not LC3A) blunted LMX1B FLAT-promoter reporter expression. Together, this constitutes the first evidence that ATG8s can act as transcriptional cofactors in mammalian cells. Interestingly, a further transcription factor that binds to ATG8 via the LDS (sequoia) has recently been identified in Drosophila, but differing from our findings, sequoia behaves as a transcriptional repressor, with its sequestration by Drosophila ATG8a indirectly derepressing autophagy gene transcription in nutrient replete conditions (Jacomin et al., 2020). Notably, human LC3 has been shown to cycle through the nucleus in a SIRT1 and DOR-dependent fashion (Huang et al., 2015), thereby increasing its likelihood of encountering transcription factors to modulate their behavior. A fraction of LC3 is also targeted to the nucleolus where it interacts with a range of ribosomal proteins (Kraft et al., 2016), including the ribosome receptor NUFIP1 (Shim et al., 2019). In addition, in response to oncogenic stimuli, nuclear LC3 mediates lamin B1 degradation (Dou et al., 2015). It is likely that further roles for ATG8 family members in the nucleus will be revealed, with the identification of additional transcription factors that are influenced by ATG8 binding being a key objective.

In conclusion, we have described the LIR-dependent interaction between the autophagy transcription factors LMX1A and LMX1B and proteins of the ATG8 family. We propose a novel regulatory axis in which ATG8 proteins act as cofactors for stress-responsive transcription factors such as these, thereby influencing autophagy gene expression when cells are challenged. Our data show that LMX1A and LMX1B are also degraded by autophagy and by the proteasome, suggesting a mechanism by which the duration and/or strength of the LMX1A/B regulated autophagy response can be controlled under chronic, sustained stress. This finding has implications for our broader understanding of the control of autophagy in disease-susceptible tissues throughout the life course.

## Materials and methods
### Antibodies, reagents, and kits
The following primary antibodies were used (key: IB, immunoblot concentration; IF, immunofluorescence concentration): rabbit anti-ATG16L1 (PM040: IB, 1:1,000; IF, 1:400; MBL); mouse anti-ATG3 (M1333: IB, 1:1,000; MBL); rabbit anti-ATG7 (2631: IB, 1:1,000; Cell Signaling); mouse anti-FLAG (A00187: IB, 1:1,500; IF, 1:300; GenScript); rabbit anti-GAPDH (G8796: IB, 1:2,000;

Sigma-Aldrich); mouse anti-GFP (MMS-118R: IB, 1:2,000; Covance); goat anti-Lamin B (sc-6216: IB, 1:500; Santa Cruz); rabbit anti-LC3A/B (L8918: IB, 1:1,000; Sigma-Aldrich); rabbit anti-LC3B (L7543: IF, 1:400; IP; Sigma-Aldrich); rabbit anti-LMX1A (AB10533: IB, 1:1,000; IF, 1:1,000; IP; Millipore); rabbit anti-LMX1A (51-606: IP; ProSci); rabbit anti-LMX1B (18278: IB, 1:1,000; IF, 1:250; IP; Proteintech); rabbit anti-LMX1B (Witzgall lab: IB, 1:2,000; IF, 1:350); mouse anti-p62 (H0008878: IB, 1:1,000; IF, 1:400; Abnova); rabbit anti-TH (AB152: IB, 1:1,000; IF, 1:300; Millipore); mouse anti-TH (sc-25269: IB, 1:1,000; IF, 1:100; Santa Cruz); mouse anti-TUJ1 (801201: IB, 1:1,000; IF, 1:1,000; BioLegend); mouse anti-WIPI2 (MCA5780GA: IF, 1:400; BioRad). Secondary antibodies for immunoblotting: chicken anti-mouse HRP (G32-62DC-SGC: 1:10,000; Stratech); goat anti-rabbit HRP (G33-62G-SGC: 1:10,000; Stratech); chicken anti-goat HRP (G34-62DC-SGC; Stratech). Secondary antibodies for immunofluorescence: goat anti-mouse Alexa Fluor 488 (A-11029: 1:400; Invitrogen); goat anti-rabbit Alexa Fluor 488 (A-11034: 1:400; Invitrogen) goat anti-mouse Alexa Fluor 568 (A-11031: 1:400; Invitrogen); goat anti-rabbit Alexa Fluor 568 (A-11036: 1:400; Invitrogen); goat anti-mouse Alexa Fluor 647 (A-21236: 1:400; Invitrogen); goat anti-rabbit Alexa Fluor 647 (A-21244: 1:400; Invitrogen).

The following reagents were used in cell-based assays: MitoSOX (M36008; Thermo Fisher Scientific); carbonyl cyanide chlorophenylhydrazone (C2759; Sigma-Aldrich); BafA1 (Enzo, BML-CM110); MG132 (M7449; Sigma-Aldrich); CHX (TOKU-E, C084); DOX (Clontech, 631311); rotenone (R8875; Sigma-Aldrich). The following kits were used: EnzChek caspase-3 assay kit (Z-DEVD-AMC; E13183; Thermo Fisher Scientific); high-capacity RNA-to-cDNA kit (4387406; Thermo Fisher Scientific); LightSwitch luciferase assay kit (LS010; SwitchGear Genomics); RNA isolation kit (74104; Qiagen); Seahorse Mito Stress Test (103010-100; Agilent).

### Culturing immortalized cells
HEK293T human embryonic kidney cell-lines (ATCC), GFP-LC3B stable HEK293 cells (a gift from Dr. Sharon Tooze, Francis Crick Institute, London, UK), and HeLa cells (wild-type and ATG8 null [Nguyen et al., 2016]) were maintained in high-glucose DMEM medium supplemented with 10% FBS (Sigma-Aldrich) at 37°C in 5% $CO_2$. Transient transfections were performed with Lipofectamine 2000 Reagent (Thermo Fisher Scientific). For luciferase experiments, cells were cultured in DMEM phenol-red free medium supplemented with 10% FBS (Thermo Fisher Scientific). For starvation experiments, the following medium was used: 140 mM NaCl, 1 mM $CaCl_2$, 1 mM $MgCl_2$, 5 mM glucose and 20 mM Hepes supplemented with 1% BSA (Sigma-Aldrich; Axe et al., 2008).

### iPSC culture
We used the wild-type α-synuclein 2 (NAS2) iPSC line derived from human dermal fibroblasts by retroviral reprogramming at passages 40–80 (kindly provided by Dr. Tilo Kunath, Center of Regenerative Medicine, University of Edinburgh, Edinburgh, UK; Devine et al., 2011). NAS2 were maintained in Essential 8 TM Medium (E8) supplemented with Essential 8 Supplement (Thermo Fisher Scientific) and RevitaCell (1/100; Thermo Fisher

Scientific) in plates previously coated with 5 µg/ml Vitronectin (in PBS; Thermo Fisher Scientific) at 37°C in 5% $CO_2$ as fully described in Stathakos et al. (2019). In brief, for routine passaging, media was removed, and cells were washed once for 10 s with EDTA (0.5 mM), then incubated with EDTA for 5 min at room temperature (1 ml/well of a 6-well plate) to dissociate cells. EDTA was removed and cells were washed rapidly with E8 complete media to neutralize and remove EDTA. Cells were collected in 2 ml E8 complete media with RevitaCell or Y-27632 (10 nM) included during the first few passages after thawing to increase cell survival, with the concentration of either being reduced (1/2) until the third passage when the supplement was omitted. Then, cells were plated into vitronectin-coated plates (5 µg/ml, 1 h) at a ratio of 1:5 (corresponding to 40,000–50,000 cells/cm²) for routine maintenance, or 1:6–1:8 for neuralization.

### mDAN differentiation and maintenance
We used an improved monolayer protocol for mDAN differentiation based on previously published protocols (Nistor et al., 2015; Torper et al., 2013), and described in Stathakos et al. (2019). mDAN differentiation was achieved by the addition of SMAD inhibitors and the pattering factors, WNT and SHH (Arenas et al., 2015). iPSCs in small colonies (commonly 3 d after initial plating) were grown in N2B27 neural differentiation media for 9 d, comprising 50% neurobasal media (Thermo Fisher Scientific) and 50% DMEM/F-12 with Glutamax (Thermo Fisher Scientific), supplemented with: N2 (1/200; Thermo Fisher Scientific); B27 (1/100; Thermo Fisher Scientific); 1 mM Glutamax (Thermo Fisher Scientific); 5 mg/ml insulin (Sigma-Aldrich); nonessential amino acids (1/100; Thermo Fisher Scientific); penicillin/streptomycin (1/100; Sigma-Aldrich); 75 mM 2-mercaptoethanol (Thermo Fisher Scientific); neural induction factors, LDN (100 nM; Sigma-Aldrich) and SB431542 (10 µM; Tocris); and patterning factors, SHH (200 ng/ml; R&D Systems) and CHIR (WNT homolog; 0.8 µM; AxonMedchem). Cells were passaged using StemPro Accutase (Thermo Fisher Scientific), and typically plated at 1:2–1:3 dilution ratio in neural differentiation media supplemented with RevitaCell on plates coated with poly-L-ornithine (Sigma-Aldrich) and laminin (1/1,000 in PBS, Sigma-Aldrich). At day 9 of neuralization, induction and patterning factors were removed. After the next passage (usually at day 11), the N2B27 media was supplemented with the neurotrophic factors: 20 ng/ml brain-derived neurotrophic factor (Peprotech), 20 ng/ml glial cell-line-derived neurotrophic factor (Peprotech), and 0.2 mM ascorbic acid (Sigma-Aldrich). For terminal differentiation/maturation, cells were passaged on poly-L-ornithine/laminin coated coverslips or plates and cultured in complete N2B27 supplemented with db-cAMP (0.5 mM; Sigma-Aldrich) and N-N-(3,5-difluorophenacetyl)-1-alanyl-S-phenylglycine t-butyl ester (5 µM; Tocris) at 37°C in 5% $CO_2$ for 7–14 d depending on the experiment.

### Plasmids and transfection
To overexpress LMX1A/LMX1B, either the Tet-On 3 G inducible system (Clontech), LMX1B-FLAG in pcDNA3.1 (GenScript), or pLVX-puro plasmids were used. The Tet-on 3 G system is based on two plasmids: (i) the CMV-Tet plasmid expressing the Tet-On

**Table 1.   Primers for mutagenesis and deletion constructs**

| Name | Sequence | Source |
|------|----------|--------|
| Human LMX1B[Y309A/L312A] | 5′-CATGATGGCTTCCGCCACGCCGGC GGCCCCACCACAG-3′/5′-CTGTGGTGG GGCCGCCGGCGTGGCGGAAGCCATCAT G-3′ | Eurofins |
| Human LMX1B[Y309A/L312A] construct | 5′-GAATGATGGCATCCGCCACCCCAC TTGCACCGCCACAACAGC-3′/5′-GCT GTTGTGGCGGTGCAAGTGGGGTGGCGG ATGCCATCATTC-3′ | Eurofins |
| Human LMX1B [Δ308-317] | 5′-CAGATCGTGGCCATGGAACAG-3′/ 5′-AGCCATCATGCCCTCCATGC-3′ | Eurofins |
| Human LMX1B [Δ308-317] construct | 5′-CAGATTGTGGCCATGGAGCAGTCA CCGTA-3′/5′-TGCCATCATTCCCTCCAT TCTAGAGGACAGG-3′ | Eurofins |
| Human LMX1A[Y290A/L293A] | 5′-GGAAGGTATCATGAATCCGGCTAC AGCGGCGCCAACACCACAGCAGTTGC- 3′/5′-GCAACTGCTGTGGTGTTGGCG CCGCTGTAGCCGGATTCATGATACCTT CC-3′ | Eurofins |
| Human LC3B[F52A/L53A] | 5′-GTTCTGGATAAAACAAAGGCCGCT GTACCTGACCATGTC-3′/5′-GACATG GTCAGGTACAGCGGCCTTTGTTTTATC CAGAAC-3′ | Eurofins |

transactivator protein, and (ii) TRE3G promoter controlling the expression of a gene of interest, in this case, LMX1A (NM_001174069.1) and LMX1B (NM_001174147.1); empty plasmid was used as a control. cDNAs were synthetized codon optimized for *Homo sapiens* (Eurofins Genomics). DOX was added for 48–72 h. For inducible expression of LMX1A/LMX1B in iPSC-derived neurons, the human synapsin promoter (hsyn) in pRRL plasmid (kindly provided by Prof. James Uney, University of Bristol, Bristol, UK) was used to drive expression of the Tet-On activator protein. The pRRL plasmid was also used as a backbone to generate shRNA reporter constructs: hsyn-GFP-U6-shRNA. Plasmids containing each of the human ATG8 family members in pEGFPC1 were provided by Dr. David McEwan (Beatson Institute for Cancer Research, Glasgow, UK). Acetylation mutant LC3B cDNA was synthesized by Eurofins and subcloned into pEGFPC1. Deletion and site-directed mutagenesis of LMX1A, LMX1B, and LC3B was carried out using the primers listed in Table 1. Plasmid sequences are available on request.

### Viruses, transduction, and stable cell lines

Lentiviruses were produced in HEK293T cells by transient transfection using polyethylenimine (PEI; Sigma-Aldrich). 27 μg of the plasmid of interest was transfected together with 20.4 μg of the packing plasmid pAX2 and 6.8 μg of the envelope plasmid pMGD2. Viruses were harvested 48 h after transfection. Media was collected and centrifuged 1,500 *g* for 5 min and filtered with a 0.45 μm filter. Viruses were concentrated using Lenti-X Concentrator (Clontech). One volume of Lenti-X Concentrator was combined with three volumes of clarified supernatant. The mixture was incubated 1 h at 4°C, then centrifuged at 1,500 *g* for 45 min at 4°C, and the pellet was resuspended in N2B27 media or DMEM media. For viral transduction, neural progenitors were

infected for 3 d with 3 μl viruses/ml of media and then media was removed and replace with complete N2B27 for another 2–4 d. The efficacy of transfection was checked using fluorescence microscopy.

HEK239 cells overexpressing LMX1B-FLAG in a Tet-On 3 G system were generated as follows: cells were plated in 6 cm dishes and transduced with the corresponding lentiviruses in the presence of 10 μg/ml polybrene (Sigma-Aldrich). After 2 d, cells were fed with media supplemented with 1 μg/ml puromycin (for non-inducible expression) together with 800 μg/ml G418 (for the Tet-On 3 G system). These concentrations were selected using a titration kill curve in HEK293T (data not shown). Cells were fed every 3 d. For Tet-On 3 G stable cell lines, prior to the experiment, 500 ng/ml DOX was added to the media for 48 h to induce the expression of the gene of interest.

### Generation of LMX1B KO HEK293T cell lines using CRISPR/Cas9 gene editing

LMX1B KO cell lines were generated using CRISPR gRNAs targeting either exon 3 (gRNA1: 5′-TGTGAACGGCAGCTACGCAA-3′) or exon 4 (gRNA2: 5′-CTTCGACGAGACCTCGAAGG-3′). gRNAs were designed using Broad Institute CRISPick online tool (https://portals.broadinstitute.org/gppx/crispick/public). CRISPR constructs were generated by ligating annealed oligonucleotides (Eurofins) into BbsI-digested pSpCas9(BB)-2A-GFP (PX458; Addgene; Ran et al., 2013). gRNA constructs were transfected into HEK293T cells using Lipofectamine 2000. After 24 h, cells were harvested using Accutase and resuspended in FACS sorting buffer (1× sterile PBS without calcium and magnesium containing 0.5% BSA and 1 mM EDTA. $5 \times 10^6$ cells were resuspended in 1 ml of sorting buffer for FACS sorting. GFP-positive cells were individually sorted by FACS into a 96-well plate, and colonies assessed for LMX1B expression by immunoblotting and qRT-PCR.

### siRNA transfection

In HEK293T cells, siRNA transfection was carried out through reverse transfection protocol with lipofectamine. 3 μl of 20 μM siRNAs (Table 2) were mixed with 2 μl lipofectamine in Opti-MEM reduced-serum media (Thermo Fisher Scientific) and combine with $300 \times 10^3$ cells per well of a 6-well plate. Cells were plated overnight in Opti-MEM media. The following day, cells were fed with DMEM + FBS media. A second forward transfection step was then carried out with the reagents described above. Samples were collected 48 h after the second transfection. For rescue experiments, the LMX1B cDNA was synthesized codon optimized for *H. sapiens*, so that the siRNA would not recognize the targeting sequence (Eurofins Genomics), and this was cloned into the pLVX plasmid (empty pLVX plasmid was used as a control).

### Luciferase assay

The LightSwitch Vector (SwitchGear Genomics) was digested with NheI and XhoI (Biolabs) and 3× tandem repeats of the putative FLAT elements for pro-insulin (German et al., 1992), NURR1, TFEB, ULK1, UVRAG, and PINK1 (a scramble sequence was used as a control [Pajares et al., 2018]; Table 3) were

**Table 2. Oligonucleotides (siRNA; shRNA) used in this study**

| Name | Source | Code |
|---|---|---|
| shRNA LMX1A #1 | Sigma-Aldrich | TRCN0000017217 |
| shRNA LMX1A #2[a] | Sigma-Aldrich | TRCN0000017215 |
| shRNA LMX1B #1 | Sigma-Aldrich | TRCN0000017514 |
| shRNA LMX1B #2[a] | Sigma-Aldrich | TRCN0000017517 |
| Mission PLKO.1-puro non-targeting shRNA | Sigma-Aldrich | SHC002 |
| siLMX1B SMARTpool | Dharmacon | L-012586 |
| MISSION esiRNA esiRNA targeting human MAP1LC3B | Sigma-Aldrich | EHU002651 |
| MISSION esiRNA esiRNA targeting human GABARAP-L1 | Sigma-Aldrich | EHU107971 |
| MISSION esiRNA esiRNA targeting human LC3A | Sigma-Aldrich | EHU088371 |
| siLuciferase (siControl): 5'-CGUACGCGG AAUACUUCGAUU-3' | Eurofins | N/A |

[a]Provided the most effective knockdown and were therefore used for shRNA experiments throughout the study.

annealed connected by linker regions with a BamHI site. Cells were seeded on CELLSTAR 96-well white plate (Greiner Bio-one) in DMEM phenol-red free medium at a density of 60–70%. The following day, cells were transfected with LightSwitch Vector with a FLAT-containing promoter—or a scrambled sequence as control—together with pcDNA3.1 LMX1B-FLAG, LMX1B$^{Y309A/L312A}$-FLAG, or LMX1B$^{\Delta308-317}$-FLAG—or pcDNA 3.1 as control. Plasmids were transfected in a ratio of 0.1 µg/each plasmid: 0.45 µl transfection reagent—lipofectamine (HEK293T). After 24 h, plates were frozen at –80°C to increase luciferase signal. To measure luciferase, the LightSwitch

Luciferase Assay Reagent assay (SwitchGear Genomics) was used according to manufacturer's instructions. Luciferase levels were measured in a Fusion Universal microplate reader (PerkinElmer) with a Photomultiplier tubes voltage of 1,100 and each well was read for 2 s. Relative luciferase levels were normalized to luciferase control signal (pcDNA 3.1 condition).

**GFP-trap IP**

HEK293T cells were plated on 10 cm dishes co-transfected with the corresponding GFP-tagged constructs and LMX1A or LMX1B-FLAG (or the LIR mutated versions). DNA was transfected using 1 mg/ml PEI reagent in a ratio of 1:6 in Opti-MEM. Cells were washed twice with ice-cold PBS, then lysed in 500 µl of GFP-trap lysis buffer containing: 50 mM Tris-HCl, 0.5% NP40, 1 mM PMSF, 200 µM Na$_3$VO$_4$ and 1/50 protease inhibitor tablets (Roche). The samples were incubated on ice for 10 min, and then lysates were slowly forced through a 20 G needle to help break the nuclei. Soluble fractions were obtained by centrifugation at 13,000 $g$ for 10 min at 4°C. 5% of the sample was kept as total lysate sample. The remainder was incubated with 20 µl GFP-trap beads (Chromotek), previously washed with GFP-trap wash buffer containing 50 mM Tris-HCl, 0.25% NP40, 1 mM PMSF, 200 µM Na$_3$VO$_4$ and 1/50 protease inhibitor tablets, for 2 h at 4°C. Then, beads were washed three times with GFP-trap wash buffer and a fourth time with GFP-trap wash buffer 2: 50 mM Tris-HCl, 1 mM PMSF, 200 µM Na$_3$VO$_4$, and 1/50 protease inhibitor tablets. Beads were then resuspended in 30 µl 2× loading buffer.

**CoIP with FLAG antibody**

GFP-LC3B HEK293T cells were plated on 10 cm dishes and transfected with LMX1B-FLAG. DNA was transfected using 1 mg/ml PEI reagent in a ratio of 1:6 in Opti-MEM. Cells were washed with ice-cold PBS and lysed in 300 µl of IP lysis buffer (50 mM Tris-HCl, pH 7.5, 150 mM NaCl, 0.5% Triton X-100, supplemented with protease inhibitors). Samples were incubated on ice for 10 min and were slowly forced through a 20 G needle to help break the nuclei. Soluble fractions were obtained

**Table 3. FLAT element sequences for luciferase reporter assays**

| Name | Sequence | Source |
|---|---|---|
| Pro-insulin FLAT | 5'-CTAGTTTAATAATCTAATTACGGATCCTTAATAATCTAATTACACGTGATTAATAATCTAATTACC-3'/5'-TCGAGGTAATTAGATTAT TAATCACGTGTAATTAGATTATTAAGGATCCGTAATTAGATTATTAAA-3' | Eurofins |
| NURR1 FLAT | 5'-CTAGTAAAATAATAACTGCGGATCCAAAATAATAACTGCACGTGAAAAATAATAACTGCC-3'/5'-TCGAGGCAGTTATTATTTTTCACG TGCAGTTATTATTTTGGATCCGCAGTTATTATTTTA-3' | Eurofins |
| ULK1 FLAT | 5'-CTAGTGATTATAATTAAGTAGGATCCGATTATAATTAAGTAACGTGAGATTATAATTAAGTAC-3'/5'-TCGAGTACTTAATTATAATCT CACGTTACTTAATTATAATCGGATCCTACTTAATTATAATCA-3' | Eurofins |
| TFEB FLAT | 5'-CTAGTGTGCAGGTTAATTACTGCCCAGGATCCGTGCAGGTTAATTACTGCCCAACGTGAGTGCAGGTTAATTACTGCCCAC-3'/5'-TCG AGTGGGCAGTAATTAACCTGCACTCACGTTGGGCAGTAATTAACCTGCACGGATCCTGGGCAGTAATTAACCTGCACA-3' | Eurofins |
| PINK1 FLAT | 5'-CTAGTGCTGTTAAATAAATTAAAAGACGGATCCGCTGTTAAATAAATTAAAAGACACGTGAGCTGTTAAATAAATTAAAAGACC-3'/5'-TCGAGGTCTTTTAATTTATTTAACAGCTCACGTGTCTTTTAATTTATTTAACAGCGGATCCGTCTTTTAATTTATTTAACAGCA-3' | Eurofins |
| UVRAG FLAT | 5'-CTAGTCATCCTTTTAATGAAATCATATGGATCCCATCCTTTTAATGAAATCATATACGTGACATCCTTTTAATGAAATCATATC-3'/5'-TCGAGATATGATTTTCATTAAAAGGATGTCACGTATATGATTTTCATTAAAAGGATGGGATCCATATGATTTTCATTAAAAGGATGA-3' | Eurofins |
| Scrambled control | 5'-CTAGTTCAGATTCACGGGATCCTCAGATTCACGGTCGACTCAGATTCACGC-3'/5'-TCGAGCGTGAATCTGAGTCGACCGTGAATCTGA GGATCCCGTGAATCTGAA-3' | Eurofins |

by centrifugation at 13,000 *g* for 10 min at 4°C. 5% of the sample was kept as total lysate sample. A total of 200 µg was used for each coIP (sample protein concentration was measured using nanodrop A280) and sample was diluted in IP wash buffer (400 µl total volume, 50 mM Tris-HCl, pH 7.5, 150 mM NaCl, 0.1% Triton X-100, supplemented with protease inhibitors) with 3 µg of FLAG antibody—or IgG mouse (Millipore) as a control—and incubated in a rotating wheel at 4°C for 3 h. Immunocomplexes were recovered by incubation on a rotating wheel at 4°C for 2 h with 30 µl protein G sepharose (GE HealthCare) previously washed with IP wash buffer. Then, beads were washed four times with IP wash buffer, resuspended in 30 µl 2× loading buffer, and analyzed by immunoblotting.

### Native LMX1B IP
HEK293T cells were seeded in 150 mm tissue culture dishes. At ~70% confluency, media was aspirated, and cells were washed with ice-cold PBS. 700 µl of cell lysis buffer (Cell Signaling Technology) supplemented with 10% glycerol, 0.5 mM DTT, and 1/50 protease inhibitor tablets was added to the dish and cells were scraped into 1.5 ml tubes. Tubes were incubated on ice for 30 min, vortexing occasionally. Samples were then sonicated (Diagenode Bioruptor) using 3 × 30 s pulses at 4°C, and lysates clarified by centrifugation at 12,000 *g* at 4°C for 15 min. Protein concentration was estimated by BCA assay. Prior to the addition of antibody, lysates (2,000 µg of protein) were precleared with IgG cross linked magnetic beads for 2 h at 4°C while rotating. 33 µl of Protein A beads (Dynabeads, Thermo Fisher Scientific) were cross-linked with either 6 µg normal rabbit IgG (SC-225; Santa Cruz) or 6 µg of anti-LMX1B antiserum (Witzgall), and these were added to precleared lysates and incubated overnight at 4°C. Beads were washed with cell lysis buffer for 5 × 5 min washes at 4°C under rotation. After the final wash, supernatant was aspirated and 40 µl of 2× sample buffer was added to the beads. Samples were boiled at 95°C for 5 min, and after separation of magnetic beads, supernatant was subjected to SDS-PAGE and immunoblotting.

### Recombinant protein purification and in vitro binding assays
Human ATG8 sequences were cloned into His-tag plasmid (ptrHisC), and these were transformed into *Escherichia coli* BL21 (DE3) bacteria (Biolabs). To induce expression, 0.5 mM IPTG (Sigma-Aldrich) was added for 3 h. The culture was centrifuged at 1,500 *g* for 10 min 4°C, and the pellet resuspended in 13 ml homogenization buffer comprising: 25 mM Tris, pH 7.5, 1% Triton X-100, 250 mM NaCl, 20 mM imidazole (Sigma-Aldrich). The sample was sonicated on ice using a sequence of 10 s on/20 s off for 5 min. The soluble fraction was harvested after centrifugation at 3,000 *g* for 30 min at 4°C and incubated with 1 ml nickel-chelating Resin (Probond) previously washed twice with homogenization buffer (1 ml resin/10 ml buffer) for 1 h at 4°C. Poly-Prep Chromatography Columns (Bio-Rad) were used to pack the resin before elution. The column was extensively washed with homogenization buffer without imidazole, and His-ATG8s were eluted with 500 µl elution buffer (20 mM Tris, pH 7.5, 1 M NaCl, 100 mM EDTA, 200 mM imidazole). Recombinant ATG8 proteins were then coupled to cyanogen bromide-activated sepharose 4 Fast Flow (CNBr-sepharose; GE Healthcare) as described (Kavran

and Leahy, 2014). First, the recombinant proteins were dialyzed using Slide-A-Lyzer cassettes (Thermo Fisher Scientific) into cold coupling buffer containing 100 mM NaHCO$_3$ and 500 mM NaCl. To activate the resin, 0.25 g of resin was incubated with 5 volumes of 1 mM HCl for 2 h at 4°C producing 1 ml of hydrated resin. Resin was washed with 1 mM HCl, then 2 mg of recombinant protein (measured using nanodrop A280) was coupled to 1 ml of hydrated resin overnight at 4°C. As a negative control, blank resin without incubation with recombinant protein was prepared to test for unspecific binding. The reaction was quenched by incubation with quenching buffer containing 100 mM Tris-HCl, pH 7.5, for 3 h at 4°C. Uncoupled protein was removed by washing with high pH/low pH wash buffers comprising: 100 mM Tris-HCl, pH 8, 500 mM NaCl/100 mM NaOAc, 500 mM NaCl. For in vitro binding assays, HEK293T cells grown to subconfluency on 10 cm dishes were washed with ice-cold PBS and lysed with 500 µl of radioimmunoprecipitation assay (RIPA) buffer supplemented with protease inhibitor. The samples were incubated on ice for 10 min and then lysates were diluted 1:2 with CNBr IP wash buffer (50 mM Tris-HCl, pH 7.5, 150 mM NaCl, 1 mM EDTA, 1/50 protease inhibitor tablets [Roche]). The homogenates were incubated on ice for 15 min and were cleared by centrifugation at 12,000 *g* for 15 min at 4°C. Prior to the incubation, resin was washed in CNBr IP wash buffer. For the binding, 250 µl of the cleared lysate diluted with 140 µl of CNBr IP wash buffer was incubated with 30 µl of the coupled resin overnight 4°C. Then, beads were washed three times with CNBr IP wash buffer, then resuspended in 40 µl 2× loading buffer.

### Cell fractionation
HEK293T cells were plated on 10 cm dishes and transfected with the LMX1B-FLAG or LMX1A with/without GFP-ATG8s cDNAs using 1 mg/ml PEI reagent at a ratio of 1:6 in Opti-MEM for 24 h. Cells were washed twice in 5 ml of ice-cold PBS and harvested by centrifugation at 200 *g* for 5 min 4°C. The cell pellet was then lysed with 400 µl of Buffer A (20 mM Hepes, pH 7 [Sigma-Aldrich], 0.15 mM EDTA, 0.015 mM EGTA [Sigma-Aldrich], 10 mM KCl [Sigma-Aldrich], and 1% NP-40 [Sigma-Aldrich] supplemented with one tablet of protease inhibitor per 10 ml of buffer, as described in Garcia-Yague et al. [2013]) and incubated on ice for 30 min. The homogenate was centrifuged at 1,000 *g* for 5 min 4°C, after which the supernatant was collected, and the nuclear pellet washed in 500 µl Buffer B (10 mM Hepes, pH 8, 25% glycerol, 0.1 M NaCl, 0.1 mM EDTA supplemented with one tablet of protease inhibitor per 10 ml of buffer). After centrifugation as above, the nuclear pellet was incubated with DNaseI (Thermo Fisher Scientific) in 100 µl of Buffer A for 20 min and then 4× loading buffer was added to the sample. If the nuclear fraction was required for GFP-trap IP, after washing with Buffer B, it was resuspended in 300 µl of Buffer C (10 mM Hepes, pH 8, 25% glycerol, 0.4 M NaCl, and 0.1 mM EDTA supplemented with one tablet of protease inhibitor per 10 ml of buffer) for 30 min at 4°C. Samples were centrifuged at 4,500 *g* 20 min 4°C with soluble nuclear proteins located in the supernatant. 120 µl of this fraction was kept as total nuclear fraction and the rest of the fraction was diluted up to 600 µl with GFP-trap wash buffer for the incubation with GFP-trap beads.

## Protein turnover experiments

Stable HEK293T LMX1B-FLAG cells were plated on 6-well plates at 70–90% confluency. Cells were treated for 16 h with CHX (50 µg/ml) in the absence or presence of BafA1 (20 nM) or MG132 (10 µM). For starvation experiments, cells were treated for 6 h with CHX (50 µg/ml) in starvation media in the absence or presence of BafA1 (20 nM). Lysates were collected for immunoblotting.

## Immunoblotting

Cells grown on 6-well plates were initially washed with ice-cold PBS, then lysed with 100–200 µl/well of ice-cold RIPA buffer consisting of 50 mM Tris-HCl (pH 7.4), 1% Triton-X-100 (Sigma-Aldrich), 0.5% sodium deoxycholate (Sigma-Aldrich), 150 mM NaCl (Sigma-Aldrich), 0.1% SDS (Sigma-Aldrich) supplemented with one tablet of protease inhibitor per 10 ml of RIPA buffer. The homogenates were incubated on ice for 15 min, and then cleared by centrifugation at 12,000 g for 15 min at 4°C. Supernatants were collected as soluble fractions. Sample protein concentration was determined by BCA protein assay (Pierce) according to manufacturer's protocol. Proteins were transferred to nitrocellulose membranes (Biolabs). Membranes were then incubated with primary antibody diluted in 2.5% milk or 2.5% BSA in Triton X-100-TBS buffer for 2 h or overnight. Primary and secondary antibodies used are listed above. Membranes were then washed three times prior to incubation with ECL chemiluminescence reagents (Geneflow), and band intensities were detected in films (GE Healthcare) using a film developer.

## ChIP

HEK293T cells or iPSC-derived mDANs were plated on 10 cm dishes at 80–90% confluency. ChIP assays were conducted based on previously published protocols (Pescador et al., 2005). Firstly, cells were fixed with 625 µl 16% formaldehyde added to the media (Thermo Fisher Scientific) on ice for 12 min and the crosslinking reaction was stopped with 125 mM glycine (Sigma-Aldrich). Cells were then washed twice with ice-cold PBS supplemented with protease inhibitors and harvested by centrifugation at 200 g for 5 min at 4°C. The pellet was lysed with 200 µl with ChIP lysis buffer (1% SDS, 10 mM EDTA, and 50 mM Tris, pH 8.1, supplemented with protease inhibitors) and incubated on ice for 10 min. Samples from different dishes were pooled for sonication on ice using a probe sonicator with a sequence of 15 s on/15 s off for 8 min/plate to obtain an adequate fragment size of DNA (800–200 bp). The homogenates were cleared by centrifugation at 12,000 g for 10 min at 4°C, and the supernatants collected as soluble fractions. Samples were then diluted in 10 volumes of ChIP dilution buffer comprising 0.01% SDS, 1.1% Triton X-100, 1.2 mM EDTA, 16.7 mM Tris, pH 8.1, and 167 mM NaCl supplemented with protease inhibitors. The lysates were then precleared for 1 h at 4°C using 10 µl protein A agarose 50% slurry/plate (Millipore) and collected by centrifugation at 300 g for 3 min at 4°C (100 µl of sample/plate was kept as input chromatin). The remaining lysate (~2 ml/plate) was used per ChIP experiment (approximately 1 × 10 cm dish per IP). Each sample was incubated overnight at 4°C with 4 µg of anti-LMX1B (Proteintech), anti-LMX1A (PriSci) or 4 µg of rabbit IgG

(Cell Signaling). Immunocomplexes were recovered by incubation with 60 µl pre-washed protein A/sample for 1 h at 4°C. Prior to the elution of DNA, samples were incubated sequentially in the following buffers to improve the removal of non-specific chromatin interactions: (1) ChIP low salt buffer containing 0.1% SDS, 1% Triton X-100, 2 mM EDTA, 20 mM Tris, pH 8.1, and 150 mM NaCl; (2) ChIP high salt buffer containing 0.1% SDS, 1% Triton X-100, 2 mM EDTA, 20 mM Tris, pH 8.1, and 500 mM NaCl; (3) ChIP lithium buffer containing 1% Igepal (Sigma-Aldrich), 1 mM EDTA, 10 mM Tris, pH 8.1, 250 mM LiCl, and 1% sodium deoxycholate; and (4) twice in TE wash buffer containing 10 mM Tris, pH 8, and 1 mM EDTA. Finally, eluted samples were obtained by incubation with 250 µl of ChIP elution buffer containing 0.1 M NaHCO$_3$ and 1% SDS for 15 min at room temperature (this process was repeated twice so the total eluate was 500 µl). Input chromatin samples were also diluted in ChIP elution buffer (500 µl final volume). To reverse the crosslinking, all samples (including input chromatin) were incubated with 20 µl 5 M NaCl at 65°C overnight. Then, 10 µl 0.5 M EDTA, 20 µl 1 M Tris, pH 6.5, and 2 µl proteinase K (Thermo Fisher Scientific) was added to each sample and samples were incubated for 1 h at 45°C. For the purification of DNA, the phenol-chloroform purification was used. Finally, the DNA pellet was resuspended in a final volume of 30 µl and diluted 1:3 for qPCR analysis with specific primers designed with the information obtained by bioinformatics analysis using the UCSC genome browser (Table 4).

## Analysis of mRNA levels by qRT-PCR

HEK293T cells and iPSC-derived mDANs were plated on 6-well or 12-well plates, respectively. Cells were allowed to mature for 3 d before transducing with the corresponding viruses for 3 d. Then, viruses were removed and replaced with complete N2B27 for another 2–4 d. After the corresponding treatment/transduction, cells were washed with PBS and then cells were lysed in 350 µl RLT buffer (Qiagen). Total RNA was extracted through columns using RNeasy kit (Qiagen) following manufacturer's instructions and genomic DNA was digested using DNaseI (Qiagen). RNA samples were reverse-transcribed using High-Capacity RNA-to-cDNA Kit (Thermo Fisher Scientific), according to manufacturer's protocol. The cDNA samples were amplified using SYBR Green (Life Technologies) using the primers listed in Table 5. The reaction was carried out using StepOnePlus System (Applied Biosystems) and the following conditions were selected: after an initial denaturation at 95°C for 10 min, 40 cycles with 95°C for 15 s (denaturation), 60°C for 30 s (annealing), and 60°C for 30 s (elongation). For the analysis, mRNA levels were estimated using the ΔΔCt method (Livak and Schmittgen, 2001) normalizing data to GAPDH levels.

## Fluorescence microscopy

For immunofluorescence, cells were seeded on coverslips (pre-coated with poly-L-ornithine and laminin for neurons). iPSC-derived mDANs were allowed to mature for 7–1 d prior to fixation. Cells were washed twice with PBS and incubated with 4% formaldehyde (Thermo Fisher Scientific) for 20 min, or with –20°C methanol for 5 min. Formaldehyde-fixed cells

**Table 4.  Primers used for ChIP qPCR**

| Name | Sequence | Source |
|---|---|---|
| ABRA | 5′-TCAGATGCCGTTGAACTCTG-3′/5′-GCCGCTATTCGT TTTTCATC-3′ | Eurofins |
| ATG16L1 | 5′-TATATCACCTCCTGTGACACC-3′/5′-ACTTTACCATAA GTTGACATATC-3′ | Eurofins |
| ATG3 | 5′-CAGGAATAACGGAAGCCGTTAAAG-3′/5′-CAGTAG TGTTTTGGTGTGTTGAGAC-3′ | Eurofins |
| ATG7 | 5′-AAAGACCTGAGCTTGTGACCATAGG-3′/5′-ATGTGC TGTGATGTTTGACAAGAC-3′ | Eurofins |
| GAPDH | 5′-CGGGATTGTCTGCCCTAATTAT-3′/5′-GCACGGAAG GTCACGATGT-3′ | Eurofins |
| NDP52 (1) | 5′-GCACTTCTTATTCGATTCATTTG-3′/5′-TTGTAGGCA CTTCACACAAAG-3′ | Eurofins |
| NDP52 (2) | 5′-GCTTTTTCCCTCCTTGGTCACT-3′/5′-CTGGGTACA AGGTGAGAAATTGT-3′ | Eurofins |
| NURR1 | 5′-GCGTGCAGAGTGAATGTATCTA-3′/5′-TTCAGGCAG TCGGAAACTCAA-3′ | Eurofins |
| OPTN | 5′-TGCCTGGCATTCTCCTCTTTCT-3′/5′-AACAGGGAC TGCTCTAAGGCGTC-3′ | Eurofins |
| p62 | 5′-CTCTCAGGCGCCTGGGCTGCTGAG-3′/5′-CGGCGG TGGAGAGTGGAAAATGCC-3′ | Eurofins |
| PINK1 | 5′-GTAGCTCAGCTCTGCTAGGTAC-3′/5′-CAGGTCCTG AATGTGAACATCA-3′ | Eurofins |
| PITX3 | 5′-GCGCCCGGCCTATAGTCTACAT-3′/5′-CATGCTGAG AGGTTCTCTGCAT-3′ | Eurofins |
| TFEB | 5′-TGTTCTGGGGACGGTTCAGCGC-3′/5′-CCTTTCCCT GAGGGGATGAAGCAGC-3′ | Eurofins |
| ULK1 | 5′-TAAATCCGCTGGGGAGGAAAGG-3′/5′-ACGACCATG TACACATTACAGG-3′ | Eurofins |
| UVRAG | 5′-CTCATCAAACTTATGGAACTCT-3′/5′-TATTGTGGT ATCAGGGAAGGT-3′ | Eurofins |

**Table 5.  Primers used for qRT-PCR following siRNA/shRNA knockdown**

| Name | Sequence | Source |
|---|---|---|
| ATG16L1 | 5′-CAGTTACGTGGCGGCAGGCT-3′/5′-ACAACG TGCGAGCCAGAGGG-3′ | Eurofins |
| ATG2B | 5′-AACTCACAAACAGAATGGTTCAAA-3′/5′-AAG GGTACCAGGAAGACACCA-3′ | Eurofins |
| ATG3 | 5′-CATGCAGGCATGCTGAGGTG-3′/CGTTAACAG CCATTTTGCCACT-3′ | Eurofins |
| ATG5 | 5′-AGCAACTCTGGATGGGATTG-3′/5′-CACTGC AGAGGTGTTTCCAA-3′ | Eurofins |
| ATG7 | 5′-CAATCTGGGCTAAATGCCATTTCTGGAAG-3′/ 5′-AGCCCAGTACCCTGGATGG-3′ | Eurofins |
| COL4A3 | 5′-GCAGATGCACTGTTTGTGAA-3′/5′-GTTTAA TGAAGCCAGCCAGA-3′ | Eurofins |
| COL4A4 | 5′-CTCCTGGTTCTCCACAGTCAG-3′/5′-TGTTGC AGTAGGCAAAGGGCA-3′ | Eurofins |
| GAPDH | 5′-TTGAGGTCAATGAAGGGGTC-3′/5′-GAAGGT GAAGGTCGGAGTCA-3′ | Eurofins |
| LMX1A | 5′-AGAGCTCGCCTACCAGGTC-3′/5′-GAAGGAGGC CGAGGTGTC-3′ | Eurofins |
| LMX1A (construct) | 5′-GATGGCTTGAAGATGGAGGA-3′/5′-TCTCAG CAGAAAGCGATCCA-3′ | Eurofins |
| LMX1B | 5′-GTGTGAACGGCAGCTACGC-3′/5′-TCATCCTCG CTCTTCACGG-3′ | Eurofins |
| LMX1B (construct) | 5′-GGTCAGACGGATTGTGCCAA-3′/5′-CTCTTC ATGCCAGCTAGACTC-3′ | Eurofins |
| MSX1 | 5′-CGAGAGGACCCCGTGGATGCAGAG-3′/5′-GGC GGCCATCTTCAGCTTCTCCAG-3′ | Eurofins |
| NDP52 | 5′-ACCATGGAGGAGACCATCAA-3′/5′-TTCTGG ACGGAATTGGAAAG-3′ | Eurofins |
| NURR1 | 5′-GTGTTCAGGCGCAGTATGG-3′/5′-TGGCAGTAA TTTCAGTGTTGG-3′ | Eurofins |
| OPTN | 5′-TGCTGAGTCCGCACATAGA-3′/5′-GGGTCCATT TCCTGTGCTT-3′ | Eurofins |
| p62 | 5′-CTGGGACTGAGAAGGCTCAC-3′/5′-GCAGCT GATGGTTTGGAAAT-3′ | Eurofins |
| PINK1 | 5′-GCCTCATCGAGGAAAACAGG-3′/5′-GTCTCG TGTCCAACGGGTC-3′ | Eurofins |
| PITX3 | 5′-ACTAGCCCTCCCTCCAT-3′/5′-TTTCAGCGA ACCGTCCT-3′ | Eurofins |
| TFEB | 5′-CCAGAAGCGAGAGCTCACAGAT-3′/5′-TGTGAT TGTCTTTCTTCTGCCG-3′ | Eurofins |
| TH | 5′-GCCGTGCTAAACCTGCTCTT-3′/5′-GTCTCA AACACCTTCACAGCTC-3′ | Eurofins |
| TUJ1 | 5′-ATGAGGGAGATCGTGCACAT-3′/5′-GCCCCT GAGCGGACACTGT-3′ | Eurofins |
| ULK1 | TCATCTTCAGCCACGCTGT-3′/5′-CACGGTGCT GGAACATCTC-3′ | Eurofins |
| UVRAG | 5′-CGGAACATTGCTGCCCGGAACA-3′/5′-TCGCCA CGTGGGATTCAAGGAAT-3′ | Eurofins |

were incubated with blocking solution containing 5% BSA and 0.3% Triton X-100 (Sigma-Aldrich) in PBS for at least half an hour at room temperature. Cells were then incubated 2 h/ overnight with primary antibody (listed above) prepared in 2.5% BSA (including 0.15% Triton X-100 for neurons). Cells were washed three times with PBS and incubated with the secondary antibodies (listed above) and counterstained with DAPI prepared in 2.5% BSA (and 0.15% Triton X-100 for neurons) for 1 h. Cells were then washed again with PBS, mounted in Mowiol, and fluorescence images were captured at room temperature using a Leica DMI6000 SP5-II confocal microscope using 40× or 63× HCX PL APO CS lenses (NA 1.25; NA 1.4). Image analysis was performed using Fiji (Schindelin et al., 2012; National Institutes of Health).

For live imaging of mDANs, droplets of cells were plated on poly-L-ornithine/laminin pre-coated live imaging dishes (35-mm glass-bottomed dishes; MatTek). After 4–6 d, mDANs were transduced with hsyn-tet TRE-GFP-LMX1B for 3 d with 3 µl viruses/ml of media, and then media was removed and replaced with complete N2B27 for another 2–4 d. Phase contrast and fluorescence images were captured in 1 h intervals for 15 h after

the addition of DOX (500 ng/ml) at 37°C using an Olympus IX-71 inverted microscope hosting a 40× UplanFL N objective (NA 0.75) fitted with a CoolSNAP HQ CCD camera (Photometrics),

with image capture and analysis performed using MetaMorph software (Molecular Devices).

## IncuCyte cell imaging

iPSC-derived mDANs were seeded to mature in 24-well imaging plates (Thermo Fisher Scientific) and were transduced with hsyn-GFP/mcherry-U6-shRNAs viruses. Cells were imaged at 37°C with 5% $CO_2$ using the IncuCyte S3 Live-Cell Analysis System (Essen Bioscience) at 20× magnification (NA 0.45), with images obtained every 1 h by phase contrast, green and red channels over a period of 4 d. Analysis was carried out by Dr. Stephen Cross (Wolfson facility, University of Bristol). Analysis of neurite length and fluorescence intensity was performed in Fiji, using the Modular Image Analysis plugin (Cross, 2017). First, regions of each phase contrast image corresponding to cell bodies were removed using a mask image. Each mask was created by applying a variance filter to the phase contrast channel image, which enhanced large objects with high contrast, such as cell bodies. The filtered image was subsequently binarized using the Huang method (Xiao et al., 2011; set to 80% absolute value), holes in the mask filled and a median filter applied to smooth the object borders. The mask was applied to the original phase contrast image such that the masked regions had intensity close to the phase contrast image to minimize false object detection at the mask boundaries. Neurites were identified in the unmasked regions using the Ridge Detection plugin for Fiji with a minimum neurite length of 15px. The length of each neurite was measured, along with the green and red fluorescence channel intensities coincident with each neurite.

## FLIP experiments

HeLa cells (wild-type and ATG8 KO [Nguyen et al., 2016]) grown in glass-bottom imaging dishes (MatTek Corp.) were transiently transfected with GFP-LMX1B using the Tet-On 3 G system (8 h) and expression was induced by addition of DOX (500 ng/ml) for 20 h. Media was replaced either with fresh growth medium or with starvation media and cells were incubated for a further 2 h before imaging. FLIP was carried out at 37°C using a Leica DMI6000 SP8 confocal microscope, using a 40× HC PL APO CS2 lens (NA 1.4). A photobleaching area was established in the cytoplasm of the target cell, and regions of interest were fixed in the nuclei of the target cell and neighboring reference cells. FLIP was carried out by continuous excitation of the bleach area at 100% laser power (65 mW Ar laser), with integrated fluorescence intensity measured continuously at 2% laser power in the regions of interest using hybrid GaAsP detectors.

## Seahorse bioenergetics

Day 30–50 iPSC-derived mDANs were plated according to manufacturer's instructions on 8-well Seahorse XFp plates (Agilent) previously coated with Poly-*L*-ornithine and laminin. Cells were allowed to mature for 3 d before being infected with hsyn-GFP-U6-shRNAs viruses for 3 d. Viruses were then removed, and media replaced with complete N2B27 for another 2–4 d. The day of the assay, culture media was replaced with Seahorse XF base medium (Agilent) supplemented with 1 mM sodium pyruvate, 2 mM glutamine, and 10 mM glucose (pH 7.4) for 1 h at 37°C.

The Mito Stress Test Kit (Agilent) was prepared according to the manufacturer's instructions: oligomycin (1 μM); carbonilcyanide ptriflouromethoxyphenylhydrazone (FCCP; 1 μM); rotenone/antimycin A (0.5 μM). After analysis, cells were lysed in 20 μl of RIPA buffer and protein levels were quantified by Nanodrop (A280).

## Cell death assays

iPSC-derived mDANs were matured for 3 d in 96-well plates, then transduced with viruses for a further 3 d. Media was replaced with complete N2B27 with DOX (500 ng/ml) for another 2–3 d. Rotenone (15 μM) or DMSO were added for 24 h, and cells washed with PBS before plates were frozen at −80°C. Caspase levels were measured using the EnzChek Caspase-3 Assay Kit Z-DEVD-AMC substrate (Thermo Fisher Scientific) according to manufacturer's instructions. Protein concentration was quantified by Nanodrop (A280) to normalize the data. Lysates were transferred to a Costar 96-well black clear bottom plate (Thermo Fisher Scientific) and substrate working solution was added for 30 min. Fluorescence was measured in Glomax plate reader (Promega) with the UV module (365 nm excitation and 410–465 emission). Relative caspase levels were normalized to the control (TRE-empty untreated condition).

Control and LMX1B KO KEK293T cell lines were seeded in 6-well plates (0.15 million/well). On day 2 cells were infected either with pLVX-CMV empty, pLVX-CMV-wild-type LMX1B or pLVX-CMV-Δ308-317 LMX1B. On day 3, virus was removed and added fresh media containing either DMSO or rotenone. After 48 h hours, all cells were harvested by Accutase treatment, then mixed with trypan blue and injected into cell counting chamber slides (C10283; Invitrogen). Live and dead cells were counted using a Countess automated cell counter (Invitrogen).

## Phylogenetic tree assembly

LMX1A/B orthologs were identified using protein-Basic Local Alignment Search Tool (NCBI). Regions of low compositional complexity were masked to avoid misleading results and an e-value threshold of $10^{-80}$ was used. Representative organisms from every phylum/family were selected. Protein sequences were aligned using MUSCLE software (EMBL). To generate the LMX1A/B phylogenetic tree, Molecular Evolutionary Genetics Analysis 7.0 (MEGA) software (Pennsylvania) was used (Hall, 2013). A maximum likelihood method with 2,000 bootstraps and Jones-Taylor-Thornton (JTT) substitution model was applied (Jones et al., 1992)—the best-fit substitution model found for our alignments. The bootstrap consensus tree was inferred from 2,000 replicates (Felsenstein, 1985), calculating percentage replicate trees with associated taxa (Jones et al., 1992). Evolutionary analyses were conducted in MEGA (Kumar et al., 2016).

## Bioinformatics

Putative LMX1A/B sites in candidate promoter sequences were predicted using MatInspector (Genomatix; Cartharius et al., 2005), which identifies transcription factor binding sites using frequency matrices based on published experimental data. Each matrix has an associated random expectation value, indicating how well a matrix is defined (being 0.08 and 0.01% for LMX1A

and LMX1B, respectively). This process assigns a maximum score (i.e., probability index), and sequences with relative scores >80%—a commonly used threshold for TFBS (computational framework for transcription factor binding site) analyses using Position Specific Scoring Matrix. To identify putative LMX1A/B LC3-interacting regions (LIR motifs), we used the iLIR Autophagy database (https://ilir.warwick.ac.uk; Jacomin et al., 2016), applying ANCHOR software to predict flanking, stabilization regions predicted to stabilize the putative binding. To identify the nuclear localization sequence (NLS) in LMX1A and LMX1B, we used the cNLS Mapper (Kosugi et al., 2009) which predicted the NLS sequence with a score of 7 and 10, respectively.

### Computational modeling of the LC3B–LMX1B interaction
Molecular graphics manipulations and visualizations were performed using VMD-1.9.1 and Chimera-1.10.2 (Pettersen et al., 2004). Chimera was used to visualize the 5d94.pdb crystal structure of the LC3B protein and the FYCO1 peptide (Edwards et al., 2015), and the FYCO1 peptide was used as the "template" to guide the positioning of the LMX1B LIR and flanking sequences. Ultimately the phenylalanine (of FYCO1 LIR) and the required tyrosine (of LMX1B LIR) were aligned and flanking residues on the FYCO1 peptide were altered to mimic the LMX1B residues. Pdb2gmx was used to prepare the assemblies using the v-site hydrogen option to allow a 5 fs time step. Hydrogen atoms were added consistent with pH 7 and parameterized with the AMBER-99SB-ildn forcefield. Each complex was surrounded by a box 2 nm larger than the polypeptide in each dimension and filled with TIP3P water. The GROMACS-5.1.5 suite of software was used to set up, energy minimize and perform the molecular dynamics simulations of the resulting assemblies. Random water molecules were replaced by sodium and chloride ions to give a neutral (uncharged overall) box and an ionic strength of 0.15 M. Each assembly was subjected to 5,000 steps of energy minimization, velocities were generated with all bonds restrained, prior to molecular dynamics simulations. The LMX1B–LIR/lC3B complex was molecular dynamics simulated for 75 ns, throughout which the LIR peptide remained in contact—this was considered a reasonable binding pose. All simulations were performed as NPT ensembles at 298 K using periodic boundary conditions. Short range electrostatic and van der Waals interactions were truncated at 1.4 nm while long range electrostatics were treated with the particle-mesh Ewald's method and a long-range dispersion correction applied. Pressure was controlled by the Berendsen barostat and temperature by the V-rescale thermostat. The simulations were integrated with a leap-frog algorithm over a 5 fs time step, constraining bond vibrations with the P-LINCS method and SETTLE for water. Structures were saved every 0.1 ns for analysis and each run over 50 ns. Simulation data were accumulated on Bristol University Bluecrystal phase 4 HPC. In each case Root Mean Square Deviation was calculated from the trajectories to give an indication of peptide flexibility over the course of the simulations. Root Mean Square Fluctuations were calculated to indicate the flexibility of individual peptide residues over the trajectories. Images were produced with Chimera and Microsoft Paintshop or GIMP (GNU Image Manipulation Program).

### Image analysis and statistics
Fluorescence intensity and puncta quantification were carried out using Fiji. Graphical results were analyzed with GraphPad Prism 7 (GraphPad Software), using an unpaired, two-tailed Student's $t$ test or ANOVA with the appropriate statistical test, as indicated in figure legends. Results are presented as mean ± SEM or mean ± SD, as indicated in figure legends. Data distribution was assumed to be normal, but this was not formally tested.

### Online supplemental material
Fig. S1 shows phylogenetic tree analysis of LMX1B with Incucyte and qRT-PCR data showing that LMX1A and LMX1B are needed for maintenance of dopaminergic neuronal fate and neuronal maturation in developing iPSC-derived mDANs. Fig. S2 demonstrates the effectiveness of doxycycline-induced expression of LMX1A and LMX1B for enhancing autophagy and associated gene expression in human mDANs. Fig. S3 shows the current Alphafold structure of LMX1B with in silico modeling of the LMX1B LIR-like peptide with LC3B. Fig. S4 shows ATG8 binding properties of human LMX1A, while Fig. S5 shows data on the minimal impact on ATG8 binding and LMX1B transcriptional control in the Y309A/L312A mutant. Fig. S6 shows how LC3A siRNA knockdown does not suppress LMX1B transcriptional control of selected target genes. Video 1 shows a computer simulation of the LMX1B–LC3B interaction. Table S1 shows MatInspector promoter FLAT sequence analysis for selected candidate genes.

## Acknowledgments
We are grateful to the Wolfson Bioimaging facility for confocal microscopy and Incucyte support, and in particular Stephen Cross for assistance with the neurite growth analysis assay.

This work was supported by a Wellcome Trust Ph.D. studentship awarded to N. Jiménez-Moreno through the Dynamic Cell Biology programme (grant number 083474). M. Kollareddy was supported by the Biotechnology and Biological Sciences Research Council (BB/T016183/1). Z. Antón was supported by the Biotechnology and Biological Sciences Research Council and Engineering and Physical Sciences Research Council through the BrisSynBio Synthetic Biology Research Centre (BB/L01386X1). J.J. Moss was supported by the Wellcome Trust (083474). P. Stathakos was supported through a Medical Research Council Doctoral Training Grant studentship.

Author contributions: J.D. Lane and N. Jiménez-Moreno conceptualized the project. J.D. Lane acquired the funding, supervised the project, analyzed the data, and assembled the figures. N. Jiménez-Moreno performed the bioinformatic analyses and carried all wet experiments with the exception of the following: the native protein IPs, the majority of the siRNA luciferase assays, the cell fractionation for Fig. 4 D, the LMX1B CRISPR editing and characterization (including Seahorse analysis, cell death, qRT-PCR, autophagy flux immunoblotting) that were carried out by M. Kollareddy; Seahorse experiments in iPSC neurons (Z. Antón); image-based autophagy flux in LMX1B KO cells (J.D. Lane). J.J. Moss designed and validated the LMX1B

CRISPR gRNAs and carried out initial KO trials. Computational modeling was performed by R.B. Sessions and D.K. Shoemark. The mDAN differentiation protocol was developed by P. Stathakos who also provided training and technical support for the iPSC work. M. Caldwell and R. Witzgall contributed to methodology and provided reagents. J.D. Lane and N. Jiménez-Moreno co-wrote the manuscript. All authors critically reviewed and edited the manuscript.

Disclosures: The authors declare no competing interests exist.

Submitted: 18 October 2019

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

# Supplemental material

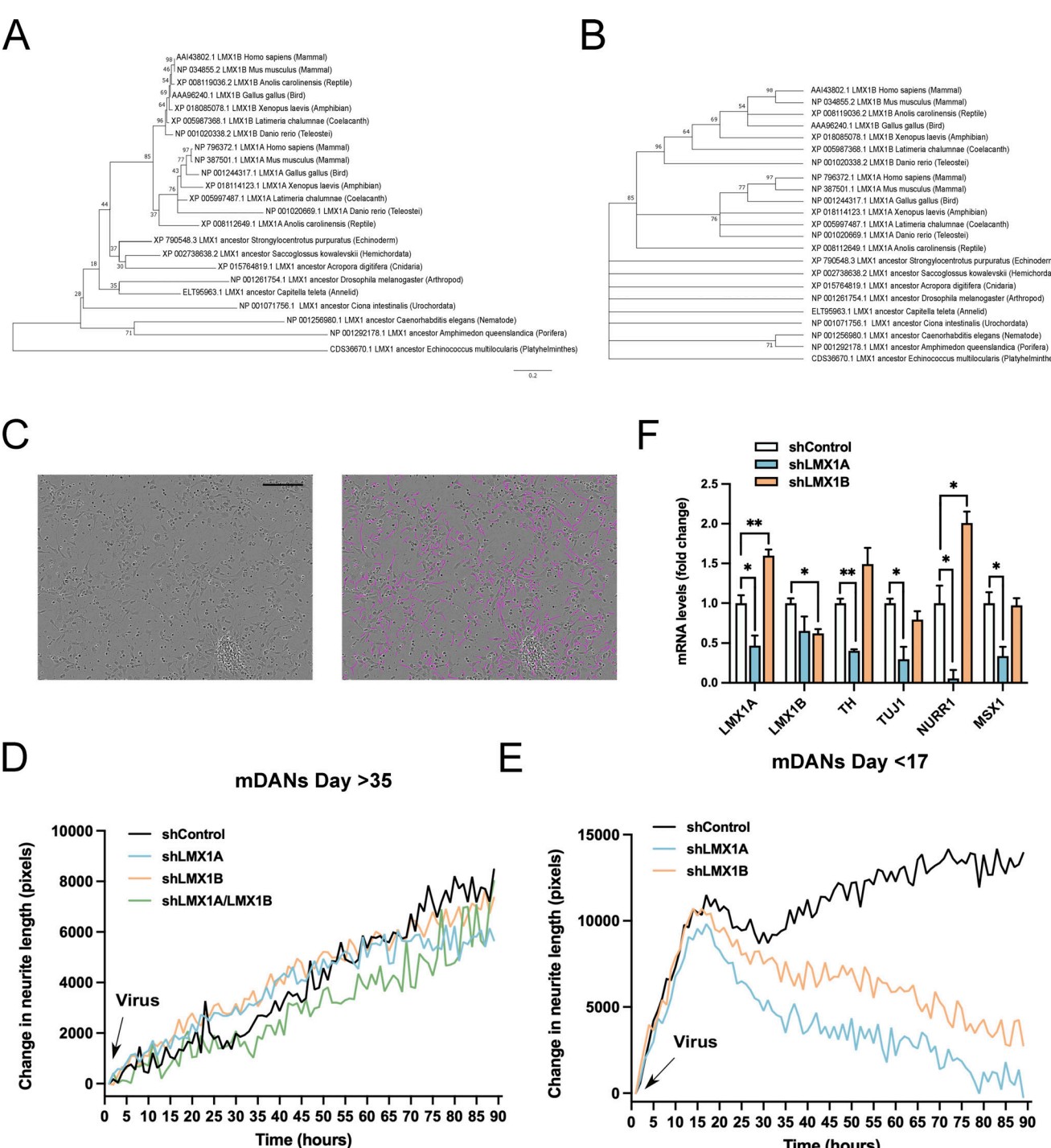

Figure S1. **LMX1A and LMX1B are evolutionary conserved human mDAN transcription factors required for early neurogenesis. (A)** Phylogenetic analysis by the Maximum Likelihood method based on the JTT matrix-based model. Initial tree for the heuristic search was obtained automatically by applying Neighbor-Join and BioNJ algorithms to a matrix of pairwise distances estimated using a JTT model, and then selecting the topology with superior log likelihood value. A discrete Gamma distribution was used to model evolutionary rate differences among sites (five categories [+G, parameter = 0.6637]). The analysis involved 23 amino acid sequences. All positions containing gaps and missing data were eliminated. There were a total of 186 positions in the final dataset. **(B)** Bootstrap consensus tree inferred from 2,000 replicates. Only branches that appeared in more than 50% bootstrap replicates are shown. The percentage of replicate trees in which the associated taxa clustered together in the bootstrap test are shown next to the branches. **(C–E)** Incucyte analysis of neurite length using automated tracking software. Bar = 100 μm. **(C)** iPSC-derived mDANs were imaged over 4 d following transduction with hsyn-GFP/mcherry-U6-shRNA lentiviruses as indicated at **(D)** day >35 and **(E)** day <17. mDANs were virally transduced on the day of plating. Shown are means from three wells of a single representative experiment. For clarity, error bars have been omitted. **(F)** qRT-PCR analysis of mRNA levels following shRNA suppression of LMX1A or LMX1B in young (day 17) cultures. Significant reductions in TH, TUJ1, NURR1, and MSX1 expression following LMX1A shRNA. LMX1B suppression increases LMX1A levels. mRNA levels normalized to shControl. Data show means ± SE of triplicate wells from a single representative experiment. Student's *t* test: *P < 0.05 and **P < 0.01.

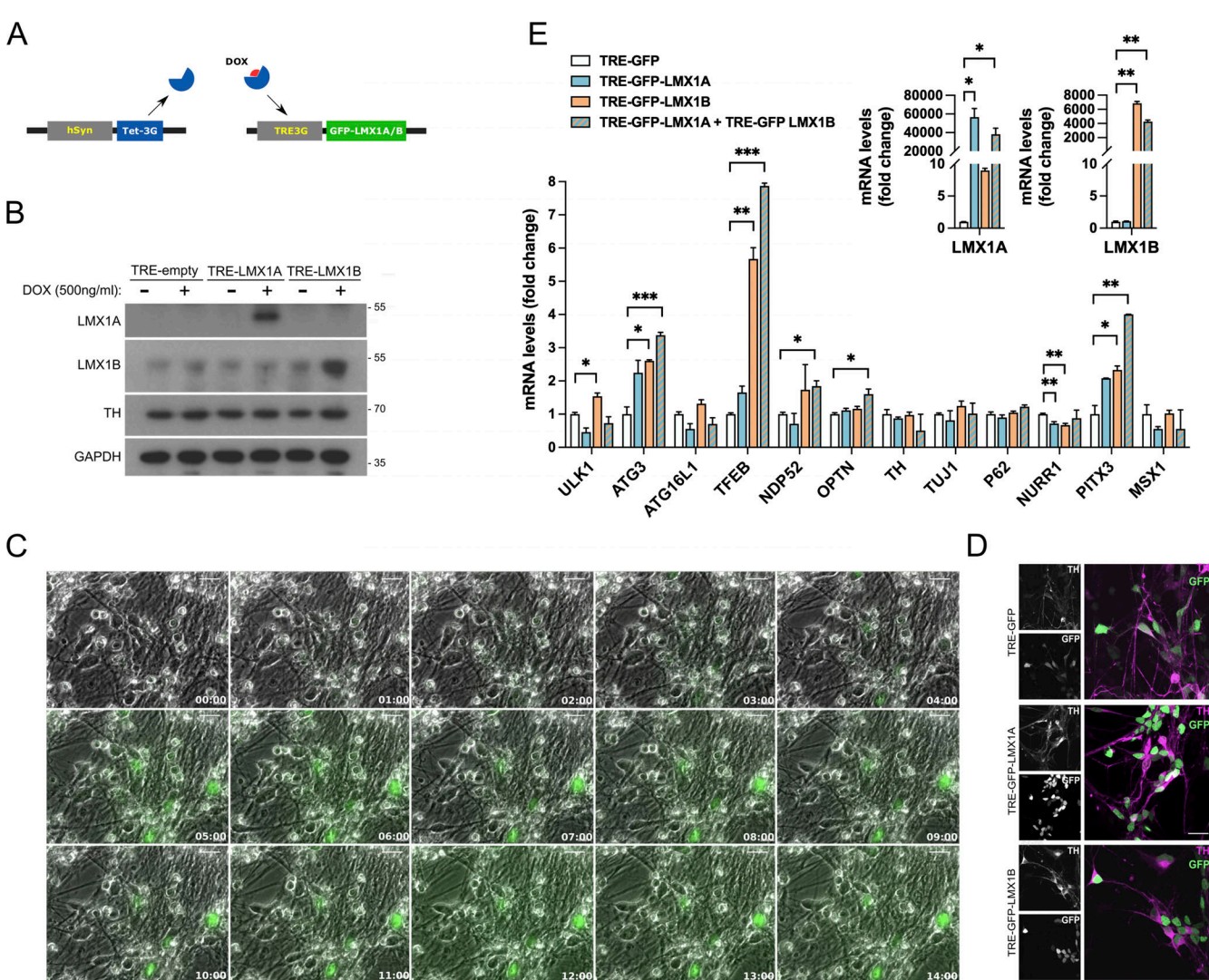

Figure S2. **LMX1A/LMX1B overexpression drives autophagy transcription in human mDANs. (A)** Schematic representation of the hsyn promoter modified Tet-On system. **(B)** Immunoblot showing TRE-LMX1A and/or TRE-LMX1B overexpression via the Tet-On system in iPSC-derived mDANs. Expression was induced with the addition of DOX (500 ng/ml) for 2 d. Molecular weight markers are shown in kD. **(C)** Wide-field time-lapse imaging of GFP-LMX1B overexpression via the Tet-On system in iPSC-derived mDANs. Expression was induced with the addition of DOX (500 ng/ml). GFP fluorescence is shown overlayed on phase contrast images. Bar = 10 µm. **(D)** Immunofluorescence labeling on iPSC-derived mDANs transduced with TRE-GFP, TRE-GFP-LMX1A, or TRE-GFP-LMX1B controlled by the Tet-ON system. Expression was induced with DOX (500 ng/ml) for 24 h. Cells were labeled with an anti-TH antibody (magenta). Bar = 20 µm. **(E)** mRNA levels of the indicated genes were determined by qRT-PCR and normalized by GAPDH levels. mRNA levels normalized to TRE-GFP. Data show means ± SD of triplicate wells from a single representative experiment. Student's *t* test: *P < 0.05, **P < 0.01, and ***P < 0.001 vs. TRE-GFP control.

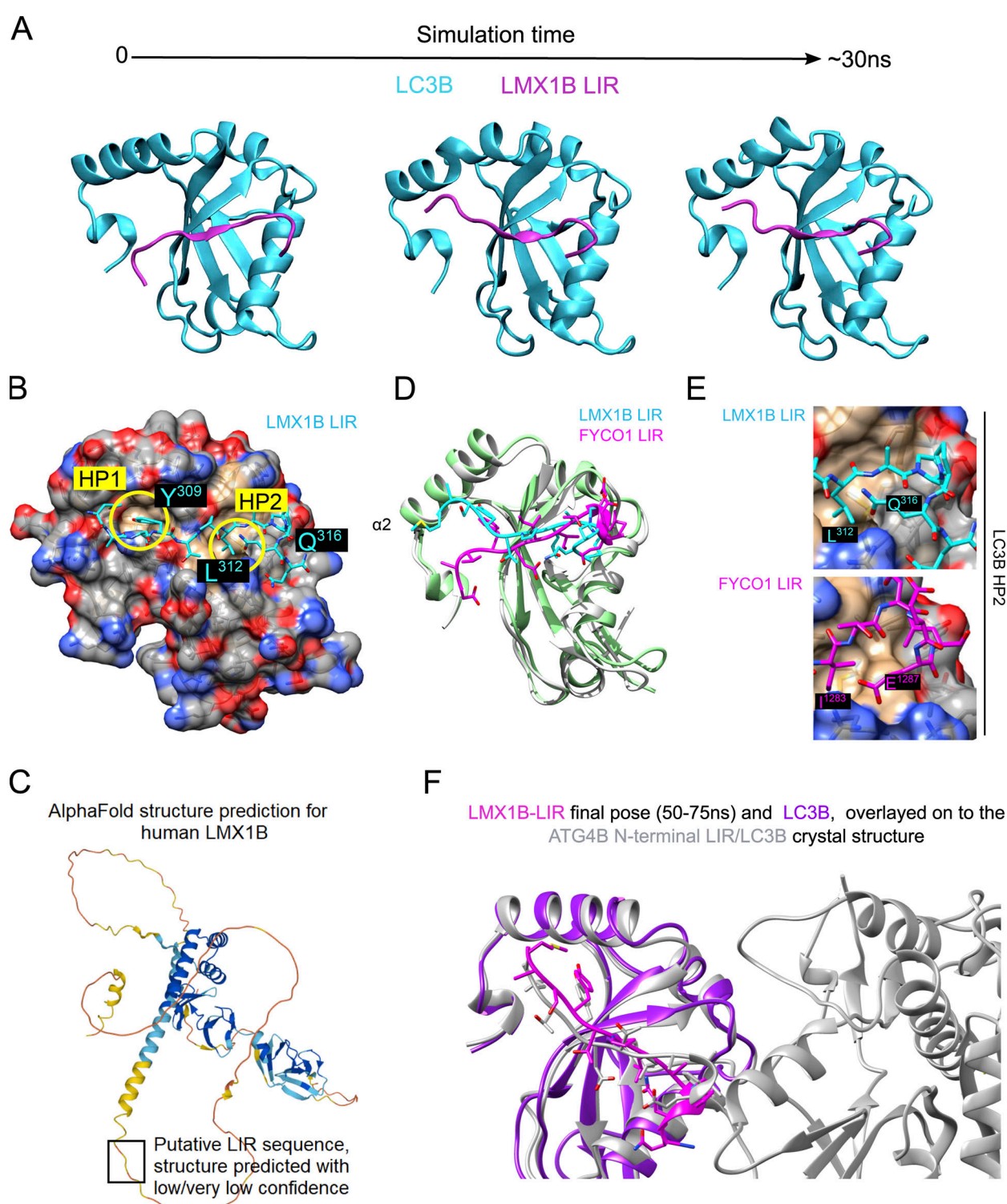

Figure S3. **In silico modeling of the LMX1B–LC3B interaction. (A)** Frames from a computer simulation of the LIR-dependent LMX1B-LC3B interaction (see Video 1) showing the dynamics of the putative LMX1B LIR motif docked by molecular replacement in the position adopted by the FYCO1 LIR (5d94.pdb). LC3B in cyan; LMX1B LIR in magenta. **(B)** LMX1B LIR (cyan) overlaid upon a space-filling model of LC3B (5d94.pdb) in final simulation pose, showing docking of the key LIR residues at $P_0$ and $P_3$ within hydrophobic pockets (HP) 1 and 2 of LC3B, respectively. **(C)** Current Alphafold prediction of the LMX1B structure showing the location of the LIR domain within a region of low/very low confidence for structure. Inherently unstructured regions in proteins requiring a cognate ligand to induce/stabilize folds will continue to pose a problem for protein structure prediction. **(D)** Ribbon structure of human LC3B (gray) in complex with the FYCO1 LIR (magenta; 5d94.pdb) overlayed with the final model simulation pose of LC3B (green) and LMX1B LIR (cyan). Close alignment between FYCO1 and LMX1B is seen within the core LIR binding region. **(E)** Side-by-side comparison of the LMX1B (left) and FYCO1 (right) LIRs docked at HP2 of LC3B (5d94.pdb) to show how LMX1B Q316 and FYCO1 E1287 fold back toward HP2 in both structures to stabilize LIR binding. **(F)** Comparison of the final LMX1B LIR pose and the position of the N-terminal ATG4B LIR identified in the ATG4B/lC3B crystal lattice.

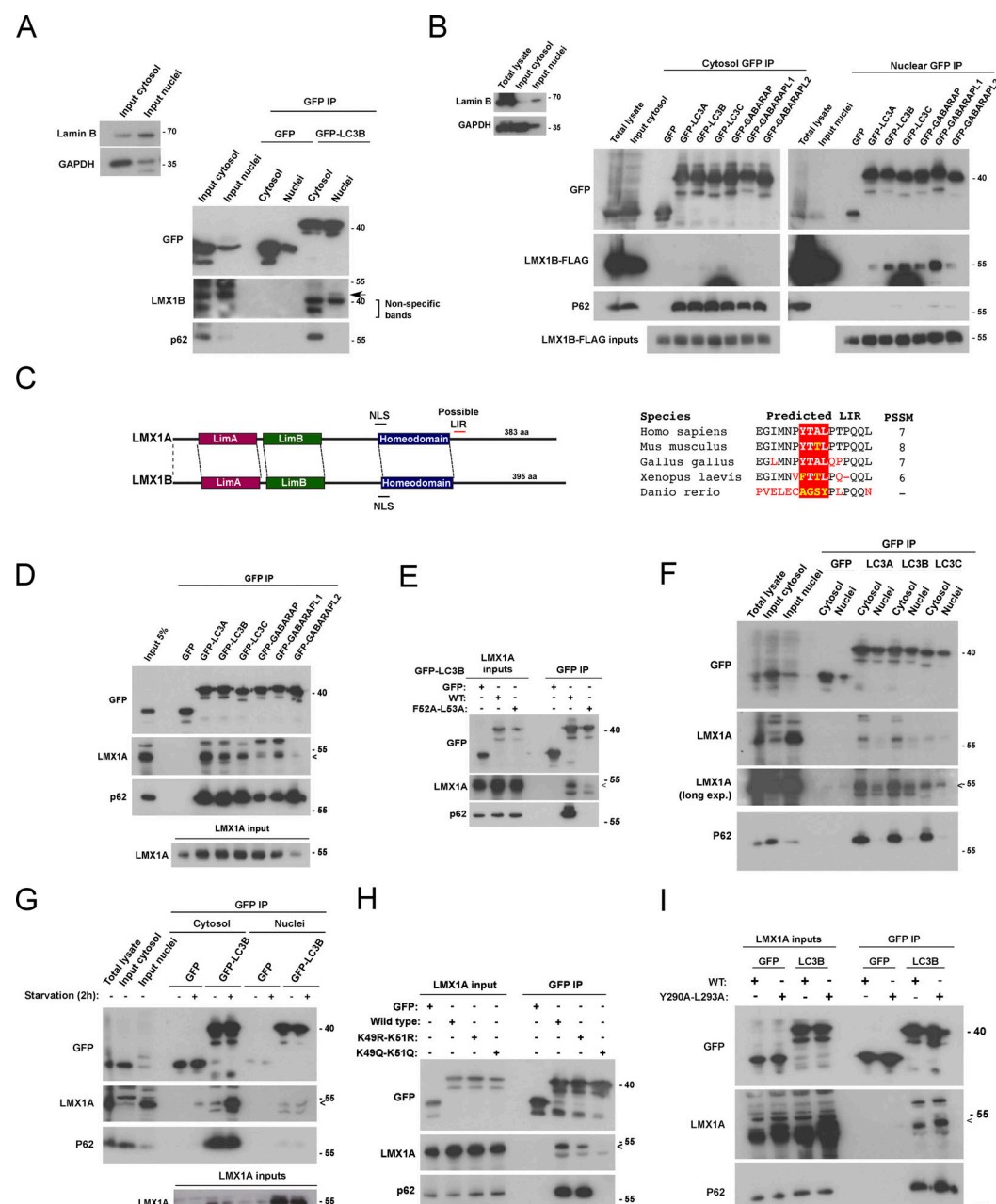

**Figure S4. Distinct binding properties of LMX1A and LMX1B for ATG8 proteins in cytosol and nucleus. (A)** GFP-TRAP IP of nuclear and cytosolic fractions from lysates of HEK293T cells expressing GFP-LC3B immunoblotted with anti-LMX1B antiserum. A faint band at the correct size for native LMX1B (arrow) is detected only in the nuclear fraction for the GFP-LC3B pull-down. **(B)** HEK293T cells expressing GFP-ATG8 proteins and LMX1B-FLAG were separated into nuclear and cytoplasmic fractions which were subjected to GFP-TRAP. Interactions with ATG8 family members were detected only in the nucleus. Immunoblotting for p62 is included as a positive control for binding. Fractionation is demonstrated to the left using Lamin B and GAPDH as markers for nuclei and cytosol, respectively. **(C)** Domain schematic of human LMX1A and LMX1B showing the position of a possible LIR motif in LMX1A (left). Alignment of the putative LMX1A LIR in different species is shown (right). **(D)** GFP-TRAP co-precipitation of LMX1A with GFP-ATG8 family members in HEK293T cells. 5% protein lysate from equivalent GFP-expressing cells is shown as "input." Immunoblotting for p62 is included as a positive control for binding. Arrow indicates the position of the LMX1A band. **(E)** GFP-TRAP co-precipitation of LMX1A with wild-type of LIR docking mutant (F52A/L53A) GFP-LC3B in HEK293T cells. 5% of protein lysates were used as control for protein expression (inputs). Immunoblotting for p62 is included as a positive control for binding. Immunoblotting for p62 is included as a positive control for binding. Arrow indicates position of the LMX1A band. **(F)** GFP-TRAP IP of nuclear and cytosolic fractions from lysates of HEK293T cells co-expressing GFP-LC3A, B, C and LMX1A under basal conditions. Immunoblotting for p62 is included as a positive control for binding. Arrow indicates position of the LMX1A band. **(G)** GFP-TRAP IP of nuclear and cytosolic fractions from lysates of HEK293T cells co-expressing GFP-LC3B and LMX1A. Comparisons of pull-downs in full nutrients following 2 h starvation. Immunoblotting for p62 is included as a positive control for binding. Arrow indicates position of the LMX1A band. **(H)** GFP-TRAP IP of LMX1A in HEK293T cells co-expressing wild-type, acetylation-deficient (K49R/K51R), and acetylation mimic (K49Q/K51Q) GFP-LC3B. Immunoblotting for p62 is included as a positive control for binding. Arrow indicates position of the LMX1A band. **(I)** GFP-TRAP pull-downs in lysates of HEK293T cells expressing LIR mutant (Y290A/L293A) LMX1A and GFP-LC3B. Immunoblotting for p62 is included as a positive control for binding. Arrow indicates position of the LMX1A band. **(A, B, D–I)** Molecular weight markers are shown in kD. PSSM, Position Specific Scoring Matrix.

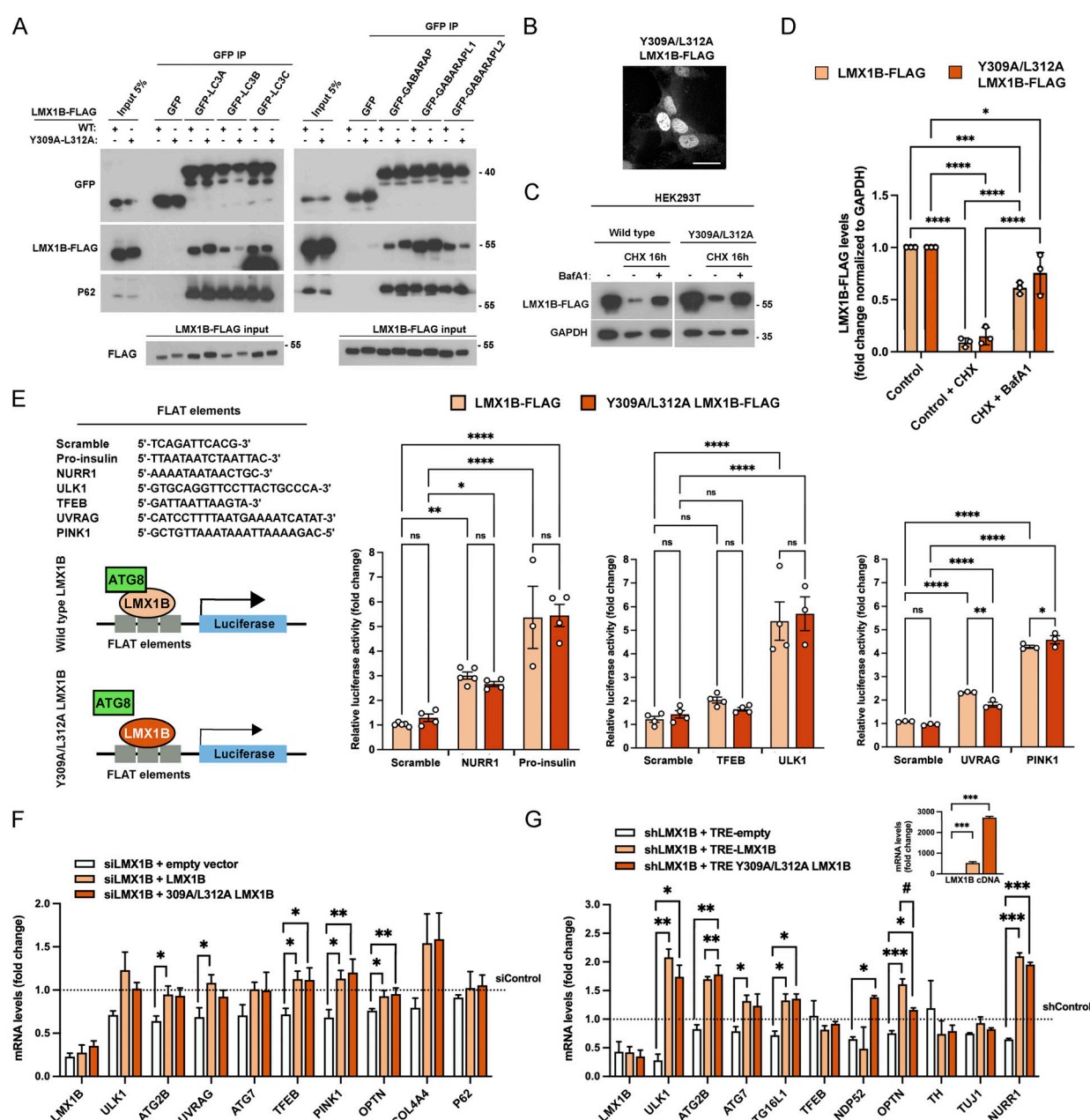

Figure S5. **Properties of the LMX1B Y309A/L312A mutant. (A)** GFP-trap immunoprecipitates of wild-type and LIR mutant (Y309A/L312A) LMX1B-FLAG in lysates of HEK293T co-expressing GFP-ATG8 family members. 5% of protein lysate from equivalent GFP-expressing cells is shown as a representative of "input." Immunoblotting for p62 is included as a positive control for binding. Molecular weight markers are shown in kD. **(B)** Immunofluorescence imaging of Y309A/L312A LMX1B-FLAG in HEK293T cells. Bar = 20 μm. **(C and D)** LMX1B turnover in HEK293T expressing wild-type or Y309A/L312A LMX1B-FLAG. Cells were treated with CHX (50 μg/ml; 16 h) in the absence or presence BafA1 (20 nM). Representative blot (C) and quantitation relative to GAPDH (D). Molecular weight markers are shown in kD. Mean ± SD ($n$ = 3); one-way ANOVA followed by Tukey's multiple comparison post-hoc test: *P < 0.05; ***P < 0.001, ****P < 0.0001. **(E)** Y309A/L312A LMX1B stimulates FLAT-luciferase activity to wild-type levels. To the left, luciferase FLAT element sequences (top), and schematic of the proposed role of ATG8 proteins as LMX1B co-factors (bottom). To the right, quantitation of expression driven by wild-type and Y309A/L312A LMX1B in HEK293T cells. Levels are presented normalized to empty vector control (pcDNA 3.1). Mean ± SEM ($n$ = 3–5); one-way ANOVA followed by Tukey's multiple comparison post-hoc test comparing scrambled sequence against NURR1, pro-insulin, TFEB, ULK1, UVRAG, and PINK1: *P < 0.05, **P < 0.01, ****P < 0.0001. **(F)** Y309A/L312A LMX1B rescues autophagy gene transcription in HEK293T cells. HEK293T cells were double co-transfected with 50 nM smartpool LMX1B siRNA and codon optimized, siRNA-resistant wild-type, Y309A/L312A LMX1B (or empty vector control). mRNA levels were normalized siControl + empty vector. Mean ± SEM ($n$ = 3); one-way ANOVA followed by Dunnett's multiple comparison post-hoc test comparing the three different LMX1B constructs to empty vector (*P < 0.05; **P < 0.01) and a Student's $t$ test to compare Y309A/L312A with wild-type LMX1B (no data sets statistically significant). (Note: the data sets for "empty vector" and "siLMX1B + LMX1B" are identical to those shown in Fig. 7 A.) **(G)** Y309A/L312A LMX1B rescues autophagy gene transcription in iPSC-derived human mDANs. Mature (day ~50) mDANs were transduced with hsyn-GFP-U6-shLMX1B and LMX1B levels were rescued with codon optimized, shRNA-resistant wild-type or Y309A/L312A TRE-LMX1B (or empty vector as a control). mRNA levels were normalized shControl + TRE-empty. Mean ± SEM ($n$ = 3 wells from a single neuralization); one-way ANOVA followed by Dunnett's multiple comparison post-hoc test comparing the two LMX1B constructs to empty vector (*P < 0.05, **P < 0.01, ***P < 0.001) and a Student's $t$ test to compare Y309A/L312A with wild-type LMX1B (*/#P < 0.05).

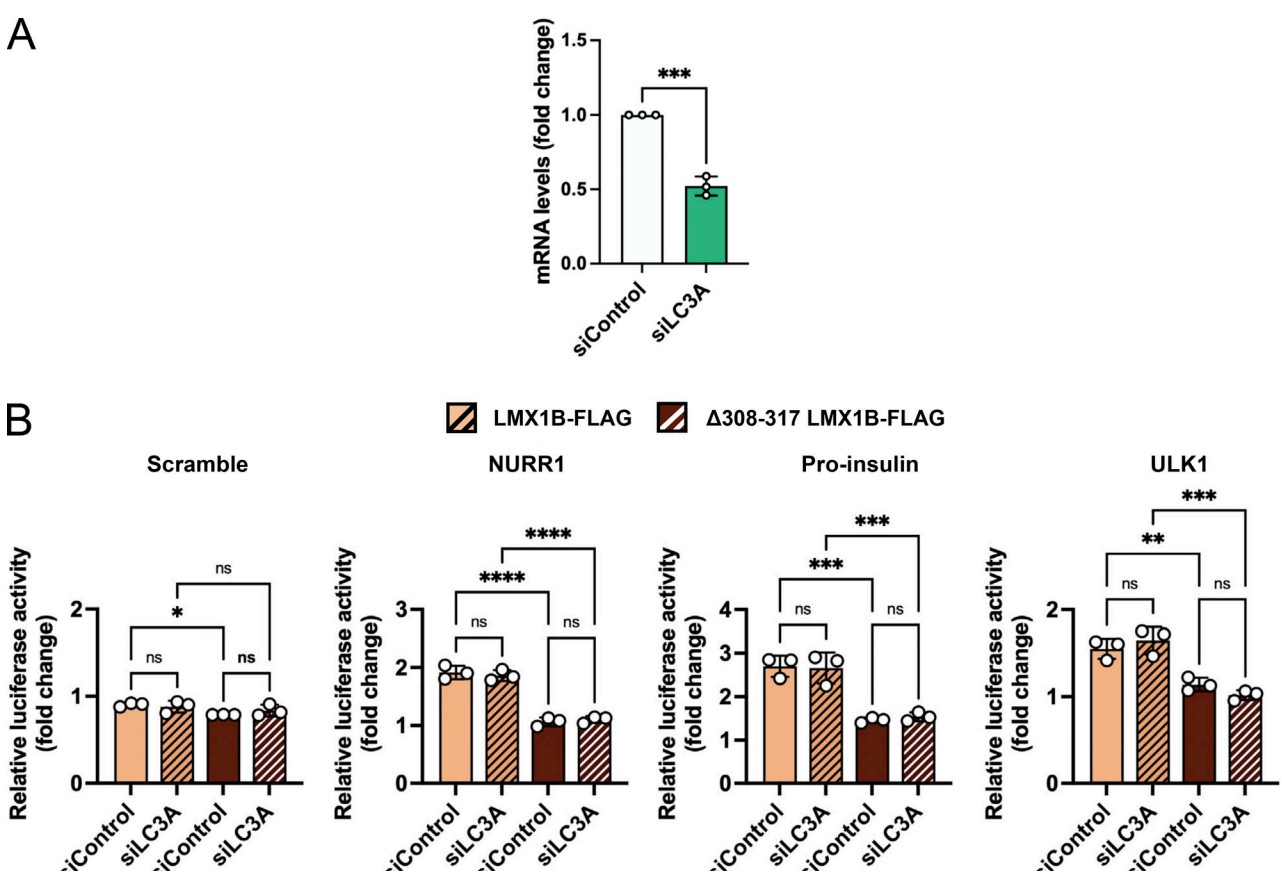

Figure S6. **LC3A is not a cofactor for LMX1B transcriptional control of selected genes. (A)** qRT-PCR measurements of LC3A mRNA levels following siRNA depletion. mRNA levels normalized to GAPDH. Mean ± SD ($n$ = 3); Student's $t$ test: ***P < 0.001 vs. siControl. **(B)** Effect of LC3A suppression on LMX1B-driven FLAT luciferase reporter expression. Scramble, NURR1, pro-insulin, and ULK1 FLAT promoter luciferase assay (LightSwitch) driven by wild-type or Δ308-317 LMX1B-FLAG in HEK293T cells siRNA depleted for siLC3A. Levels are normalized to empty vector control (pcDNA 3.1). Mean ± SD ($n$ = 3); one-way ANOVA followed by Fishers LSD test for planned comparisons: *P < 0.05, **P < 0.01, ***P < 0.001, ****P < 0.0001.

Video 1. **Computer simulation of the interaction between the LMX1B peptide and LC3B.** Movie of an example computer simulation of the LMX1B–LC3B interaction (relates to Fig. S3 A). The LMX1B peptide motif was docked by molecular replacement in the position adopted by the FYCO1 LIR (5d94.pdb). LC3B is shown in cyan; LMX1B LIR in magenta. The movie spans 50 ns.

**Provided online is Table S1, which shows MatInspector promoter FLAT sequence analysis for selected candidate gene.**

