## [Peer Review File · The Journal of Cell Biology]

ATG8-dependent LMX1B-autophagy crosstalk shapes human midbrain dopaminergic neuronal resilience

Natalia Jimenez-Moreno, Madhu Kollareddy, Petros Stathakos, Joanna Moss, Zurine Anton, Deborah Shoemark, Richard Sessions, Ralph Witzgall, Maeve Caldwell, and Jon Lane

Corresponding Author(s): Jon Lane, University of Bristol

Review Timeline:

Submission Date:	2019-10-18
Editorial Decision:	2020-01-06
Revision Received:	2022-11-14
Editorial Decision:	2023-01-20
Revision Received:	2023-02-03
Accepted:	2023-02-21

Monitoring Editor: Richard Youle

Scientific Editor: Tim Spencer

Transaction Report:

DOI: <https://doi.org/10.1083/jcb.201910133>

December 16, 2019

Re: JCB manuscript #201910133

Dr. Jon D Lane
University of Bristol
School of Biochemistry School of Medical Sciences University Walk
Bristol BS8 1TD
United Kingdom

Dear Dr. Lane,

Thank you for submitting your manuscript entitled "LIR-dependent LMX1B-autophagy crosstalk shapes human midbrain dopaminergic neuronal resilience". The manuscript has been evaluated by expert reviewers, whose reports are appended below. Unfortunately, after an assessment of the reviewer feedback, our editorial decision is against publication in JCB.

You will see that there is a substantial amount of interest in the claim that LC3 affects LMX1B transcriptional activity and the role this plays in the regulation of autophagy but the reviewers suggest focusing more precisely on this aspect of the paper and a significant amount of additional experimental work necessary for strengthening this claim. We feel that the majority of the comments need to be substantively addressed but the Fundc1 recommendation of Rev. #2 beyond the scope and not needed.

Although your manuscript is intriguing, I feel that the points raised by the reviewers are more substantial than can be addressed in a typical revision period. If you wish to expedite publication of the current data, it may be best to pursue publication at another journal.

Given interest in the topic, I would be open to resubmission to JCB of a significantly revised and extended manuscript that fully addresses the reviewers' concerns and is subject to further peer-review. If you would like to resubmit this work to JCB, please contact the journal office to discuss an appeal of this decision or you may submit an appeal directly through our manuscript submission system. ***As a significant amount of work would be necessary for this study to be considered for re-review, we strongly recommend you contact us at an early stage with a detailed revision plan for feedback to avoid spending time on experimental revisions that may not be appropriate.*** Please note that priority and novelty would be reassessed at resubmission.

Regardless of how you choose to proceed, we hope that the comments below will prove constructive as your work progresses. We would be happy to discuss the reviewer comments further once you've had a chance to consider the points raised in this letter. You can contact the journal office with any questions, cellbio@rockefeller.edu or call (212) 327-8588.

Thank you for thinking of JCB as an appropriate place to publish your work.

Sincerely,

Richard Youle, Ph.D.
Monitoring Editor

Marie Anne O'Donnell, Ph.D.
Scientific Editor

Journal of Cell Biology

Reviewer #1 (Comments to the Authors (Required)):

This paper from Jiménez-Moreno and coworkers present data suggesting that the LIM homeodomain transcription factors LMX1A and LMX1B, which are essential mediators of midbrain dopaminergic neuronal (mDAN) differentiation and survival, bind to ATG8 family proteins in a LIR-dependent manner and that the LMX1B-ATG8 family interactions is important for transcriptional regulation of several autophagy genes. SiRNA or shRNA-mediated reduction of LMX1A and -B expression reduces basal autophagy. LMX1B interacts with LC3B in the nucleus under basal conditions and to cytosolic LC3B upon starvation, and is degraded by autophagy during starvation.

This is a very interesting story with the novel aspect of LIR-dependent regulation of transcription of autophagy genes upon

recruitment of nuclear ATG8 proteins to LMX1B. Also, LMX1B is degraded both by the proteasome (presumably in the nucleus) and by autophagy in the cytosol. Although, there is not always a "streamlined" consistency in the results when the effects on multiple genes are monitored and compared based on the different assays/experimental strategies used, the picture is still clear and reflects complicated regulations where such variation may be expected.

Specific comments:

1. Not a dramatic effect on the shRNA- or siRNA-mediated knock down (KD) experiments neither on mRNA nor protein levels and not entirely consistent for all tested genes. However, the KD of LMX1B as such is not very efficient which may explain this, at least partly. We see in Fig S5 that LMXA1 is actually expressed in the HEK293 cells. Hence, did the authors try to KD both LMX1B and LMX1A to look for redundancy in the regulation?
2. Fig. 1F: It is hard to evaluate the images and know something about how well the antibody used work in IF. Can the authors comment on this?
3. Fig 1H: What is siGL2 serving as the siControl here?
4. Fig 4B: Since degradation of LMX1B during starvation is very important the authors need to add data on the effect of proteasomal inhibition (using MG132) under starvation conditions. In short, MG132 treatment is missing Fig 4B.
5. Fig 4C: There is so little LMX1B protein in the cytosol that it becomes very unreliable to interpret the data. It would be nice if fractionation data for the 6 hr starvation experiments were shown too.
6. Fig 4D: Here only overexpressed GFP-LC3B is shown. The authors should co-stain to check co-localization of endogenous LC3B with the cytoplasmic LMXB1-FLAG following 0, 2hrs and 6 hrs starvation.
7. The intranuclear distribution of LMXB1-FLAG as revealed by IF suggest a preferential localization at the nuclear lamina. Is this known from before? It is not seen in the HEK293 cells stained in Fig 1F. Is this an artefact of overexpression? As the FLAG antibody may recognize some cellular proteins the authors need to show IF staining with anti-FLAG antibodies of cells not expressing LMXB1-FLAG.
8. The experiment in Fig 5K is very nice. Can the authors IP LMXB1-FLAG with endogenous ATG8s from the nuclear fraction? The experiment in Fig 5K would be nice to see with endogenous LC3B or GABARAP.
9. The data in Fig 6 largely shows little effects of mutating the aromatic Y and hydrophobic L residues of the putative LIR motif of LMXB1 and should be shown as supplemental data or alternatively deleted from the paper.
10. In Fig. 7 the deletion mutation delta308-317 removing the core LIR and some flanking C-terminal residues has a much more detrimental effect on ATG8 binding. Inconsistent with a role for the putative LIR in lysosomal degradation of LMXB1 BafA1 treatment stabilized the LIR mutant just as well as WT (Fig 7C). It is interesting that the LIR here is not involved in the degradation in the lysosome but in the regulation of transcription.

Reviewer #2 (Comments to the Authors (Required)):

The study by Jiménez-Moreno and co-authors found that LMX1A and LMX1B, two genes important for midbrain dopaminergic neuronal (mDAN) differentiation and survival (also associated with Parkinson development), can interact with ATG8 family members, can regulate the expression of critical autophagy genes and can protect mDANs from mitochondrial stress. Notably, they provide evidence that ATG8s play a central role acting as cofactors for LMX1B and influencing autophagy gene expression through the activity of this transcriptional factor. This story would be of great interest and generally the data are well shown and of good quality; however, there are some issues described below, which should be addressed to improve the manuscript:

- Authors described that "LMX1A-B suppression also elevated mitochondrial ROS levels and negatively impacted oxidative phosphorylation, causing reduced basal oxygen consumption rates, maximal respiration"; on the other hand, LMX1A/LMX1B over-expression protected mDNA from rotenone-induced cell death. I think that this is a critical aspect: how do LMX1A-B protect from Rotenone? Are they able to promote mitophagy? LMX1B has a LIR domain and the authors observed that upon 6h of STV both LMX1 translocated from nucleus to cytoplasm; does it play a role as a cargo receptor such as FUNDC1?
- They used GFP-LC3: it is not clear if they handled a GFP-LC3 stable cell line or not; as reported in Guidelines for autophagy 2016, "high levels of over-expressed GFP-LC3 can result in its nuclear localisation": I thus recommend to repeat the IP using endogenous LC3B.
- Based on Fig 6F (knockdown/rescue experiment), Y309A/L312A LMX1B has an effect comparable to that of wt LMX1B on autophagy gene expression, while Δ 308-317 LMX1B failed to rescue some autophagy targets, indicating that the LIR domain of LMX1B may be fundamental for some targets but not for others; could the authors comment this aspect? Furthermore, they analysed the effect of SiLC3B, but I think that they should repeat the experiment by down-regulating the other ATG8 proteins (LMX1B highly interacts with GABARAP and GABARAP-L1 since these could have a stronger effect on the regulation of autophagy gene expression.

Minor points:

- Please improve the quality of some WBs (for example, in Figure 4C). Based on that image, the quantification would almost be impossible

Reviewer #3 (Comments to the Authors (Required)):

Jimenez-Moreno et al. investigated the regulation of autophagy by the transcription factors LMX1A and 1B, which participate to midbrain dopaminergic neuronal differentiation and survival. The role of these TFs in autophagy regulation was previously demonstrated, thus some of the findings are not entirely novel. In the first part of the manuscript the authors demonstrated that LMX1B and A binds to the promoters of several autophagy genes, and that the expression levels of several of them is reduced in cells depleted for LMX1B. The authors also show that silencing of LMX1B reduced LC3 and WIPI2 puncta, although data are not entirely convincing and not supported by a quantitative analysis of autophagy flux (e.g. substrate degradation rate). Thus it is unclear the relevance of LMX1 factors in autophagy.

The authors also demonstrated that during starvation LMX1 factors translocate from nucleus to cytoplasm, although the kinetics is not entirely clear to me, since biochemical evidences are not entirely in line with immunofluorescent analysis, particularly during acute starvation (2h). Also in this case the functional relevance of this shuttling in the context of autophagy regulation can only be speculated.

Lastly, the authors proposed that ATG8 binding stimulates transcription of LMX1 proteins. This is potentially a very novel and original concept. However, this claim is based on very few experimental evidences with results showing at best very marginal differences. Furthermore, functional impact on autophagy flux was not investigated.

In conclusion I believe that this work is not suitable for publication in JCB.

In addition, below I list my technical concerns:

- Fig 1A lacks error bars and statistics.
- Fig 1G: the co-staining with LMX1B is needed to quantify the number of LC3 puncta in silenced vs non-silenced cells. Currently, the difference are not clear.
- ChIP in fig 2 lack error bars and statistics.
- Data in fig 3A are not convincing
- Fig 4, a time course experiment of nuclear/cytosolic localization should be performed both by IF and WB. Figure 5K shows a different result.
- Immunoprecipitation experiments using nuclear and cytosolic fractions should be supported by appropriate loading control of subcellular markers.
- Fig 6A, the input of LMX1B have loading problems.

GENERAL SUMMARY: We are delighted that the reviewers found plenty of interesting aspects to our work, and in particular, recognised its novelty and significance, and are very grateful for their input and for the opportunity to revise our manuscript. We also highlight, as is only too obvious, the extended time-period between these submissions. As the journal office is aware, a combination of the global pandemic (whose timing could not have been any worse for this project), the need to acquire additional grant funding and recruit a researcher focused on the revision (MK), and the substantial amount of extra data that we have produced has contributed to this. We nevertheless are confident that the referees will still find this work of impact and value to the field.

SUMMARY OF THE UPDATED/ADDITIONAL DATA IN THE REVISED MANUSCRIPT: In the following, we summarise the updates to the revised manuscript, including entirely new figures, new figure parts, and any data that have been moved to the Supplemental section to accommodate these improvements. The headline updates are as follows –

1. The generation and characterisation of 2 LMX1B CRISPR knockout HEK293T lines including qRT-PCR data for autophagy target gene expression (**Fig. 1G** *new data*), and autophagy flux analysis (**Fig. 1H, I** *new data*).
2. Experiments in Figure 4 showing turnover of LMX1A (**Fig. 4A** *new data*), the effect of MG132 on LMX1B turnover during starvation (**Fig. 4C** *new data*), improved fractionation data for LMX1B localisation (**Fig. 4D** *new data*), and FLIP analysis of LMX1B nucleus-cytoplasmic transport (**Fig. 4F** *new data*).
3. New data in Figure 5 showing immunoprecipitation of native LC3B using antibodies against endogenous LMX1B (**Fig. 5E** *new data*); new data in Fig. S2 showing interaction between native LMX1B and GFP-LC3B in the nucleus (**Fig. S4A** *new data*).
4. New data in an expanded Figure 6 using the Δ 308-317 LMX1B (LIR mutant), including an extended set of FLAT-luciferase reporter constructs (pro-insulin, NURR1, UVRAG, PINK1, ULK1, TFEB) (**Fig. 6E** *new data*), the same expanded set of reporters in an LC3B knockdown background (**Fig. 6F** *new data*), and the same expanded set in the context of GABARAP-L1 suppression (**Fig. 6G** *new data*).
5. A new Figure 7 showing target gene expression in an siRNA knockdown/rescue setting using the Δ 308-317 LMX1B construct in HEK293 cells (**Fig. 7A** – previously shown in Fig. 7E of the original submission), and a parallel shRNA knockdown/rescue experiment in iPSC-derived mDANs (**Fig. 7B** *new data*). Also included are rescue data for autophagosome formation in starved HEK293T LMX1B KO cells (**Fig. 7C** *new data*), and data on rotenone protection in LMX1B KO HEK293Ts rescued with wild type or Δ 308-317 LMX1B (**Fig. 7E** *new data*) and in mDANs expressing wild type LMX1B, Y309A/L312A LMX1B and Δ 308-317 LMX1B (**Fig. 7F** *new data*).
6. A new supplementary Figure S5 that contains the previous data from Figure 6 of the original submission addressing ATG8 binding, localisation and stability of the Y309A/L312A mutant LMX1B (**Fig. S5A-D**) alongside new data using the expanded FLAT-luciferase reporter expression tool set (**Fig. S5E** *new data*) and knockdown/rescue gene expression data in HEK293T and iPSC-derived mDANs (**Fig. S5F-G** – previously shown in Fig. 6F, 7E of the original submission).
7. A new supplementary Figure S6 showing the effect of LC3A siRNA suppression on FLAT-luciferase reporter assays for a limited set of constructs (**Fig. S6** *new data*).

RESPONSES TO THE REVIEWERS: Below we present the reviewers' original comments (italics) with our responses in standard text (blue font).

Reviewer #1 (Comments to the Authors (Required)):

This paper from Jiménez-Moreno and coworkers present data suggesting that the LIM homeodomain transcription factors LMX1A and LMX1B, which are essential mediators of midbrain dopaminergic neuronal (mDAN) differentiation and survival, bind to ATG8 family proteins in a LIR-dependent manner and that the LMX1B-ATG8 family interactions is important for transcriptional

regulation of several autophagy genes. siRNA or shRNA-mediated reduction of LMX1A and -B expression reduces basal autophagy. LMX1B interacts with LC3B in the nucleus under basal conditions and to cytosolic LC3B upon starvation, and is degraded by autophagy during starvation.

This is a very interesting story with the novel aspect of LIR-dependent regulation of transcription of autophagy genes upon recruitment of nuclear ATG8 proteins to LMX1B. Also, LMX1B is degraded both by the proteasome (presumably in the nucleus) and by autophagy in the cytosol. Although, there is not always a "streamlined" consistency in the results when the effects on multiple genes are monitored and compared based on the different assays/experimental strategies used, the picture is still clear and reflects complicated regulations where such variation may be expected.

We are very grateful for the reviewer's pragmatic and positive outlook on our work. We of course agree that the overall story is very interesting and agree with the reviewer that the picture is clear, despite the complexity of the pathway meaning that results from certain orthogonal approaches show natural variability. As the reviewer will see, we have included substantial additional data (summarised above; detailed in the point-to-point response below) which in our view pulls the data together into a much more cohesive and robust story.

Specific comments:

1. Not a dramatic effect on the shRNA- or siRNA-mediated knock down (KD) experiments neither on mRNA nor protein levels and not entirely consistent for all tested genes. However, the KD of LMX1B as such is not very efficient which may explain this, at least partly. We see in Fig S5 that LMXA1 is actually expressed in the HEK293 cells. Hence, did the authors try to KD both LMX1B and LMX1A to look for redundancy in the regulation?

We routinely achieved siRNA/shRNA knockdown levels of suppression in the region of 50-75% reduced, depending on cell-type and knockdown methodology, which are reasonable suppression levels (particularly in hiPSC-derived mDANs). The accompanying reductions in LMX1B target protein expression may not approach these values in all cases, but this is to be expected given that a broad range of transcription factors contribute to autophagy gene expression (e.g., Di Malta et al., 2019 Front. Cell Dev. Biol. 7: 114; Füllgrabe et al., 2016 JCS 129: 3059-3066), and the likelihood of short-term adaptation.

In the new LMX1B CRISPR knockout HEK293T lines (see data presented in **Figs 1 & 7**), reduced expression of some target genes is perhaps clearer (e.g., ATG3, COL4A4, TFEB); however, due to adaptation/compensation observed during extended culture, these levels tended to recover towards baseline in LMX1B KO cell-lines over multiple passages (data not shown). We were not surprised by this, given the array of autophagy-associated transcription factors that are likely able to compensate to return transcript levels to baseline norms. Hence, in some respects the more acute, albeit less efficient acute knockdowns seen with siRNA/shRNA is preferable. On the topic of dual knockdown, the reviewer is mistaken in thinking that the original Fig. S5 showed LMX1A expression in HEK293T cells – these data, which are depicted in the revised version in **Fig. S4**, show only expression of exogenous, transfected LMX1A cDNA. HEK293T cells do not express detectable LMX1B mRNA (see **Reviewers' Fig. 1**) For this reason, we did not attempt this as dual LMX1A/LMX1B knockdown in HEK293T cells. Where knockdown of both targets may have been of value is in mDANs; however, we did not carry this out because the shRNA targeting of either gene on its own significantly reduced expression of the other in this setting, therefore effectively being a double knockdown (see **Fig. 2I**).

Reviewers' Fig. 1: qRT-PCR analysis of LMX1A and LMX1B transcript levels in HEK293T cells normalised against expression in human iPSC-derived mDANs. Student's t-test: ** $p < 0.01$; *** $p < 0.001$.

2. Fig. 1F: It is hard to evaluate the images and know something about how well the antibody used work in IF. Can the authors comment on this?

We thank the reviewer for pointing this out and can only think that the resolution of the reviewers' PDF may have been affected, as our original version looked ok. We will certainly discuss this issue more broadly with the journal production team should the manuscript progress to that stage. For this revision we have in any case chosen to remove these images and the accompanying quantitation as we felt they were unnecessary for the overall narrative.

3. Fig 1H: What is siGL2 serving as the siControl here?

We apologise to the reviewer for not explaining this – GL2 is a lab code for a control (luciferase targeting) siRNA that we routinely use. We have now amended the manuscript to use “siControl” instead, and detail in the Materials and Methods Resources Table.

4. Fig 4B: Since degradation of LMX1B during starvation is very important the authors need to add data on the effect of proteasomal inhibition (using MG132) under starvation conditions. In short, MG132 treatment is missing Fig 4B.

We are grateful to the reviewer for highlighting this. We have now included a full data set with MG132 +/- starvation in LMX1B-FLAG expressing HEK293T cells as Fig. 4C. The data show that both the lysosome and the proteasome contribute to the accelerated LMX1B degradation seen during nutrient starvation (with proteasome inhibition appearing to have a stronger effect).

5. Fig 4C: There is so little LMX1B protein in the cytosol that it becomes very unreliable to interpret the data. It would be nice if fractionation data for the 6 hr starvation experiments were shown too.

This is a very valid point. To address this, we carried out a series of additional repeat experiments based on the 2-hour starvation timepoint and present much “cleaner” data (Fig. 4D). The quantitation gave the same overall result of increased nuclear LMX1B during early phases of starvation, but the blot is now much nicer. We attempted to generate complimentary data for the 6-hour timepoint as suggested by the reviewer; however, this proved very challenging because HEK293T cells starved for this length of time were highly fragile, with many undergoing apoptosis and floating away, meaning that the cell fractionation stage became unreliable due loss of integrity of these stressed cells.

6. Fig 4D: Here only overexpressed GFP-LC3B is shown. The authors should co-stain to check co-localization of endogenous LC3B with the cytoplasmic LMXB1-FLAG following 0, 2hrs and 6 hrs starvation.

This is a reasonable suggestion; however, we have not been able to obtain reliable data for this due to different fixation requirements for the target proteins. Cytosolic LMX1B foci seemingly do not fix well in methanol, whereas good LC3B antibody staining requires this fixation method.

7. The intranuclear distribution of LMXB1-FLAG as revealed by IF suggest a preferential localization at the nuclear lamina. Is this known from before? It is not seen in the HEK293 cells

stained in Fig 1F. Is this an artefact of overexpression? As the FLAG antibody may recognize some cellular proteins the authors need to show IF staining with anti-FLAG antibodies of cells not expressing LMXB1-FLAG.

We agree that LMX1B-FLAG staining appears to be enriched at the margins of the nucleus under these conditions (we are not aware of previous reports of this); however, the same was apparently not true using antibodies against endogenous LMX1B (e.g., Fig. 1F in the original submission) or with GFP-tagged LMX1B in living cells (Fig. 4F in the revised version). We believe that this may be caused by fixation in formaldehyde and possibly restricted antibody accessibility. To show that this is not the result of the FLAG antibody cross-reacting with lamina, below we show example images of mixed populations of cells transiently expressing or not LMX1B-FLAG (wild type & NLS deleted for reference), showing that the antibody does not stain untransfected cells. (Reviewers' Fig. 2).

Reviewers' Fig. 2: anti-FLAG immunofluorescence staining of cells expressing (or not expressing) wild type or Δ NLS LMX1B-FLAG. High mag zooms are to the right.

8. The experiment in Fig 5K is very nice. Can the authors IP LMXB1-FLAG with endogenous ATG8s from the nuclear fraction? The experiment in Fig 5K would be nice to see with endogenous LC3B or GABARAP.

We thank the reviewer for their positive comments. We are pleased to say that we have been able to carry out successful IPs of endogenous LC3B pulled down by antibodies against native LMX1B (i.e., both proteins endogenous). This appears as Fig. 5E in the revised manuscript. We achieved this by cross-linking a really good batch of anti-LMX1B antiserum to magnetic beads. For practical reasons this was carried out using whole cell lysates; however, we also managed to IP a small fraction of endogenous LMX1B specifically in the nuclear fraction of cells expressing GFP-LC3B by GFP-TRAP (Fig. S4A in the revised manuscript). Although the second example is the other way around (reviewer suggested targeting endogenous ATG8s using LMA1B-FLAG), we hope that the 2 results together will be enough to convince the reviewer that the interaction is real.

9. The data in Fig 6 largely shows little effects of mutating the aromatic Y and hydrophobic L residues of the putative LIR motif of LMXB1 and should be shown as supplemental data or alternatively deleted from the paper.

We agree that the Y309A/L312A mutant showed little effect on binding across the broader family of ATG8s and on transcriptional control in rescue experiments. We initially included these data for completeness, but now that we have extended our investigations using the more effective Δ 308-317 LMX1B (LIR mutant), we agree that the data focused on the previous construct can now take a back seat. As suggested, we have made a new supplemental figure that contains most of the Y309A/L312A LMX1B data (new Fig. S5), including luciferase experiments and the knockdown/rescue experiments. In doing so, we have separated out the knockdown/rescue qRT-PCR data in HEK293T cells for Δ 308-317 LMX1B (Fig. 7E in the original submission), and these data appear as a new Fig. 7A accompanying complimentary data in mDANs (Fig. 7B in the new manuscript). The knockdown/rescue qRT-PCR data for Y309A/L312A LMX1B that previously appeared as Fig. 6F is now Fig. S5F, G in the revised manuscript. Finally, we have kept the Y309A/L312A construct cell death rescue experiment in iPSC mDANs together with the new Δ 308-317 LMX1B data as new Fig. 7F as an important reference point for completeness.

10. In Fig. 7 the deletion mutation delta308-317 removing the core LIR and some flanking C-terminal residues has a much more detrimental effect on ATG8 binding. Inconsistent with a role for the putative LIR in lysosomal degradation of LMXB1 BafA1 treatment stabilized the LIR mutant just as well as WT (Fig 7C). It is interesting that the LIR here is not involved in the degradation in the lysosome but in the regulation of transcription.

We agree that our finding that LIR dependent binding of LMX1B to ATG8s relates to transcription not degradation is both interesting and novel, and indeed highlight this as broader paradigm as the primary advance provided by this study. We hope that the overall narrative around this is now very much stronger with additional transcriptional rescue experiments (Fig. 7A, B; Fig. S5F, G), extended and improved luciferase reporter experiments (Fig. 6E-G; Fig. S5E; Fig. S6B), and the LMX1B CRISPR knockout work (see Figs 1 & 7).

Reviewer #2 (Comments to the Authors (Required)):

The study by Jiménez-Moreno and co-authors found that LMX1A and LMX1B, two genes important for midbrain dopaminergic neuronal (mDAN) differentiation and survival (also associated with Parkinson development), can interact with ATG8 family members, can regulate the expression of critical autophagy genes and can protect mDANs from mitochondrial stress. Notably, they provide evidence that ATG8s play a central role acting as cofactors for LMX1B and influencing autophagy gene expression through the activity of this transcriptional factor. This story would be of great interest and generally the data are well shown and of good quality; however, there are some issues described below, which should be addressed to improve the manuscript:

We thank the reviewer for their overwhelmingly positive comments. We also recognise the novelty of this finding and how interactions between ATG8 family members and transcription factors such as LMX1A/LMX1B could have a profound impact on cell and tissue development and stress coping mechanisms. We hope that the additional data we have included will improve the quality and further advance the overall impact of the work.

- Authors described that "LMX1A-B suppression also elevated mitochondrial ROS levels and negatively impacted oxidative phosphorylation, causing reduced basal oxygen consumption rates, maximal respiration"; on the other hand, LMX1A/LMX1B over-expression protected mDNA from rotenone-induced cell death. I think that this is a critical aspect: how do LMX1A-B protect from Rotenone? Are they able to promote mitophagy? LMX1B has a LIR domain and the authors observed that upon 6h of STV both LMX1 translocated from nucleus to cytoplasm; does it play a role as a cargo receptor such as FUNDC1?

This is an interesting suggestion, but we do not believe that LMX1A/LMX1B play any direct receptor/scaffolding roles for mitophagy (e.g., we have never observed mitochondrial coincident staining with LMX1B). Rather, we believe that increased and/or protected expression of stress response genes when these transcription factors are expressed (as previously reported: Doucet-Beaupre et al., 2016 PNAS) is likely due to enhanced neuronal tolerance to oxidative stress, possibly involving non-selective autophagy. Mitophagy may indeed be involved in this process, although we have not directly tested this in the overexpression system.

- They used GFP-LC3: it is not clear if they handled a GFP-LC3 stable cell line or not; as reported in Guidelines for autophagy 2016, "high levels of over-expressed GFP-LC3 can result in its nuclear localisation": I thus recommend to repeat the IP using endogenous LC3B.

This is a valid point. Interestingly, nuclear LC3 does become more apparent in autophagy-null (e.g., ATG3/7 KO) backgrounds, suggesting that this is a feature of native LC3 biology as/when levels of protein and/or autophagy dictate, but one that is enhanced following overexpression. Nuclear roles for ATG8s/autophagy are now being described (e.g., Dou et al., 2015, Nature 527:105-9), but we accept that overexpression of these proteins in cell-lines could indeed influence the normal balance of distribution seen with their endogenous counterparts (true for any overexpressed protein). That said, we have been able to carry out successful IPs of endogenous LC3B pulled down by antibodies against native LMX1B (i.e., both proteins endogenous). This

appears as **Fig. 5E** in the revised manuscript. We achieved this by cross-linking the anti-LMX1B antiserum to magnetic beads. We also managed to IP a small fraction of endogenous LMX1B specifically in the nuclear fraction of cells expressing GFP-LC3B by GFP-TRAP (**Fig. S4A** in the revised manuscript), which although not a native LC3B IP, does at least use endogenous LMX1B.

- Based on Fig 6F (knockdown/rescue experiment), Y309A/L312A LMX1B has an effect comparable to that of wt LMX1B on autophagy gene expression, while Δ 308-317 LMX1B failed to rescue some autophagy targets, indicating that the LIR domain of LMX1B may be fundamental for some targets but not for others; could the authors comment this aspect? Furthermore, they analysed the effect of siLC3B, but I think that they should repeat the experiment by down-regulating the other ATG8 proteins (LMX1B highly interacts with GABARAP and GABARAP-L1 since these could have a stronger effect on the regulation of autophagy gene expression).

We thank the reviewer for highlighting this interesting and important complexity and agree that there seems to be a level of selectivity in those genes whose expression appears to be mediated via the LMX1B LIR and those that does not. To expand on this, and in response to the reviewer's suggestions, we have carried out extended luciferase reporter assays with both Y309/L312 LMX1B and the Δ 308-317 LMX1B construct and have also measured luciferase readouts in siGABARAP-L1 and siLC3A cells (**Fig. 7E-G, Fig. S5E, Fig. S6B** in the revised manuscript). The overall picture is one of different targets being affected by suppression of different ATG8s, with some targets apparently resistant when some ATG8s are suppressed (e.g., ULK1 for siLC3B (**Fig. 6F**); TFEB and PINK1 for siGABARAP-L1 (**Fig. 6G**); all tested for siLC3A (**Fig. S6B**)). This raises the prospect of cells being capable of decoding fluctuating levels of available ATG8s under specific nutrient and/or other stress conditions through transcriptional factor responsive to fine-tune the cellular stress response and/or to target specific classes of selective autophagy substrates. This is an avenue that we are exploring currently in the lab.

Minor points:

- Please improve the quality of some WBs (for example, in Figure 4C). Based on that image, the quantification would almost be impossible.

Selected immunoblots have been improved (e.g., Fig. 4C in the original submission, now **Fig. 4D**).

Reviewer #3 (Comments to the Authors (Required)):

Jimenez-Moreno et al. investigated the regulation of autophagy by the transcription factors LMX1A and 1B, which participate to midbrain dopaminergic neuronal differentiation and survival. The role of these TFs in autophagy regulation was previously demonstrated, thus some of the findings are not entirely novel.

We acknowledge that a previous publication in mouse (Laguna et al., 2015 Nature Neuroscience) concluded that LMX1A/1B are potential autophagy transcription factors. However, here we demonstrate this directly through a variety of orthogonal tests, including for the first time in human neurons, and take this observation much further with mechanistic data on ATG8-LMX1B (1A) interactions via conserved LIRs, and how this shapes the transcriptional profile of these transcription factors.

In the first part of the manuscript the authors demonstrated that LMX1B and A binds to the promoters of several autophagy genes, and that the expression levels of several of them is reduced in cells depleted for LMX1B. The authors also show that silencing of LMX1B reduced LC3 and WIPI2 puncta, although data are not entirely convincing and not supported by a quantitative analysis of autophagy flux (e.g. substrate degradation rate). Thus it is unclear the relevance of LMX1 factors in autophagy.

The reviewer is correct that we did not provide substrate flux assessment in the original submission. We have done this now using LMX1B CRISPR KO HEK293T cells, and the data appear as **Fig. 11**. In brief, we find that basal p62 levels are significantly higher than in control cells, and that levels remain higher than control cells when flux is blocked (BafA1).

The authors also demonstrated that during starvation LMX1 factors translocate from nucleus to cytoplasm, although the kinetics is not entirely clear to me, since biochemical evidences are not entirely in line with immunofluorescent analysis, particularly during acute starvation (2h). Also in this case the functional relevance of this shuttling in the context of autophagy regulation can only be speculated.

We agree that the immunofluorescence and cell fractionation/immunoblotting data are not particularly consistent in this context. We are not overly surprised by this since these techniques are prone to giving different qualitative vs quantitative outcomes. To clarify these observations, we have included improved cell fractionation data with LMX1B in new **Fig. 4D**, and additional FLIP data in **Fig. 4F**. We did attempt cell fractionation at the 6-hour timepoint, but this was not a success because HEK293T cells starved for this length of time were dying and floating away, and were too fragile for reliable cell fractionation.

Lastly, the authors proposed that ATG8 binding stimulates transcription of LMX1 proteins. This is potentially a very novel and original concept. However, this claim is based on very few experimental evidences with results showing at best very marginal differences. Furthermore, functional impact on autophagy flux was not investigated.

We agree with the reviewer that our finding that ATG8 binding stimulates LMX1B-based transcription is novel and original, and would go as far as to say that this could be of major importance for understanding how nutrient flux and cell stress can shape the transcriptional landscape. To increase the impact of this part of the study and provide a more convincing argument, we have substantially expanded the relevant datasets to include: (i) wider analysis of FLAT-luciferase transcription comparing wild type with 2 LIR mutant LMX1Bs (**Fig. 6E-G; Fig. S5E, F; Fig. S6B**); (ii) qRT-PCR in knockdown/rescue settings (**Fig. 6A, B; Fig. 7A-B; Fig. S5F, G**); (iii) autophagic flux assessments in knockout/rescue cells (**Fig. 7C**).

In conclusion I believe that this work is not suitable for publication in JCB.

We are sorry that the reviewer was not convinced about the merit of our work in the original submission. We have added a significant amount of new data that emphasise the novelty of this study and help establish this work as an important advance in our understanding of integrated autophagy stress response pathways. We therefore hope that the additional data will change the referee's overall opinion in doubting the suitability of this work for JCB.

In addition, below I list my technical concerns:

- Fig 1A lacks error bars and statistics.

We have repeated the ChIP experiments in HEKs and in mDANs (see below) several times, with consistent results. We favour showing an indicative experiment with supporting/validation data using si/shRNA and qRT-PCR, as this provides a more mechanistic readout.

- Fig 1G: the co-staining with LMX1B is needed to quantify the number of LC3 puncta in silenced vs non-silenced cells. Currently, the difference are not clear.

We do routinely label for the target protein to determine which cells are siRNA suppressed for analysis because in our hands, transient siRNA transfection does not generate populations of cells with target gene expression either at normal (high) or suppressed (low) low levels. More often there is a general reduction across the field, and this is indeed the case during LMX1B siRNA suppression in HEK293T cells (see **Reviewers' Fig. 3**). For this reason, it would not be possible to identify LMX1B-suppressed cells in a population.

Reviewers' Fig. 3: LMX1B levels in HEK293T cells after siRNA suppression. Example images to the left; quantitation to the right. Bar = 10 μ m. Student's t-test, **** $p < 0.0001$ [these data appeared as Fig. 1F in the original submission].

- ChIP in fig 2 lack error bars and statistics.

Please see the response above.

- Data in fig 3A are not convincing

We regret that the reviewer was not convinced by these data. We took great care to ensure that we were analysing only TH-positive mDANs, and only those expressing the GFP/shRNA construct. We have carried out extensive analyses of autophagy marker validity in human iPSC mDANs (Stathakos et al. 2021, Autophagy) and conclude that the most reliable means of testing autophagy in this setting is through the use of anti-WIP12 antibodies. These require methanol fixing (as does anti-LC3B staining), but give a much more meaningful readout of autophagic activity as the signal to noise is much stronger.

- Fig 4, a time course experiment of nuclear/cytosolic localization should be performed both by IF and WB. Figure 5K shows a different result.

This is a valid suggestion, although we would first stress that qualitative immunofluorescence and quantitative immunoblotting readouts can often give quite different readouts. We carried out a series of additional repeat experiments based on the 2-hour starvation timepoint and present much "cleaner" data (**Fig. 4D**). The quantitation gave the same overall result of increased nuclear LMX1B during early phases of starvation, but the blot is now much nicer. We attempted to generate complimentary data for the 6-hour timepoint, but this proved very challenging because HEK293T cells starved for this length of time were highly fragile and/or were undergoing apoptosis. Consequently, the cell fractionation stage became unreliable due loss of integrity of these stressed cells.

- Immunoprecipitation experiments using nuclear and cytosolic fractions should be supported by appropriate loading control of subcellular markers.

We have provided loading controls for the fractionation experiments using markers for cytosolic (GAPDH) and nuclear (lamin B) fractions.

- Fig 6A, the input of LMX1B have loading problems.

We are not clear what the reviewer is concerned about here. The loading controls for LMX1B-FLAG show alternate bands of different masses because the second in each pair is a deletion construct. Please clarify if there was a separate issue with this figure (new **Fig. 7A**).

January 20, 2023

RE: JCB Manuscript #201910133R-A

Prof. Jon D Lane
University of Bristol
School of Biochemistry School of Medical Sciences University Walk
Bristol BS8 1TD
United Kingdom

Dear Prof. Lane:

Thank you for submitting your revised manuscript entitled "LIR-dependent LMX1B-autophagy crosstalk shapes human midbrain dopaminergic neuronal resilience". The paper has now been seen again by the reviewers and we would be happy to publish your paper in JCB pending final revisions necessary to meet our formatting guidelines (see details below).

As we have discussed, reviewer #1 has raised a few remaining points that we would like for you to address in the final revision. However, as mentioned to you previously, JCB does not allow "data not shown" to be included in any of our papers and so we do not agree with your proposal to remove the modeling data from the paper. Instead, we would prefer for you to leave the modeling data in the paper but move it all to the supplementary information/figures and be sure to explicitly discuss any limitations/caveats to the data in the manuscript.

A. MANUSCRIPT ORGANIZATION AND FORMATTING:

1) Text limits: Character count for Articles and Tools is < 40,000, not including spaces. Count includes the abstract, introduction, results, discussion, and acknowledgments. Count does not include title page, materials and methods, figure legends, references, tables, or supplemental legends. You are slightly over this limit at the moment but we will be able to give you the extra space this time. Please try to be as concise as possible when adding new text, however.

2) Figure formatting: Scale bars must be present on all microscopy images, including inset magnifications. Molecular weight or nucleic acid size markers must be included on all gel electrophoresis.

3) Statistical analysis: Error bars on graphic representations of numerical data must be clearly described in the figure legend. The number of independent data points (n) represented in a graph must be indicated in the legend. Statistical methods should be explained in full in the materials and methods. For figures presenting pooled data the statistical measure should be defined in the figure legends. Please also be sure to indicate the statistical tests used in each of your experiments (both in the figure legend itself and in a separate methods section) as well as the parameters of the test (for example, if you ran a t-test, please indicate if it was one- or two-sided, etc.).

****Also, since you used parametric tests in your study (e.g. t-tests, ANOVA, etc.), you should have first determined whether the data was normally distributed before selecting that test. In the stats section of the methods, please indicate how you tested for normality. If you did not test for normality, you must state something to the effect that "Data distribution was assumed to be normal but this was not formally tested."****

4) Materials and methods: Should be comprehensive and not simply reference a previous publication for details on how an experiment was performed. Please provide full descriptions (at least in brief) in the text for readers who may not have access to referenced manuscripts. The text should not refer to methods "...as previously described."

5) Please be sure to provide the sequences for all of your primers/oligos and RNAi constructs in the materials and methods. You must also indicate in the methods the source, species, and catalog numbers (where appropriate) for all of your antibodies.

6) Microscope image acquisition: The following information must be provided about the acquisition and processing of images:

- a. Make and model of microscope
- b. Type, magnification, and numerical aperture of the objective lenses
- c. Temperature
- d. imaging medium
- e. Fluorochromes

- f. Camera make and model
- g. Acquisition software
- h. Any software used for image processing subsequent to data acquisition. Please include details and types of operations involved (e.g., type of deconvolution, 3D reconstitutions, surface or volume rendering, gamma adjustments, etc.).

7) References: There is no limit to the number of references cited in a manuscript. References should be cited parenthetically in the text by author and year of publication. Abbreviate the names of journals according to PubMed.

****Please note that we do not allow "supplemental references". Please remove this section and add any non-duplicated references to the main reference list.****

8) Supplemental materials: There are normally strict limits on the allowable amount of supplemental data. Articles may usually have up to 5 supplemental figures. At the moment, you currently have 6 such figures (and may need to add another once you move the modeling data, though if you can fit the modeling data into one supplemental figure, that would be great). We will be able to give you the extra figures in this case, but please try to be as concise as possible.

Please also note that tables, like figures, should be provided as individual, editable files. A summary of all supplemental material (that is, in addition to the supplementary figure legends) should appear at the end of the Materials and methods section. Please see any recent JCB paper for an example of this.

9) Conflict of interest statement: JCB requires inclusion of a statement in the acknowledgements regarding competing financial interests. If no competing financial interests exist, please include the following statement: "The authors declare no competing financial interests." If competing interests are declared, please follow your statement of these competing interests with the following statement: "The authors declare no further competing financial interests."

10) A separate author contribution section is required following the Acknowledgments in all research manuscripts. All authors should be mentioned and designated by their first and middle initials and full surnames. We encourage use of the CRediT nomenclature (<https://casrai.org/credit/>).

11) ORCID IDs: ORCID IDs are unique identifiers allowing researchers to create a record of their various scholarly contributions in a single place. At resubmission of your final files, please consider providing an ORCID ID for as many contributing authors as possible.

B. FINAL FILES:

****It is JCB policy that if requested, original data images must be made available to the editors. Failure to provide original images upon request will result in unavoidable delays in publication. Please ensure that you have access to all original data images prior to final submission.****

****The license to publish form must be signed before your manuscript can be sent to production. A link to the electronic license to publish form will be sent to the corresponding author only. Please take a moment to check your funder requirements before choosing the appropriate license.****

Thank you for your attention to these final processing requirements. Please revise and format the manuscript and upload

materials within 7-14 days. If complications arising from measures taken to prevent the spread of COVID-19 will prevent you from meeting this deadline (e.g. if you cannot retrieve necessary files from your laboratory, etc.), please let us know and we can work with you to determine a suitable revision period.

Please contact the journal office with any questions, cellbio@rockefeller.edu.

Thank you for this interesting contribution, we look forward to publishing your paper in Journal of Cell Biology.

Sincerely,

Richard Youle, PhD
Monitoring Editor
Journal of Cell Biology

Tim Spencer, PhD
Executive Editor
Journal of Cell Biology

Reviewer #1 (Comments to the Authors (Required)):

In this revised version the authors have done a lot to respond satisfactorily to the referee comments and I think this story is very interesting and clearly worthy of acceptance after a final revision. I have one main major point that needs to be revised without requiring any experimentations before I think this work is ready for acceptance. This concerns the LIR motif which is not actually a canonical functional LIR motif. This is about avoidance of confusion in the literature. As there is not clear evidence for a bona fide canonical LIR-dependent interaction between the identified region 308 to 317 in LMX1B (identified by loss of binding in IPs and pull down assays) any reference to LIR-dependency is not making sense in a strict meaning. The authors show a lot of data demonstrating interaction with ATG8 proteins and that the 308-317 region is required for this interaction and that the ATG8-interaction is required for LMX1B-mediated transcription for efficient autophagy and cell stress protection. However, point mutations of the aromatic and hydrophobic residues to Ala does not take out the binding to the ATG8s. There is a reduction of binding for LC3B and GABARAPL2 while other ATG8s show unchanged or even increased strong binding to the Y309A/L312A mutant of LMX1B. In the functional assays the authors do not see any effect of the Y309A/L312A mutant while the deletion mutant delta 308-317 do have effects. The authors show that the LIR Docking Site (LDS) is involved in the binding to the LC3B. Taken together this suggests that this is a non-canonical binding going on here. The candidate LIR motif of LMX1B has a tyrosine at the aromatic position that should dock into hydrophobic pocket 1 of the LDS and it has a proline in the core LIR sequence which is usually destabilizing for most LIRs as seen from mutagenesis and binding experiments (i.e. PMID: 31053714 and PMID: 23043107).

1. It is OK if the authors refers to this motif as perhaps "LIR-like" or perhaps better as a non-canonical binding to ATG8s involving the LDS and this needs to be discussed in the paper's Discussion.
2. The title of the paper should be changed by replacing LIR with ATG8 giving this slight but important change of title: ATG8-dependent LMX1B-autophagy crosstalk shapes human midbrain
3. All modeling should be removed from the paper as the structure models are based on a wrong premise since the mutation of Y309 and L312 do not affect binding for most ATG8s. The modeling done based on the FYCO1 LIR interaction even underscores that this likely not a strong LIR since there are very few opportunities for establishment of strong electrostatic interactions. The authors should remove Figures 5H-J and Figure S3.
4. In the Abstract rewrite "Crucially, LIR-dependent ATG8 binding stimulates..." to "Crucially, ATG8 binding stimulates..."
5. In the Abstract ... "bind to multiple ATG8 proteins via LIR-type interactions.... Delete "via LIR-type interactions"
6. Summary: Jiménez-Moreno et al. demonstrate that human ATG8 proteins stimulate transcriptional cell stress protection via LIR-mediated binding to the LMX1B.... Delete "LIR-mediated".
7. On page 4 "both LMX1A and LMX1B bind to autophagy ATG8 family members, with LMX1B interactions dependent on a conserved LIR motif (309YTPL312)." Correct to a "conserved region C-terminal to the homeodomain"
8. Heading on bottom of page 8: "LMX1B contains a LIR motif and bind multiple ATG8 proteins via LIR-type interactions" is misleading and should be corrected.....Suggestion: "LMX1B bind multiple ATG8 proteins via a conserved region containing a LIR-like motif"
9. Heading on page 9: "The LIR-dependent LMX1B-LC3B interaction is location and context dependent" should be changed to "The LMX1B-LC3B interaction is location and context dependent"
10. Also from the heading on page 12" LMX1B LIR-dependent target gene expression ensures a robust autophagy starvation response and protects human mDANs against rotenone toxicity" LIR-dependent" should be removed and the heading rephrased.
11. In the Discussion and other relevant parts of the text the reference to a LIR should also be corrected.

I am sorry if these points may seem to detailed. Doing these corrections do not in my opinion weaken the paper or reduce the interest and impact of this paper as there are ways of interacting with the ATG8 family proteins, besides the well documented

LIR-LDS interactions, we have not yet detailed information on, and we lack structural data for most such interactions with the notable exception of ATG4B-LC3B interaction (PMID: 19322194) and the yeast Hfl1-Atg8 interaction (PMID: 30451685).

Reviewer #2 (Comments to the Authors (Required)):

All requirements were satisfied in full, and I congratulate with the Authors for the significant effort. The paper is now complete, it complies with JCB caliber, and deserves publication as it stands.

RE: JCB Manuscript #201910133R-A (Jiménez-Moreno et al.)—response to Referees/Editor.

Dear Richard & Tim,

Thank you for providing guidance on finalizing our manuscript. We are extremely grateful for the input from yourselves and from the referees—our work is all the stronger for this, and we are delighted to be publishing in JCB.

From here, our responses to the final suggestions by Reviewer #1 and from yourselves are in blue font / Editor and Reviewers' comments are in italics/black font.

Editor's email

Thank you for submitting your revised manuscript entitled "LIR-dependent LMX1B-autophagy crosstalk shapes human midbrain dopaminergic neuronal resilience". The paper has now been seen again by the reviewers and we would be happy to publish your paper in JCB pending final revisions necessary to meet our formatting guidelines (see details below).

We are extremely pleased for this positive outcome and have taken care to adjust our manuscript to meet these formatting guidelines.

As we have discussed, reviewer #1 has raised a few remaining points that we would like for you to address in the final revision. However, as mentioned to you previously, JCB does not allow "data not shown" to be included in any of our papers and so we do not agree with your proposal to remove the modeling data from the paper. Instead, we would prefer for you to leave the modeling data in the paper but move it all to the supplementary information/figures and be sure to explicitly discuss any limitations/caveats to the data in the manuscript.

As agreed in our email exchange, we have adopted terminology such as LIR-like or non-canonical LIR-type binding (involving the LDS) throughout the manuscript, including the title where we replaced "LIR-dependent" with "ATG8-dependent". Further, we have taken the modeling data out of Fig. 5 and moved it to an extended Fig. S3 (rather than refer to it as data not shown). To ensure that this is fully contextualized, we have provided an extended explanation in the Discussion section around the topic of canonical and non-canonical LIR-type binding, as follows: *"To determine the significance of the LMX1A/LMX1B-ATG8 interaction, we first modelled LC3B binding to a LMX1B peptide incorporating the putative LIR in silico, templated on the LC3B-FYCO1 structure. With the clear reduction in binding to LDS mutant LC3B, the modelling data suggested a LIR-to-LDS type interaction (with differences predicted in the positioning of residues upstream of the core LMX1B LIR), so we next used mutagenesis to change the two key LIR-associated residues (the aromatic phenylalanine at Θ_0 and the aliphatic leucine at Γ_3) to alanine (Y309A/L312A). This was sufficient to reduce binding to LC3B and GABARAP-L2, but it did not impact markedly on interactions with other ATG8. Functionally, the Y309A/L312A mutant LMX1B efficiently restored LMX1B target gene expression in knockdown/rescue experiments and drove strong LMX1B FLAT-promoter luciferase expression, whilst providing good protection in rotenone toxicity assays. Together with the observed presence of a destabilising proline within the core LIR-type motif (see (Alemu et al., 2012; Wirth et al., 2019)), and the paucity of acidic residues upstream of the proposed LIR, these observations are indicative of non-canonical or LIR-like binding, such as those detailed in other example ATG8 interactions (e.g., (Liu et al., 2018; Satoo et al., 2009)). We have therefore chosen to use this terminology throughout when referring to LMX1B-ATG8 interactions."*

Formatting points where responses are needed

1) Text limits: You are slightly over this limit at the moment but we will be able to give you the extra space this time. Please try to be as concise as possible when adding new text, however: We have been very careful not to increase the character count, and are grateful for this flexibility.

2) Figure formatting: Completed

- 3) *Statistical analysis*: We have complied with requirements around statistical analysis, including adding a statement on data normality as requested.
- 4) *Materials and methods*: We have extended the section on iPSC culture so as to not simply refer to previous publications.
- 5) Please be sure to provide the sequences for all of your primers/oligos and RNAi constructs in the materials and methods: We provide separate tables for oligonucleotide primers etc.
- 6) *Microscope image acquisition*: To our knowledge this is all now all correct.
- 7) *References*: ****Please note that we do not allow "supplemental references". Please remove this section and add any non-duplicated references to the main reference list. **** We have done this as requested.
- 8) *Supplemental materials*: Articles may usually have up to 5 supplemental figures. At the moment, you currently have 6 such figures (and may need to add another once you move the modeling data, though if you can fit the modeling data into one supplemental figure, that would be great). We will be able to give you the extra figures in this case, but please try to be as concise as possible: We have moved the modelling from Fig. 5 (main text) to an expanded Fig. S3. This means we remain on 6 supplemental figures and are grateful for your flexibility with this.
- 9) *Conflict of interest statement*: This has been completed.
- 10) A separate author contribution section is required following the Acknowledgments in all research manuscripts. This has been carried out.
- 11) ORCID IDs: ORCID IDs are unique identifiers allowing researchers to create a record of their various scholarly contributions in a single place. At resubmission of your final files, please consider providing an ORCID ID for as many contributing authors as possible. This has been completed.

Additionally, JCB encourages authors to submit a short video summary of their work. These videos are intended to convey the main messages of the study to a non-specialist, scientific audience. Think of them as an extended version of your abstract, or a short poster presentation. We encourage first authors to present the results to increase their visibility. The videos will be shared on social media to promote your work. For more detailed guidelines and tips on preparing your video, please visit <https://rupress.org/jcb/pages/submission-guidelines#videoSummaries>. We are currently strongly considering this and will communicate with the office as to whether/when this will be forthcoming.

Reviewer #1 (Comments to the Authors (Required)):

In this revised version the authors have done a lot to respond satisfactorily to the referee comments and I think this story is very interesting and clearly worthy of acceptance after a final revision.

We thank the reviewer for their positive comments and for their input to improving our manuscript.

I have one main major point that needs to be revised without requiring any experimentations before I think this work is ready for acceptance. This concerns the LIR motif which is not actually a canonical functional LIR motif. This is about avoidance of confusion in the literature. As there is not clear evidence for a bona fide canonical LIR-dependent interaction between the identified region 308 to 317 in LMX1B (identified by loss of binding in IPs and pull down assays) any reference to LIR-dependency is not making sense in a strict meaning. The authors show a lot of data demonstrating interaction with ATG8 proteins and that the 308-317 region is required for this interaction and that the ATG8-interaction is required for LMX1B-mediated transcription for efficient autophagy and cell stress protection. However, point

mutations of the aromatic and hydrophobic residues to Ala does not take out the binding to the ATG8s. There is a reduction of binding for LC3B and GABARAPL2 while other ATG8s show unchanged or even increased strong binding to the Y309A/L312A mutant of LMX1B. In the functional assays the authors do not see any effect of the Y309A/L312A mutant while the deletion mutant delta 308-317 do have effects. The authors show that the LIR Docking Site (LDS) is involved in the binding to the LC3B. Taken together this suggests that this is a non-canonical binding going on here. The candidate LIR motif of LMX1B has a tyrosine at the aromatic position that should dock into hydrophobic pocket 1 of the LDS and it has a proline in the core LIR sequence which is usually destabilizing for most LIRs as seen from mutagenesis and binding experiments (i.e. PMID: 31053714 and PMID: 23043107).

We thank the reviewer for this careful and welcome analysis. Broadly, we agree that binding between LMX1B and the ATG8 family has features in common with but also clearly somewhat different from classical LIR-mediated binding. The presence of the proline and the failure of alanine point mutations (Y309A/L312A) to disable binding to most ATG8s certainly implies a role for residues flanking the identified region in this interaction, which in the strictest sense does point to binding better described as “non-canonical” or “LIR-like” with respect to classical LIR containing example proteins. Hence, as reiterated in the following, we have toned-down reference to LIR-mediated binding, and instead use alternative terminology as suggested. This includes an extended section in the Discussion to explain our thinking around the issue of canonical vs. non-canonical LIRs (see below).

1. It is OK if the authors refers to this motif as perhaps “LIR-like” or perhaps better as a **non-canonical binding to ATG8s involving the LDS and this needs to be discussed in the paper's Discussion**.

As indicated above, we will adopt the term “LIR-like” or “non-canonical”, and will refer to the LDS as appropriate throughout the manuscript, and will include an extra section(s) in the Discussion where we will cover this point in the context of published examples of LIR-type interactions.

2. The title of the paper should be changed by replacing LIR with ATG8 giving this slight but important change of title: **ATG8-dependent LMX1B-autophagy crosstalk shapes human midbrain**

The suggested replacement title would be in keeping with the slight change of emphasis in the paper, so we are happy to make this change.

3. **All modeling should be removed from the paper** as the structure models are based on a wrong premise since the mutation of Y309 and L312 do not affect binding for most ATG8s. The modeling done based on the FYCO1 LIR interaction even underscores that this likely not a strong LIR since there are very few opportunities for establishment of strong electrostatic interactions. **The authors should remove Figures 5H-J and Figure S3.**

The modeling has been useful in proposing how the flanking regions of LMX1B are likely to engage with LC3B, compared with other LIRs (namely, FYCO1 and the N-terminal LIR in ATG4B). But we concede that any model can be misleading if not supported by extensive experimental validation; thus, and with the guidance of the JCB office, we have moved the modeling figures to the Supplementary section. We have explained this in an extended paragraph in the Discussion as follows: “To determine the significance of the LMX1A/LMX1B-ATG8 interaction, we first modelled LC3B binding to a LMX1B peptide incorporating the putative LIR in silico, templated on the LC3B-FYCO1 structure. With the clear reduction in binding to LDS mutant LC3B, the modelling data suggested a LIR-to-LDS type interaction (with differences predicted in the positioning of residues upstream of the core LMX1B LIR), so we next used mutagenesis to change the two key LIR-associated residues (the aromatic phenylalanine at Θ_0 and the aliphatic leucine at Γ_3) to alanine (Y309A/L312A). This was sufficient to reduce binding to LC3B and GABARAP-L2, but it did not impact markedly on interactions with other ATG8. Functionally, the Y309A/L312A mutant LMX1B efficiently restored LMX1B target gene expression in knockdown/rescue experiments and drove strong

LMX1B FLAT-promoter luciferase expression, whilst providing good protection in rotenone toxicity assays. Together with the observed presence of a destabilising proline within the core LIR-type motif (see (Alemu et al., 2012; Wirth et al., 2019)), and the paucity of acidic residues upstream of the proposed LIR, these observations are indicative of non-canonical or LIR-like binding, such as those detailed in other example ATG8 interactions (e.g., (Liu et al., 2018; Satoo et al., 2009)). We have therefore chosen to use this terminology throughout when referring to LMX1B-ATG8 interactions."

4. *In the Abstract rewrite "Crucially, LIR-dependent ATG8 binding stimulates..." to "Crucially, ATG8 binding stimulates..."*

This has been carried out as requested.

5. *In the Abstract ..." bind to multiple ATG8 proteins via LIR-type interactions.... Delete "via LIR-type interactions"*

This has been carried out as requested.

6. *Summary: Jiménez-Moreno et al. demonstrate that human ATG8 proteins stimulate transcriptional cell stress protection via LIR-mediated binding to the LMX1B.... Delete "LIR-mediated".*

This has been carried out as requested.

7. *On page 4 "both LMX1A and LMX1B bind to autophagy ATG8 family members, with LMX1B interactions dependent on a conserved LIR motif (309YTPL312)." Correct to a "conserved region C-terminal to the homeodomain"*

This has been carried out as requested.

8. *Heading on bottom of page 8: "LMX1B contains a LIR motif and bind multiple ATG8 proteins via LIR-type interactions" is misleading and should be corrected.....Suggestion: "LMX1B bind multiple ATG8 proteins via a conserved region containing a LIR-like motif"*

This has been carried out as requested.

9. *Heading on page 9: "The LIR-dependent LMX1B-LC3B interaction is location and context dependent" should be changed to "The LMX1B-LC3B interaction is location and context dependent"*

This has been carried out as requested.

10. *Also from the heading on page 12" LMX1B LIR-dependent target gene expression ensures a robust autophagy starvation response and protects human mDANs against rotenone toxicity" LIR-dependent" should be removed and the heading rephrased.*

This has been carried out as requested.

11. *In the Discussion and other relevant parts of the text the reference to a LIR should also be corrected.*

This has been carried out as requested.

I am sorry if these points may seem to detailed. Doing these corrections do not in my opinion weaken the paper or reduce the interest and impact of this paper as there are ways of interacting with the ATG8 family proteins, besides the well documented LIR-LDS interactions, we have not yet detailed information on, and we lack structural data for most such interactions with the notable exception of ATG4B-LC3B interaction (PMID: 19322194) and the yeast Hfl1-Atg8 interaction (PMID: 30451685).

Not a problem at all – we are grateful to the referee for their careful assessment of our paper and welcome the opportunity to clarify our manuscript accordingly.

Reviewer #2 (Comments to the Authors (Required)):

All requirements were satisfied in full, and I congratulate with the Authors for the significant effort. The paper is now complete, it complies with JCB caliber, and deserves publication as it stands. We are extremely grateful for the Reviewer's input and for their very positive comments on our work.